# DETECTING MACHINE-GENERATED TEXTS BY MULTI-POPULATION AWARE OPTIMIZATION FOR MAXIMUM MEAN DISCREPANCY

**Shuhai Zhang**[12]*, **Yiliao Song**[3]*, **Jiahao Yang**[1], **Yuanqing Li**[21†], **Bo Han**[5†], **Mingkui Tan**[14†]
South China University of Technology[1] Pazhou Laboratory[2] The University of Adelaide[3]
Key Laboratory of Big Data and Intelligent Robot, Ministry of Education[4]
Department of Computer Science, Hong Kong Baptist University[5]
sezhangshuhai@mail.scut.edu.cn; mingkuitan@scut.edu.cn

## ABSTRACT

Large language models (LLMs) such as ChatGPT have exhibited remarkable performance in generating human-like texts. However, machine-generated texts (MGTs) may carry critical risks, such as plagiarism issues, misleading information, or hallucination issues. Therefore, it is very urgent and important to detect MGTs in many situations. Unfortunately, it is challenging to distinguish MGTs and human-written texts because the distributional discrepancy between them is often very subtle due to the remarkable performance of LLMs. In this paper, we seek to exploit *maximum mean discrepancy* (MMD) to address this issue in the sense that MMD can well identify distributional discrepancies. However, directly training a detector with MMD using diverse MGTs will incur a significantly increased variance of MMD since MGTs may contain *multiple text populations* due to various LLMs. This will severely impair MMD's ability to measure the difference between two samples. To tackle this, we propose a novel *multi-population* aware optimization method for MMD called MMD-MP, which can *avoid variance increases* and thus improve the stability to measure the distributional discrepancy. Relying on MMD-MP, we develop two methods for paragraph-based and sentence-based detection, respectively. Extensive experiments on various LLMs, *e.g.*, GPT2 and ChatGPT, show superior detection performance of our MMD-MP. The source code is available at https://github.com/ZSHsh98/MMD-MP.

## 1 INTRODUCTION

With the advancement of large language models (LLMs), texts generated by these models, such as GPT3 (Brown et al., 2020), are natural, fluent and of high quality. These machine-generated texts (MGTs) closely resemble human-generated texts (HWTs) and have many promising applications in natural language processing, *e.g.*, text summarization (Liu & Lapata, 2019), dialogue generation (Li et al., 2016) and machine translation (Bahdanau et al., 2014). However, existing LLMs may generate fake news (Zellers et al., 2019), spam (Guo et al., 2020), and phishing (Hong, 2012), suffering from factual errors, hallucination and bias (Zhang et al., 2023b; Li et al., 2023). This poses threats to online information's credibility and security (Liu et al., 2023; Zhou et al., 2023a;b), necessitating advanced MGT detection techniques. Unfortunately, it is challenging to distinguish MGTs and HWTs because the distributional differences between them are inherently subtle (Tian et al., 2023).

To detect MGTs, existing *metric-based methods* (Gehrmann et al., 2019; Mitchell et al., 2023; Solaiman et al., 2019) use some statistics (*e.g.*, log-likelihood) to score the probability of the test texts being MGTs, which is less effective when a large language-domain gap exists between the texts used to train the scoring model and the tested MGTs. Another strategy, *model-based methods* (Solaiman et al., 2019; Guo et al., 2023), relying severely on specific MGT types, struggles to adapt to other types of MGTs. These methods face challenges in effectively capturing the distributional discrepancy between MGTs and HWTs, thus limiting their detection capabilities.

*Equal contribution. †Corresponding author.

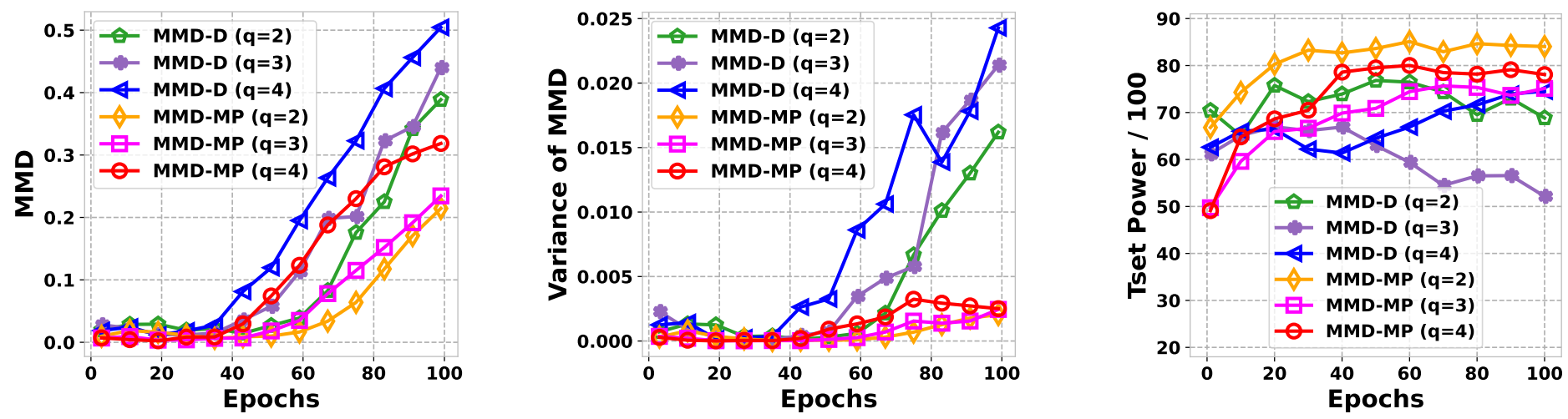

(a) MMD (b) Variance of MMD (c) Test Power of MMD

Figure 1: Illustration of MMD values, MMD variances, and the test power of MMD-D and our MMD-MP during the optimization process. As the number of $S_{\mathbb{Q}}^{tr}$ populations (*i.e.*, $q$) increases, MMD-D shows an increase in MMD, accompanied by a sharp rise in variance, resulting in unstable test power during testing. In contrast, our MMD-MP exhibits minimal variance in MMD values, leading to higher and more stable test power during testing.

In this paper, we seek to exploit *maximum mean discrepancy* (MMD) to address the above issue, given its powerful ability to identify distributional discrepancies (Liu et al., 2020; 2021). *However*, directly applying MMD cannot effectively detect MGTs. This task often involves data from various populations, *e.g.*, texts generated by different LLMs (*e.g.*, GPT-3 (Brown et al., 2020), ChatGPT (OpenAI, 2022)) or different LLM settings (*e.g.*, temperature, top-k sampling (Vilnis et al., 2023)). These populations can substantially differ in language styles and syntactic structures, resulting in significant variations of MGTs. Training a deep kernel MMD (MMD-D, Liu et al. (2020)) under such circumstances will incur the issue of *high variance* (*i.e.*, the large variance of MMD-D in Figure 1 (b)). This means the estimated discrepancy between HWTs and MGTs fluctuates considerably and thus lead to unreliable and unstable detection (the low test power of MMD-D in Figure 1 (c)).

This paper pioneers exploring the optimization mechanism of kernel-based MMD. As we train the kernel with data from multiple populations, the estimated MMD increases with its variance growing significantly (see Figures 1 (a)-(b) and more explanations in Section 2.3). This phenomenon arises due to the *intra-class distance* within MGTs in kernel-based MMD's optimization objective. This distance largely hinders the optimization process that aims to aggregate MGTs, resulting in highly fluctuating MMD for MGT detection. In this paper, we propose a novel **multi-population aware optimization method for MMD** called MMD-MP, which uses a multi-population proxy to remove the constraint on aggregating all instances in MGTs. In this way, we can achieve a low variance of the MMD between MGTs and HWTs, resulting in more stable discrepancy estimation and more reliable detection (see MMD-MP results in Figures 1 (b)-(c)). Furthermore, with the trained deep kernel, we develop two approaches for paragraph-based detection and sentence-based detection, respectively. Empirical evidence on various LLMs such as ChatGPT, GPT2 series, GPT3 series and GPT-Neo series cexhibits the superiority of our methods. Our contributions are summarized as:

1) We delve into the optimization mechanism of MMD and reveal that high variance of the MMD when handling training data from multiple different populations can result in an unstable discrepancy estimation for MGT detection.

2) We propose a novel multi-population aware optimization method for training kernel-based MMD (called MMD-MP), which can alleviate the poor optimization of MMD-D and improve the stability of discrepancy measures.

3) Relying on the proposed MMD-MP, we devise two novel MGT detection methods. Extensive experiments across numerous LLMs, including ChatGPT, GPT2 series, GPT3 series, GPT-Neo series, demonstrate that our methods consistently outperform existing baselines.

## 2 Preliminaries and Motivations

### 2.1 Preliminaries

**Two-sample test (2ST).** Let $\mathbb{P}$, $\mathbb{Q}$ be Borel probability measures on $\mathcal{X}\subset\mathbb{R}^d$. We observe *independent identically distributed* (IID) data $S_{\mathbb{P}}=\{\mathbf{x}_i\}_{i=1}^n\sim\mathbb{P}^n$ and $S_{\mathbb{Q}}=\{\mathbf{y}_j\}_{j=1}^m\sim\mathbb{Q}^m$. 2ST aims to determine if $\mathbb{P}$ and $\mathbb{Q}$ come from the same distribution, *i.e.*, $\mathbb{P}=\mathbb{Q}$ (Borgwardt et al., 2006; Liu et al., 2020).

**Single-instance detection (SID).** Let $\mathbb{P}$ be a Borel probability measure on $\mathcal{X}\subset\mathbb{R}^d$ and *IID* observations $S_{\mathbb{P}}=\{\mathbf{x}_i\}_{i=1}^n\sim\mathbb{P}^n$, SID aims to tell if the test instance $\tilde{\mathbf{y}}$ is from the distribution $\mathbb{P}$.

**Maximum mean discrepancy.** Following Gretton et al. (2012); Liu et al. (2020), *maximum mean discrepancy* (MMD) aims to measure the closeness between two distributions, which is defined as:

**Definition 1.** *Let $k : \mathcal{X}\times\mathcal{X}\to\mathbb{R}$ be the bounded kernel of a reproducing kernel Hilbert space $\mathcal{H}_k$, $\mathcal{F}$ be a class of functions $f : \mathcal{X}\to\mathbb{R}$, and $X\sim\mathbb{P}, Y\sim\mathbb{Q}$ be two random variables,*

$$\mathrm{MMD}\left(\mathbb{P},\mathbb{Q};\mathcal{H}_k\right)=\sup_{f\in\mathcal{F},\|f\|_{\mathcal{H}_k}\leq 1}|\mathbb{E}[f(X)]-\mathbb{E}[f(Y)]|=\sqrt{\mathbb{E}\left[k\left(X,X'\right)+k\left(Y,Y'\right)-2k(X,Y)\right]}.$$

Intuitively, we could view $k(X, X')$ or $k(Y, Y')$ as an *intra-class* distance and $k(X, Y)$ as an *inter-class* distance. When $n{=}m$, we can estimate MMD via a U-statistic estimator unbiased for $\mathrm{MMD}^2$:

$$\widehat{\mathrm{MMD}}_u^2(S_{\mathbb{P}},S_{\mathbb{Q}};k)=\frac{1}{n(n-1)}\sum_{i\neq j}H_{ij},\ \text{ where } H_{ij}:=k(\mathbf{x}_i,\mathbf{x}_j)-k(\mathbf{x}_i,\mathbf{y}_j)-k(\mathbf{y}_i,\mathbf{x}_j)+k(\mathbf{y}_i,\mathbf{y}_j). \quad (1)$$

In this paper, we consider a kernel-based MMD (Liu et al., 2020), where the kernel is defined as:

$$k_\omega(\mathbf{x},\mathbf{y})=[(1-\epsilon)\kappa(\phi_{\hat{f}}(\mathbf{x}),\phi_{\hat{f}}(\mathbf{y}))+\epsilon]q(\hat{f}(\mathbf{x}),\hat{f}(\mathbf{y})), \quad (2)$$

where $\epsilon\in(0,1)$, $\phi_{\hat{f}}(\mathbf{x})=\phi(\hat{f}(\mathbf{x}))$ is a deep neural network with feature extractor $\hat{f}$, $\kappa$ and $q$ are Gaussian kernels with bandwidth $\sigma_\phi$ and bandwidth $\sigma_q$, respectively, *e.g.*, $\kappa(a,b)=\exp\left(-\|a-b\|^2/2\sigma_\phi^2\right)$. Since $\hat{f}$ is fixed, the set of parameters of $k_\omega$ is $\omega=\{\epsilon,\phi,\sigma_\phi,\sigma_q\}$.

**Test power.** Test power is the probability of rejecting the null hypothesis ($\mathfrak{H}_0 : \mathbb{P}=\mathbb{Q}$) when $\mathbb{P}\neq\mathbb{Q}$. A higher test power indicates a greater level of certainty regarding the distributional discrepancy. For reasonably large $n$, Liu et al. (2020) find that the power is nearly proportional to

$$J(\mathbb{P},\mathbb{Q};k_\omega)=\mathrm{MMD}^2(\mathbb{P},\mathbb{Q};k_\omega)/\sigma_{\mathfrak{H}_1}(\mathbb{P},\mathbb{Q};k_\omega),\ \text{where }\ \sigma_{\mathfrak{H}_1}^2:=4\left(\mathbb{E}\left[H_{ij}H_{i\ell}\right]-\mathbb{E}\left[H_{ij}\right]^2\right).$$

We can estimate $J(\mathbb{P},\mathbb{Q};k_\omega)$ with a regularized estimator by

$$\hat{J}(S_{\mathbb{P}},S_{\mathbb{Q}};k_\omega)=\frac{\widehat{\mathrm{MMD}}_u^2(S_{\mathbb{P}},S_{\mathbb{Q}};k_\omega)}{\sqrt{\hat{\sigma}_{\mathfrak{H}_1}^2(S_{\mathbb{P}},S_{\mathbb{Q}};k_\omega)+\lambda}},\ \text{where }\ \hat{\sigma}_{\mathfrak{H}_1}^2:=\frac{4}{n^3}\sum_{i=1}^{n}\left(\sum_{j=1}^{n}H_{ij}\right)^2-\frac{4}{n^4}\left(\sum_{i=1}^{n}\sum_{j=1}^{n}H_{ij}\right)^2. \quad (3)$$

### 2.2 High Variance Problem of Kernel-based MMD in Multiple Populations

In practice, we may collect training data $S_{\mathbb{Q}}^{tr}$ with multiple different populations from a mixture of texts generated by different LLMs such as GPT3 (Brown et al., 2020), ChatGPT (OpenAI, 2022). Due to the diverse language styles generated by different language models, this can result in significant variations in the generated text. Under such circumstances, although we can maximize the criterion $\hat{J}$ (3) to optimize the kernel $k_\omega$, a high-variance discrepancy exists.

To validate the above phenomenon, we demonstrate the MMD value and its variance during training by maximizing Eqn. (3) under different numbers of $S_{\mathbb{Q}}^{tr}$ populations (*i.e.*, $q$). According to Figures 1 (a)-(b), the MMD between $S_{\mathbb{P}}^{tr}$ and $S_{\mathbb{Q}}^{tr}$ for MMD-D increases, which is desirable for MGT detection, but its variance simultaneously increases, which will deteriorate the detection performance. The impact of this phenomenon worsens with the increase of $q$. This indicates that the population $S_{\mathbb{Q}}^{tr}$ with larger variations makes the optimization of the original MMD more challenging.

**High variance causes poor optimization of kernel-based MMD.** Although kernel-based MMD is widely used to identify distributional discrepancies, limited literature has explored its optimization mechanism, possibly due to its complex mathematical format. Specifically, when maximizing $\hat{J}$ in Eqn. (3), it is challenging to determine the individual changes of the MMD value and its variance, resulting in intricate analyses of each term in MMD. To address this, we decompose the variance of MMD as below and conduct empirical studies to demonstrate the trends of each component and their variances during training in Figure 2. Further detailed analyses are provided in Section 2.3.

**Components in MMD's variance.** We introduce $H^*{=}k_\omega(\mathbf{x},\mathbf{x}')-k_\omega(\mathbf{x},\mathbf{y}')-k_\omega(\mathbf{y}',\mathbf{x})$ with its variance $\mathrm{Var}(\mathbb{E}[H^*]){=}\mathrm{Var}(\mathbb{E}[k_\omega(\mathbf{x},\mathbf{x}')]){-}2\mathrm{Cov}(\mathbb{E}[k_\omega(\mathbf{x},\mathbf{x}')],\mathbb{E}[2k_\omega(\mathbf{x},\mathbf{y})]){+}\mathrm{Var}(\mathbb{E}[2k_\omega(\mathbf{x},\mathbf{y})])$. Here, $\mathbb{E}$ denotes taking expectations across two populations sampled from MGTs and HWTs and Var denotes taking variances within these sampled populations. The variance of MMD can then be decomposed into: $\mathrm{Var}(\mathbb{E}[H^*]){+}\mathrm{Var}(\mathbb{E}[k_\omega(\mathbf{y},\mathbf{y}')]){+}2\mathrm{Cov}(\mathbb{E}[H^*],\mathbb{E}[k_\omega(\mathbf{y},\mathbf{y}')])$. By this decomposition, we find the changes of MMD's variance are essentially from the changes of variance of $k_\omega(\mathbf{x},\mathbf{x}')$, $k_\omega(\mathbf{x},\mathbf{y})$ and $k_\omega(\mathbf{y},\mathbf{y}')$, presented as kxx, kxy and kyy in Figure 2.

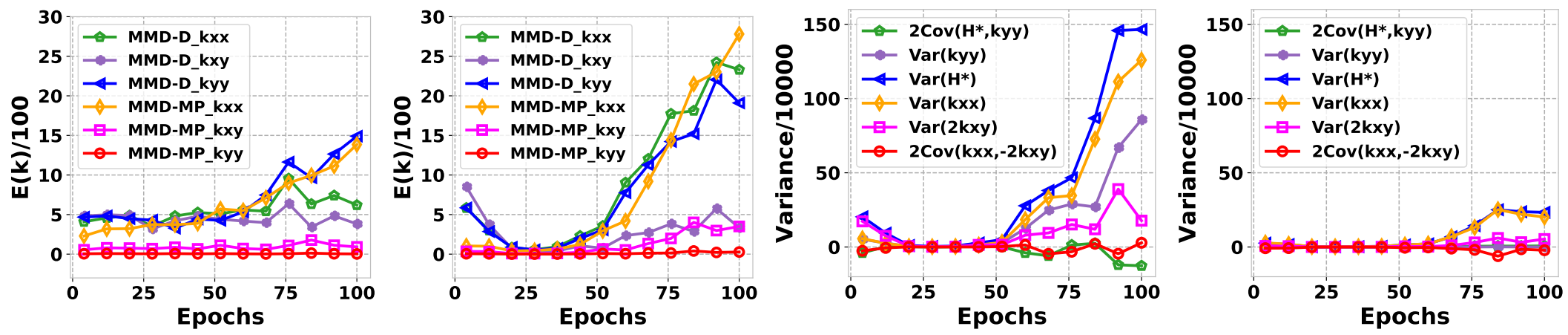

(a) $\mathbb{E}(k)$, $q$=1 (b) $\mathbb{E}(k)$, $q$=3 (c) Var(MMD-D), $q$=3 (d) Var(MMD-MP), $q$=3

Figure 2: $\mathbb{E}(k)$ in MMD and their variances under two optimization methods (MMD-MP is ours). Subfigures (a) and (b) depict the value of each $\mathbb{E}(k)$ in MMD during training by MMD-D and MMD-MP with $q$=1 and $q$=3, respectively. Subfigures (c) and (d) illustrate the variances of some terms associated with MMD, *i.e.*, $\sigma^2_{\mathfrak{H}_1}$ when training by MMD-D and MMD-MP, respectively.

### 2.3 Optimization Mechanism of Kernel-based MMD

We conclude that we should exclude the intra-class distance in $S^{tr}_{\mathbb{Q}}$ during the optimization. To elucidate this, we illustrate some critical observations, followed by explaining these phenomena.

**Observations: i)** In Figures 2 (a)-(b), both $\mathbb{E}[k_\omega(\mathbf{x},\mathbf{x}')]$ and $\mathbb{E}[k_\omega(\mathbf{y},\mathbf{y}')]$ exhibit generally increasing trends, while $\mathbb{E}[k_\omega(\mathbf{x},\mathbf{y})]$ shows relatively minor changes. **ii)** As the number of populations $q$ increases, $\mathbb{E}[k_\omega(\mathbf{y},\mathbf{y}')]$ becomes smaller than $\mathbb{E}[k_\omega(\mathbf{x},\mathbf{x}')]$, and their gap between them widens. **iii)** In Figure 2 (c), the variance of MMD is mainly determined by the variances of $\mathbb{E}[k_\omega(\mathbf{x},\mathbf{x}')]$ and $\mathbb{E}[k_\omega(\mathbf{y},\mathbf{y}')]$ rather than other terms. **iv)** When MGTs in $S^{tr}_{\mathbb{Q}}$ comprise multiple distinct populations (*e.g.*, $q=3$ in Figure 2 (c)), $\mathbb{E}[k_\omega(\mathbf{y},\mathbf{y}')]$ optimized by MMD-D has a significant variance, as well as $\mathbb{E}[k_\omega(\mathbf{x},\mathbf{x}')]$, which is consistent with the results of different $q$ in Appendix H.

**Explanations: First**, **i)** and **ii)** indicate that as $q$ increases, aggregating instances in $S^{tr}_{\mathbb{Q}}$ (MGTs) is more challenging than aggregating instances in $S^{tr}_{\mathbb{P}}$ (HWTs) when using Gaussian kernel to optimize both $k_\omega(\mathbf{x},\mathbf{x}')$ and $k_\omega(\mathbf{y},\mathbf{y}')$ simultaneously since optimizing smaller $\mathbb{E}\ k_\omega$ is more challenging. **Second**, **iii)** and **iv)** indicate that the objective $\hat{J}$ (3) affects the optimization of each term regarding $S^{tr}_{\mathbb{P}}$ and $S^{tr}_{\mathbb{Q}}$ in a similar manner. Thus, the characteristics of their distributions after mapping by the same kernel function, such as the mean and variance of $k_\omega$, exhibit similar changing trends.

**Furthermore**, the optimized kernel function $\mathbb{E}[k_\omega(\mathbf{y},\mathbf{y}')]$ will not only a) map random pairwise MGT instances $(\mathbf{y},\mathbf{y}')$ in $S^{tr}_{\mathbb{Q}}$ close to each other, making the mapped MGTs *more uniform*; but also b) enforce implicit *"pairing rules"* for aggregating MGTs in $S^{tr}_{\mathbb{Q}}$. These rules are shared to pair HWT instances in $S^{tr}_{\mathbb{P}}$ throughout optimization. When MGTs in $S^{tr}_{\mathbb{Q}}$ comprise different populations, the differences in $S^{tr}_{\mathbb{Q}}$-pairs might be large. Applying the paring rules may inadvertently map HWT instances in $S^{tr}_{\mathbb{P}}$ far from their center, leading to increased fluctuations of $k_\omega(\mathbf{x},\mathbf{x}')$ and thus larger $\mathrm{Var}(\mathbb{E}[k_\omega(\mathbf{x},\mathbf{x}')])$. Similarly, the pairing rules for HWTs in $S^{tr}_{\mathbb{P}}$ negatively affect MGTs in $S^{tr}_{\mathbb{Q}}$ but to a lesser extent because $S_{\mathbb{P}}$ is *IID*, meaning $S_{\mathbb{P}}$-instances share more similar statistical characteristics. Pairing rules for HWTs in $S^{tr}_{\mathbb{P}}$ do not need to be as strong as those for aggregating non-*IID* MGTs in $S^{tr}_{\mathbb{Q}}$. **Therefore, we will exclude the intra-class distance in $S^{tr}_{\mathbb{Q}}$ associated with $\mathbb{E}[k_\omega(\mathbf{y},\mathbf{y}')]$ throughout optimization.** We also explore the case of excluding $\mathbb{E}[k_\omega(\mathbf{x},\mathbf{x}')]$ in Appendix G.

## 3 Proposed Methods

### 3.1 Problem Definition

**Problem Definition.** *Let $\mathbb{P}$ be a Borel probability measure on a separable metric space $\mathcal{X}\subset\mathbb{R}^d$ and IID observations $S_{\mathbb{P}}=\{\mathbf{x}_i\}_{i=1}^n$ from the HWT distribution $\mathbb{P}$, we aim to tell if the upcoming data $S_{\mathbb{Q}}=\{\mathbf{y}_j\}_{j=1}^m$ is from the distribution $\mathbb{P}$. Note that $S_{\mathbb{Q}}$ can be HWTs or MGTs generated by multiple different LLMs. When $m$=1, the problem can be regarded as a single-instance detection task.*

**Challenges of MGT detection.** The distinctions between HWTs and MGTs (*e.g.*, text from LLMs like GPT-4) are inherently small, especially in shorter texts like single sentences, making it challenging to distinguish between them. Moreover, the diversity of LLMs leads to significant variations in the generated language style, which further increases the difficulty of MGT detection. Although

deep kernel MMD (MMD-D) is effective in measuring distributional discrepancies, texts generated by multiple LLMs with substantial variations pose challenges in training the deep kernel, *e.g.*, *high variance* of the MMD. Such high variance in MMD will lead to unstable estimations of distributional discrepancies, ultimately resulting in unsatisfactory performance in MGT detection.

### 3.2 MMD-MP FOR MGT DETECTION

As aforementioned, we do not consider optimizing the intra-class distance in $S^{tr}_{\mathbb{Q}}$. Instead, we propose a **multi-population aware optimization for kernel-based MMD (MMD-MP)** with a proxy MPP by maximizing a novel objective as Eqn. (4), and show the training algorithm in Algorithm 1.

$$J(\mathbb{P}, \mathbb{Q}; k_\omega) = \mathrm{MPP}(\mathbb{P}, \mathbb{Q}; k_\omega)/\sigma_{\mathfrak{H}_1^*}(\mathbb{P}, \mathbb{Q}; k_\omega), \tag{4}$$

$$\mathrm{MPP}\left(\mathbb{P}, \mathbb{Q}; \mathcal{H}_k\right) := \mathbb{E}\left[k_\omega\left(X, X'\right) - 2k_\omega(X, Y)\right]. \tag{5}$$

Empirically, we can approximate MPP with an unbiased estimator

$$\widehat{\mathrm{MPP}}_u(S_{\mathbb{P}}, S_{\mathbb{Q}}; k_\omega) = \frac{1}{n(n-1)} \sum_{i \neq j} H^*_{ij}, \text{ where } H^*_{ij} := k_\omega(\mathbf{x}_i, \mathbf{x}_j) - k_\omega(\mathbf{x}_i, \mathbf{y}_j) - k_\omega(\mathbf{y}_i, \mathbf{x}_j). \tag{6}$$

Moreover, we can estimate Eqn. (4) by

$$\hat{J}(S_{\mathbb{P}}, S_{\mathbb{Q}}; k_\omega) = \frac{\widehat{\mathrm{MPP}}_u(S_{\mathbb{P}}, S_{\mathbb{Q}}; k_\omega)}{\sqrt{\hat{\sigma}^2_{\mathfrak{H}_1^*}(S_{\mathbb{P}}, S_{\mathbb{Q}}; k_\omega) + \lambda}}, \tag{7}$$

$$\hat{\sigma}^2_{\mathfrak{H}_1^*} := \frac{4}{n^3} \sum_{i=1}^{n} \left( \sum_{j=1}^{n} H^*_{ij} \right)^2 - \frac{4}{n^4} \left( \sum_{i=1}^{n} \sum_{j=1}^{n} H^*_{ij} \right)^2. \tag{8}$$

We next provide some theoretical analyses to elaborate the objective function Eqn. (4).

**Algorithm 1** Training deep kernel with MMD-MP

**Input:** $S^{tr}_{\mathbb{P}}$, $S^{tr}_{\mathbb{Q}}$, a frozen feature extractor $\hat{f}$;
$\omega \leftarrow \omega_0$; $\lambda \leftarrow 10^{-8}$;
**for** $T = 1, 2, \ldots, T_{max}$ **do**
  $k_\omega \leftarrow$ kernel function using Eqn. (2);
  $M(\omega) \leftarrow \widehat{\mathrm{MPP}}_u(S^{tr}_{\mathbb{P}}, S^{tr}_{\mathbb{Q}}; k_\omega)$ using Eqn. (6);
  $V_\lambda(\omega) \leftarrow \hat{\sigma}^2_{\mathfrak{H}_1^*}(S^{tr}_{\mathbb{P}}, S^{tr}_{\mathbb{Q}}; k_\omega)$ using Eqn. (8);
  $\hat{J}_\lambda(\omega) \leftarrow M(\omega)/\sqrt{V_\lambda(\omega)}$ as in Eqn. (7);
  $\omega \leftarrow \omega + \eta \nabla_{\text{Adam}} \hat{J}_\lambda(\omega)$;
**end for**
**Output:** $k_\omega$

Unlike MMD (Borgwardt et al., 2006; Gretton et al., 2012), the proxy MPP in Eqn. (6) does not incorporate $k_\omega(\mathbf{y}, \mathbf{y})$ related to $S_{\mathbb{Q}}$. However, $\widehat{\mathrm{MPP}}_u$ is still a $U$-statistic (Serfling, 2009) like $\widehat{\mathrm{MMD}}^2_u$, with numerous desirable statistical properties that facilitate convenient theoretical analysis. Note that although maximizing $\widehat{\mathrm{MPP}}_u$ for kernel training is straightforward, it ignores the variance and could lead to an unstable discrepancy (see more details in Appendix E). To address this, we analyze the asymptotics of $\widehat{\mathrm{MPP}}_u$ and derive its test power as follows.

**Proposition 1.** *(Asymptotics of $\widehat{\mathrm{MPP}}_u$) Under the alternative $\mathfrak{H}_1 : \mathbb{P} \neq \mathbb{Q}$, based on a standard central limit theorem, we have:*

$$\sqrt{n}(\widehat{\mathrm{MPP}}_u - \mathrm{MPP}) \xrightarrow{d} \mathcal{N}(0, \sigma^2_{\mathfrak{H}_1^*}), \tag{9}$$

*where $\sigma^2_{\mathfrak{H}_1^*} := 4\left(\mathbb{E}[H^*_{12}H^*_{13}] - \mathbb{E}[H^*_{12}]^2\right)$, $H^*_{12}$, $H^*_{13}$ denote different $H^*_{ij}$.*

**Corollary 1.** *(Test power of $\widehat{\mathrm{MPP}}_u$) For reasonably large $n$, the probability of rejecting the null hypothesis $\mathfrak{H}_0 : \mathbb{P} = \mathbb{Q}$ when $\mathbb{P} \neq \mathbb{Q}$ is given by:*

$$\mathrm{Pr}^{\mathrm{MPP}}_{\mathfrak{H}_1^*, r} \to \Phi\Big(\frac{\sqrt{n}(\mathrm{MPP} + R(S_{\mathbb{Q}}))}{\sigma_{\mathfrak{H}_1^*}} - \frac{r}{\sqrt{n}\,\sigma_{\mathfrak{H}_1^*}}\Big), \tag{10}$$

*where $\mathrm{Pr}^{\mathrm{MPP}}_{\mathfrak{H}_1^*, r} := \mathrm{Pr}\left(n\left[\widehat{\mathrm{MPP}}_u + R(S_{\mathbb{Q}})\right] > r\right)$ and $R(S_{\mathbb{Q}}) = \frac{1}{n(n-1)} \sum_{i \neq j} k_\omega(\mathbf{y}_i, \mathbf{y}_j) > 0$, $\Phi$ is the standard normal cumulative distribution function.*

**Remark 2** *Note that we do not exclude the term $R(S_{\mathbb{Q}})$ in Eqn. (10) due to the uncertain convergence of $n\widehat{\mathrm{MPP}}_u$ (which could be related to the kernel $k_\omega$) when $\mathbb{P}=\mathbb{Q}$. Instead, $n\widehat{\mathrm{MMD}}^2_u = n[\widehat{\mathrm{MPP}}_u + R(S_{\mathbb{Q}})]$ has been proven to be convergent (Gretton et al. (2012), Theorem 12). This enables us to find an approximate power with a rejection threshold as $r$ (Liu et al., 2020).*

**Algorithm 2** Testing with MMD-MP for 2ST

**Input:** Testing texts $S^{te}_{\mathbb{P}}$, $S^{te}_{\mathbb{Q}}$, $\hat{f}$, $k_\omega$;
$est \leftarrow \widehat{\mathrm{MMD}}^2_u(S^{te}_{\mathbb{P}}, S^{te}_{\mathbb{Q}}; k_\omega)$ using Eqn. (1);
**for** $i = 1, 2, \ldots, n_{perm}$ **do**
  Shuffle $S^{te}_{\mathbb{P}} \cup S^{te}_{\mathbb{Q}}$ into $S_X$ and $S_Y$;
  $perm_i \leftarrow \widehat{\mathrm{MMD}}^2_u(S_X, S_Y; k_\omega)$ using Eqn. (1);
**end for**
**Output:** $p$-value $\frac{1}{n_{perm}} \sum_{i=1}^{n_{perm}} \mathbf{1}(perm_i \geq est)$

**Algorithm 3** Testing with MMD-MP for SID

**Input:** Referenced HWT $S^{re}_{\mathbb{P}}$, testing texts $S^{te}_{\mathbb{P}}$, $S^{te}_{\mathbb{Q}}$;
**for** $\mathbf{x}_i, \mathbf{y}_j$ in $S^{te}_{\mathbb{P}}$, $S^{te}_{\mathbb{Q}}$ **do**
  $P_i \leftarrow \widehat{\mathrm{MMD}}^2_b(S^{re}_{\mathbb{P}}, \{\mathbf{x}_i\}; k_\omega)$ using Eqn. (11);
  $Q_j \leftarrow \widehat{\mathrm{MMD}}^2_b(S^{re}_{\mathbb{P}}, \{\mathbf{y}_j\}; k_\omega)$ using Eqn. (11);
**end for**
**Output:** AUROC value with two sets $\{P_i\}, \{Q_j\}$

Corollary (1) shows that, given $r$ and $\sigma_{\mathfrak{H}_1^*}$ being constants, for reasonably large $n$, the test power of MPP is dominated by the first term inside $\Phi$. As suggested by Section 2.3, when removing the intra-class distance in $S^{tr}_{\mathcal{Q}}$ (*i.e.*, $R(S_{\mathbb{Q}}){>}0$), we last optimize Eqn. (4) for MGT detetcion.

We now study the uniform convergence of our proposed optimization function as follows.

**Theorem 1.** ***(Uniform bound of MMD-MP)*** *Let $\omega$ parameterize uniformly bounded kernel functions $k_\omega$ in a Banach space of dimension $D$ with $\|\omega\| \leq R_\Omega$, $k_\omega$ be uniformly bounded by $\sup_{\omega\in\Omega} \sup_{\mathbf{x},\mathbf{x}'\in\mathcal{X}} k_\omega(\mathbf{x},\mathbf{x}') \leq \nu$ with $L_k$-Lipschitz, i.e., $|k_\omega(\mathbf{x},\mathbf{x}') - k_{\omega'}(\mathbf{x},\mathbf{x}')| \leq L_k\|\omega - \omega'\|$. Let $\bar{\Omega}_s$ be a set of $\omega$ for which $\sigma^2_{\mathfrak{H}_1^*} \geq s^2 > 0$. Taking $\lambda = n^{-1/3}$, with probability at least $1-\delta$, we have*

$$\sup_{\omega\in\bar{\Omega}_s} \|\hat{J}(S_{\mathbb{P}}, S_{\mathbb{Q}}; k_\omega) - J(\mathbb{P}, \mathbb{Q}; k_\omega)\| = \mathcal{O}\left(\frac{\nu}{s^2 n^{1/3}}\left[\nu^2\sqrt{D\log(R_\Omega n) + \log\frac{1}{\delta}} + \nu L_k + \frac{1}{s}\right]\right).$$

Detailed constants and proofs are given in Appendix A.3. Theorem 1 shows that our estimator $\hat{J}(S_{\mathbb{P}}, S_{\mathbb{Q}}; k_\omega)$ converges uniformly over a ball in parameter space as $n$ increases. With enough training data, the estimator converges the optimal solution if the best kernel exists.

### 3.3 Exploring MMD-MP for MGT Detections

We consider MGT detection in two scenarios: paragraph-based detection and sentence-based detection. The former aims to detect whether the test paragraph follows the distribution of human-written paragraphs. We address this as a two-sample test. The latter focuses on distinguishing one single machine-generated sentence from HWTs. We consider this as a single-instance detection task.

**MGT Detection Under two-sample test (2ST).** For paragraph-based detection, we consider each sentence within the paragraph as an instance. The detailed procedure for 2ST using MMD-MP can be found in Algorithm 2. Note that we only optimize the kernel using MMD-MP during training and employ the MMD instead of the MPP to measure the distance between $S_{\mathbb{P}}$ and $S_{\mathbb{Q}}$ during testing. The rationale behind this is that we prefer the distance between $S_{\mathbb{P}}$ and $S_{\mathbb{Q}}$ to be zero when $\mathbb{P}{=}\mathbb{Q}$, rather than a negative value, *i.e.*, $-R(S_{\mathbb{Q}}){<}0$. Empirically, the performance of these two strategies is almost identical. We defer more discussion in Appendix F.

**MGT detection under single-instance detection (SID).** While paragraph-based detection is widely employed, there exist practical applications that require single-sentence detection, *e.g.*, online content filtering or false information recognition. Despite numerous works have shown MMD as a powerful means to measure the discrepancy between two distributions or populations, we still hope it can be employed to single-instance detection due to the powerful deep kernel. To achieve this, with a trained kernel, we calculate the distance between a set of referenced HWTs $S^{re}_{\mathbb{P}}$ and a test text $\tilde{\mathbf{y}}$ with Eqn. (11). The detailed procedure for SID using MMD-MP is shown in Algorithm 3.

$$\widehat{\mathrm{MMD}}^2_b(S^{re}_{\mathbb{P}}, \{\tilde{\mathbf{y}}\}; k_\omega) = \frac{1}{n^2}\sum_{i,j=1}^{n} k_\omega(\mathbf{x}_i, \mathbf{x}_j) - \frac{2}{n}\sum_{i=1}^{n} k_\omega(\mathbf{x}_i, \tilde{\mathbf{y}})) + k_\omega(\tilde{\mathbf{y}}, \tilde{\mathbf{y}})). \tag{11}$$

**Advantages of MMD-MP for MGT Detection over MMD-D.** We highlight two key benefits of MMD-MP over MMD-D: 1) **More stable discrepancy estimation**: While MMD-D has a similar $\mathbb{E}[k(\mathbf{x}, \mathbf{x})]$ with MMD-MP, its variance is much greater than MMD-MP (see Figures 2 (a), (c)-(d)), indicating that MMD-D exhibits poorer aggregation effects for $S^{tr}_{\mathbb{P}}$ compared to MMD-MP. Moreover, training MMD using MMD-MP results in a significantly lower variance (see Figures 2 (c)-(d)), mitigating the challenge of high variance in MMD optimization and thereby enhancing the

Table 1: Test power/100 on HC3 given $3,100$ processed paragraphs in training data.

| Method | ChatGPT | GPT3-S | Neo-S | ChatGPT
Neo-S | ChatGPT
GPT3-S |
|---|---|---|---|---|---|
| C2ST-S | $62.83_{\pm 0.90}$ | $43.64_{\pm 5.92}$ | $30.68_{\pm 2.37}$ | $34.62_{\pm 2.73}$ | $46.66_{\pm 2.95}$ |
| C2ST-L | $89.82_{\pm 1.02}$ | $75.74_{\pm 4.90}$ | $60.97_{\pm 1.87}$ | $68.50_{\pm 1.81}$ | $78.22_{\pm 3.12}$ |
| MMD-O | $26.43_{\pm 1.40}$ | $21.17_{\pm 3.12}$ | $19.83_{\pm 2.81}$ | $25.23_{\pm 0.47}$ | $25.18_{\pm 1.41}$ |
| MMD-D | $91.76_{\pm 1.58}$ | $86.98_{\pm 2.53}$ | $75.45_{\pm 4.96}$ | $86.44_{\pm 1.07}$ | $91.46_{\pm 0.47}$ |
| MMD-MP (Ours) | $\mathbf{93.21}_{\pm 1.35}$ | $\mathbf{89.36}_{\pm 2.91}$ | $\mathbf{79.68}_{\pm 2.42}$ | $\mathbf{89.63}_{\pm 1.94}$ | $\mathbf{91.96}_{\pm 0.62}$ |

Table 2: Test power/100 on HC3 given $1,000$ processed paragraphs in training data.

| Method | ChatGPT | GPT3-S | Neo-S | ChatGPT
Neo-S | ChatGPT
GPT3-S | ChatGPT
GPT2-S
GPT2-M | ChatGPT
Neo-S
GPT3-S | ChatGPT
Neo-S
Neo-L |
|---|---|---|---|---|---|---|---|---|
| C2ST-S | $60.32_{\pm 2.56}$ | $38.06_{\pm 4.49}$ | $27.65_{\pm 2.34}$ | $34.48_{\pm 3.70}$ | $40.89_{\pm 3.79}$ | $24.97_{\pm 2.05}$ | $32.04_{\pm 2.41}$ | $24.53_{\pm 3.08}$ |
| C2ST-L | $87.81_{\pm 1.48}$ | $74.29_{\pm 4.16}$ | $61.05_{\pm 3.35}$ | $67.47_{\pm 3.17}$ | $75.49_{\pm 2.21}$ | $56.24_{\pm 3.53}$ | $67.10_{\pm 2.69}$ | $54.91_{\pm 3.24}$ |
| MMD-O | $27.23_{\pm 3.53}$ | $19.96_{\pm 5.03}$ | $19.58_{\pm 2.02}$ | $27.34_{\pm 1.42}$ | $26.03_{\pm 1.63}$ | $20.05_{\pm 2.86}$ | $23.91_{\pm 0.92}$ | $20.92_{\pm 1.10}$ |
| MMD-D | $91.38_{\pm 2.09}$ | $84.01_{\pm 5.04}$ | $72.81_{\pm 3.23}$ | $74.22_{\pm 4.06}$ | $83.29_{\pm 3.05}$ | $62.34_{\pm 4.00}$ | $77.76_{\pm 2.93}$ | $63.15_{\pm 2.38}$ |
| MMD-MP (Ours) | $\mathbf{92.31}_{\pm 2.30}$ | $\mathbf{86.34}_{\pm 5.37}$ | $\mathbf{76.35}_{\pm 3.51}$ | $\mathbf{85.30}_{\pm 1.99}$ | $\mathbf{89.05}_{\pm 1.64}$ | $\mathbf{79.92}_{\pm 3.88}$ | $\mathbf{85.54}_{\pm 1.93}$ | $\mathbf{79.69}_{\pm 0.78}$ |

stability of discrepancy estimation. 2) **Enhanced transferability**: Our MMD-MP prioritizes fitting HWTs $S^{tr}_{\mathbb{P}}$ compared to MMD-D when training the deep kernel, reducing its reliance on MGTs. This manner enhances the transferability in detecting unknown MGTs (as verified in Section 4.4).

## 4 Experiments

**Datasets and LLM architectures.** We evaluate our method on Human ChatGPT Comparison Corpu (HC3) (Guo et al., 2023), which is a ChatGPT text detection dataset with both long and short-level corpus, and XSum dataset (Güera & Delp, 2018), which is a news dataset. We choose paragraphs with more than 5 sentences for testing in paragraph-based detection and sentences with more than 5 words for testing in sentence-based detection. For LLMs, we consider ChatGPT (OpenAI, 2022), GPT2 series (Radford et al., 2019), GPT3-S (Toan, 2023), GPT-Neo series (Black et al., 2021), GPT4all-j (Anand et al., 2023). For each experiment, except for ChatGPT using MGTs in the original HC3, for other LLMs, we generate MGTs with the first 20 prompts of HWT in HC3.

**Two-sample test baselines.** 1) **MMD-O**: MMD with a Gaussian kernel whose bandwidth is optimized; 2) **MMD-D**: MMD with a trained deep kernel (Liu et al., 2020); 3) Classifier two-sample tests: **C2ST-S** (Lopez-Paz & Oquab, 2017) and **C2ST-L** (Cheng & Cloninger, 2022).

**Single-instance detection baselines.** 1) **Metric-based detectors**: Log-Likelihood (Solaiman et al., 2019), Entropy, Rank (Gehrmann et al., 2019), Log-Rank and DetectGPT (Mitchell et al., 2023); 2) **Model-based detectors**: OpenAI-D (Solaiman et al., 2019) and ChatGPT-D (Guo et al., 2023). We also use cross-entropy loss to optimize a deep neural network as a baseline, called CE-Classifier, whose model is the same as that of MMD-D and MMD-MP except for an additional binary classifier.

**Evaluation metrics.** We evaluate the detection performance using **test power** for two-sample test (Gretton et al., 2012) and the area under the receiver operating characteristic curve (**AUROC**) (Jiménez-Valverde, 2012) for single-instance detection. Through all experiments, we randomly take 100 paragraphs or $1,000$ sentences for testing and repeat the experiments 10 times for synthetic data or 5 times for real-world data. We use **bold** numbers to indicate the best results in tables.

### 4.1 Results on Synthetic Data

We investigate the impact of variation (*i.e.*, variance) of training data on test power. To this end, we synthesize a four-center Gaussian mixture data. Specifically, let $\mathbf{1}^d$ and $\mathbf{I}^d$ represent a $d$-dimensional all-one vector and a $d$-dimensional identity matrix, we denote $\mathbb{P}{=}\mathcal{N}(\mathbf{0}^d, \mathbf{I}^d)$ and $\mathbb{Q}(\mu, \delta)$ as:

$$\mathbb{Q}(\mu,\delta){=}\frac{1}{4}\mathcal{N}\left(\mu\begin{bmatrix}\mathbf{1}^{\frac{d}{2}}\\ \mathbf{1}^{\frac{d}{2}}\end{bmatrix}, \delta\,\mathbf{I}^d\right)+\frac{1}{4}\mathcal{N}\left(\mu\begin{bmatrix}-\mathbf{1}^{\frac{d}{2}}\\ \mathbf{1}^{\frac{d}{2}}\end{bmatrix}, \delta\,\mathbf{I}^d\right)+\frac{1}{4}\mathcal{N}\left(\mu\begin{bmatrix}\mathbf{1}^{\frac{d}{2}}\\ -\mathbf{1}^{\frac{d}{2}}\end{bmatrix}, \delta\,\mathbf{I}^d\right)+\frac{1}{4}\mathcal{N}\left(\mu\begin{bmatrix}-\mathbf{1}^{\frac{d}{2}}\\ -\mathbf{1}^{\frac{d}{2}}\end{bmatrix}, \delta\,\mathbf{I}^d\right).$$

We consider various $\mathbb{Q}$ by setting $\mu{\in}\{0.2{+}0.02{\times}i\}_{i=1}^{10}$ and $\delta{=}1.3$ with $d{=}100$. Note that we use these four-center Gaussian mixture data for training the kernel but only sample a center Gaussian data for testing. We use $L_2$-norm of the variance of data in $\mathbb{Q}$ to represent its variance.

From Figure 3, we draw two observations: 1) As $\mu$ increases, the test powers of MMD-D and our MMD-MP become higher since the distributional discrepancy between $\mathbb{P}$ and each single-center

Table 3: AUROC/100 on HC3 given 3,100 processed paragraphs.

| Method | ChatGPT | GPT3-S | Neo-S | ChatGPT Neo-S | ChatGPT GPT3-S |
|---|---|---|---|---|---|
| Likelihood | $89.82_{\pm 0.03}$ | $60.56_{\pm 1.32}$ | $61.18_{\pm 1.25}$ | $75.81_{\pm 0.51}$ | $75.05_{\pm 0.25}$ |
| Rank | $73.20_{\pm 1.49}$ | $71.96_{\pm 1.01}$ | $72.09_{\pm 0.51}$ | $72.74_{\pm 0.74}$ | $72.34_{\pm 1.38}$ |
| Log-Rank | $89.58_{\pm 0.07}$ | $63.78_{\pm 1.29}$ | $64.92_{\pm 1.04}$ | $77.57_{\pm 0.55}$ | $76.47_{\pm 0.12}$ |
| Entropy | $31.53_{\pm 0.90}$ | $54.34_{\pm 1.33}$ | $56.19_{\pm 0.33}$ | $44.08_{\pm 0.24}$ | $42.08_{\pm 2.01}$ |
| DetectGPT-d | $77.92_{\pm 0.74}$ | $53.41_{\pm 0.41}$ | $52.07_{\pm 0.38}$ | $66.01_{\pm 0.29}$ | $65.70_{\pm 1.14}$ |
| DetectGPT-z | $81.07_{\pm 0.77}$ | $53.45_{\pm 0.53}$ | $52.28_{\pm 0.31}$ | $67.54_{\pm 0.19}$ | $67.32_{\pm 1.02}$ |
| OpenAI-D | $78.57_{\pm 1.55}$ | $84.05_{\pm 0.71}$ | $84.86_{\pm 0.87}$ | $81.20_{\pm 0.95}$ | $80.68_{\pm 1.64}$ |
| ChatGPT-D | $95.64_{\pm 0.13}$ | $61.89_{\pm 1.04}$ | $54.45_{\pm 0.10}$ | $75.47_{\pm 0.63}$ | $78.95_{\pm 1.00}$ |
| CE-Classifier | $96.19_{\pm 0.17}$ | $92.44_{\pm 0.63}$ | $88.88_{\pm 0.19}$ | $90.93_{\pm 0.72}$ | $92.97_{\pm 0.28}$ |
| MMD-O | $56.34_{\pm 0.66}$ | $59.90_{\pm 0.87}$ | $63.19_{\pm 0.76}$ | $60.46_{\pm 1.28}$ | $57.79_{\pm 1.25}$ |
| MMD-D | $95.83_{\pm 0.37}$ | $94.86_{\pm 0.48}$ | $91.12_{\pm 0.38}$ | $91.39_{\pm 0.86}$ | $93.49_{\pm 0.46}$ |
| MMD-MP (Ours) | $\mathbf{96.20}_{\pm 0.28}$ | $\mathbf{95.08}_{\pm 0.32}$ | $\mathbf{92.04}_{\pm 0.58}$ | $\mathbf{92.48}_{\pm 0.37}$ | $\mathbf{94.61}_{\pm 0.22}$ |

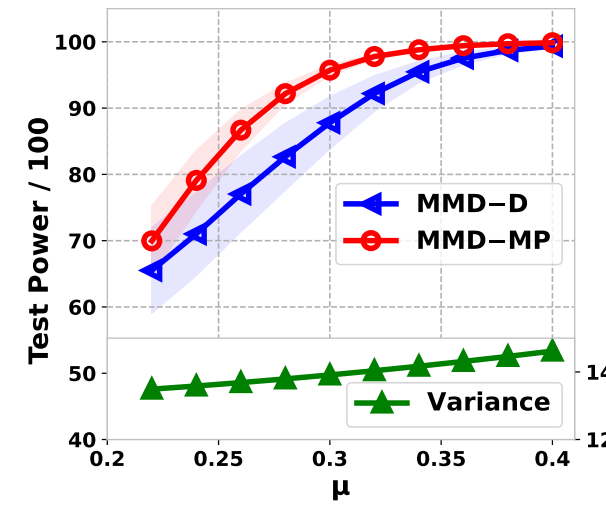

Figure 3: Impact of variance in training data on test power.

Table 4: AUROC/100 on HC3 given 1,000 processed paragraphs in training data.

| Method | ChatGPT | GPT3-S | Neo-S | ChatGPT Neo-S | ChatGPT GPT3-S | ChatGPT GPT2-S GPT2-M | ChatGPT Neo-S GPT3-S | ChatGPT Neo-S Neo-L |
|---|---|---|---|---|---|---|---|---|
| CE-Classifier | $\mathbf{95.99}_{\pm 0.40}$ | $91.40_{\pm 0.37}$ | $87.27_{\pm 0.52}$ | $88.13_{\pm 0.44}$ | $91.59_{\pm 0.36}$ | $84.89_{\pm 0.61}$ | $88.91_{\pm 0.51}$ | $84.15_{\pm 1.30}$ |
| MMD-O | $54.64_{\pm 1.69}$ | $61.52_{\pm 1.18}$ | $61.93_{\pm 2.22}$ | $58.28_{\pm 1.65}$ | $57.92_{\pm 1.32}$ | $57.86_{\pm 1.39}$ | $59.78_{\pm 0.61}$ | $58.07_{\pm 1.20}$ |
| MMD-D | $93.86_{\pm 0.70}$ | $91.50_{\pm 1.24}$ | $81.10_{\pm 0.83}$ | $89.28_{\pm 0.91}$ | $90.28_{\pm 1.59}$ | $85.50_{\pm 0.85}$ | $88.07_{\pm 0.87}$ | $84.20_{\pm 2.33}$ |
| MMD-MP (Ours) | $95.95_{\pm 0.42}$ | $\mathbf{94.28}_{\pm 0.57}$ | $\mathbf{89.61}_{\pm 0.44}$ | $\mathbf{90.83}_{\pm 0.79}$ | $\mathbf{93.46}_{\pm 0.52}$ | $\mathbf{87.03}_{\pm 0.59}$ | $\mathbf{91.25}_{\pm 0.56}$ | $\mathbf{86.93}_{\pm 0.52}$ |

Gaussian data in $\mathbb{Q}$ becomes larger; 2) When the variance of data in $\mathbb{Q}$ increases with $\mu$, the test power of MMD-MP surpasses that of MMD-D by a maximum margin of approximately $9\%$ power. This suggests that forcing the aggregation of all data in $\mathbb{Q}$ will hinder MGT detection performance when the variance of training data is significant.

### 4.2 Test Power on Paragraph-based Detection

We compare our MMD-MP with the state-of-the-art (SOTA) two-sample test method for detecting MGTs on HC3 in terms of test power and defer the results on XSum in Appendix J.1. To broadly evaluate detection performance, we conduct experiments on various scenarios, including training on full training data, a limited number of training data, and unbalanced training data.

**Test power on full training data.** We conduct this experiment using $3,100$ processed paragraphs to train the model. Table 1 shows the detection performance under different training populations in terms of test power compared with other baselines, including one and two populations. The results show that MMD-MP exhibits superior test power compared with other methods, particularly in detecting Neo-S texts, where the test power is approximately $6\% \uparrow$ higher than MMD-D.

**Test power on a limited number of training data.** We utilize $1,000$ processed paragraphs to train the models with one, two, and three training populations, respectively. Table 2 demonstrates that our method achieves significantly higher test power performance compared with others. Remarkably, our method outperforms MMD-D by an average of $8.20\% \uparrow$ on test power over the two training populations and exhibits a $13.97\% \uparrow$ increase over the three training populations, suggesting that extreme instability of discrepancy estimation of MMD-D when dealing with multiple populations.

**Test power on challenging unbalanced training data.** In real-world scenarios, obtaining HWTs is easily feasible, while collecting MGTs poses more challenges. To thoroughly assess the performance of our approach, we conduct an evaluation with $2,000$ processed HWT and $400$ MGT training paragraphs. As illustrated in the top of Figure 4, our approach exhibits significantly superior performance compared with other methods, *e.g.*, surpassing the test power of $6.96\%{\sim}14.40\% \uparrow$ than MMD-D, highlighting its stability in detecting MGTs under unbalanced training data scenarios.

### 4.3 AUROC on Sentence-based Detection

In this section, we compare our MMD-MP with the SOTA single-instance detection method for detecting MGTs on HC3 in terms of AUROC and defer the results on XSum in Appendix J.2.

**AUROC on full training data.** Table 3 shows that our MMD-MP achieves better AUROC than other baselines. Notably, our MMD-MP outperforms the SOTA model-based method, *i.e.*, ChatGPT-D with $1.20\% \uparrow$ of AUROC when detecting ChatGPT texts. Moreover, MMD-MP achieves an improvement of $0.22\%{\sim}1.71\% \uparrow$ on AUROC over MMD-D and $0.61\%{\sim}2.64\% \uparrow$ over the CE-Classifier method, illustrating the superiority of our method in detecting the single sentence.

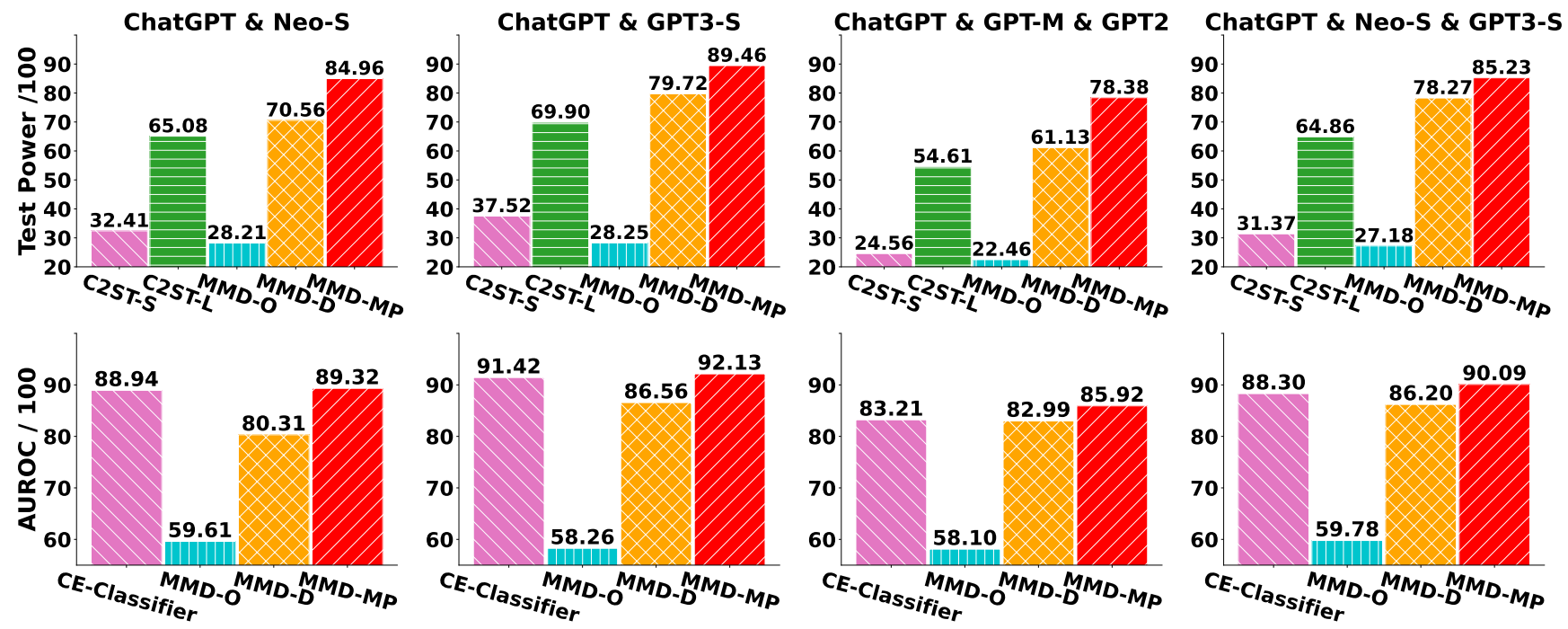

Figure 4: Test power and AUROC on HC3 given 2,000 HWT and 400 MGT training paragraphs.

**AUROC on a limited number of training data.** From Table 4, our MMD-MP achieves performance comparable to CE-classifier in detecting ChatGPT texts and surpasses other baselines in other scenarios. Particularly, our method outperforms MMD-D by 2.46%↑ and CE-Classifier by 1.39%↑ on average. Note that although the model of CE-Classifier is the same as MMD-D and MMD-MP except for an additional classifier, our MMD-MP demonstrates superior detection performance over CE-Classifier, indicating the powerful distinguishability of our method.

**AUROC on challenging unbalanced training data.** From the bottom of Figure 4, our approach consistently outperforms baselines for sentence-based detection. Critically, our MMD-MP achieves an improvement of 3.89%∼9.01% ↑ on AUROC over MMD-D and 0.38%∼1.79% ↑ over the CE-Classifier method. Combining Tables 3, 4 and Figure 4, we conclude that our method is superior in detecting a single sentence under various scenarios on different LLMs compared with other methods in total, suggesting the stability of our method for MGT detection.

### 4.4 Results on Unknown LLM texts

In light of poor performance for MGT detection baselines on unknown LLM-text, we evaluate our method in the context of this type of detection. We train the models using texts generated by ChatGPT, GPT2 and GPT2-m on HC3, and then test on texts generated by GPT-Neo-L, GPT-j-6b and GPT4all-j. From Tables 5-6, our method outperforms the baselines by a large margin on test power and AUROC. Critically, our MMD-MP achieves an absolute improvement of 23.61%∼27.65%↑ on test power over MMD-D. Moreover, our method outperforms MMD-D by 3.25% ↑ and CE-Classifier by 3.37% ↑ of AUROC on average. These results demonstrate the superior transferability of our method.

Table 5: Test Power/100 on unknown LLMs.

| Method | Neo-L | GPT-j-6b | GPT4all-j |
|---|---|---|---|
| C2ST-S | $11.20_{\pm 3.39}$ | $7.72_{\pm 0.72}$ | $14.30_{\pm 2.38}$ |
| C2ST-L | $34.12_{\pm 6.09}$ | $29.64_{\pm 3.64}$ | $42.37_{\pm 3.18}$ |
| MMD-O | $12.93_{\pm 1.51}$ | $7.71_{\pm 2.66}$ | $17.08_{\pm 1.14}$ |
| MMD-D | $38.18_{\pm 4.13}$ | $31.92_{\pm 4.93}$ | $51.28_{\pm 0.11}$ |
| MMD-MP (Ours) | $\mathbf{61.79}_{\pm 3.54}$ | $\mathbf{59.57}_{\pm 4.33}$ | $\mathbf{77.69}_{\pm 0.46}$ |

Table 6: AUROC/100 on unknown LLMs.

| Method | Neo-L | GPT-j-6b | GPT4all-j |
|---|---|---|---|
| CE-Classifier | $78.00_{\pm 1.69}$ | $74.56_{\pm 1.49}$ | $82.57_{\pm 0.91}$ |
| MMD-O | $54.86_{\pm 0.31}$ | $53.85_{\pm 0.86}$ | $52.92_{\pm 1.33}$ |
| MMD-D | $77.91_{\pm 0.87}$ | $75.47_{\pm 1.41}$ | $82.11_{\pm 0.51}$ |
| MMD-MP (Ours) | $\mathbf{81.08}_{\pm 0.71}$ | $\mathbf{78.41}_{\pm 0.98}$ | $\mathbf{85.75}_{\pm 0.30}$ |

### 4.5 Visualization of Kernel Feature $\phi_f$

We visualize the feature ($d$=300) extracted by $\phi_f$ over two LLM-texts via t-SNE (Van der Maaten & Hinton, 2008) for MMD-D and MMD-MP. In Figure 5, two types of LLM-text features by MMD-D are entangled and disorganized, while the MGT features obtained by our MMD-MP exhibit lower overlap, confirming that our method indeed relaxes the constraint of aggregating all MGT instances.

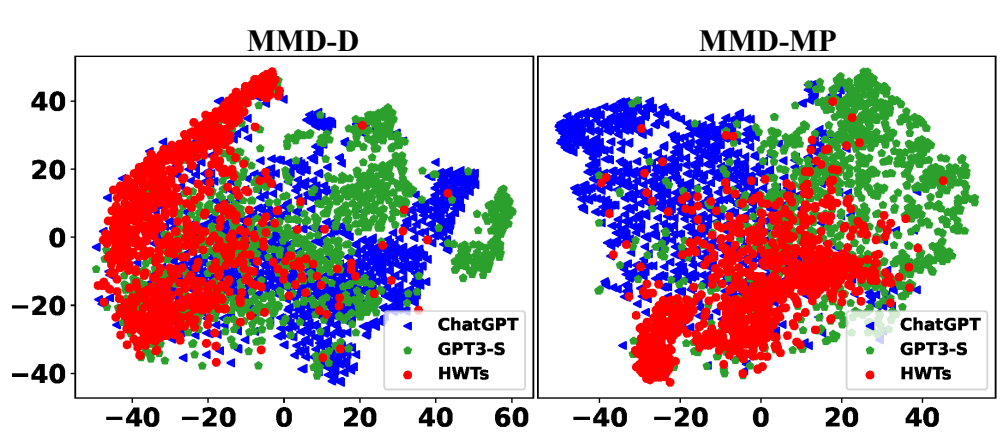

Figure 5: Features Visualization via t-SNE.

## 5 Conclusion

In this paper, we propose a multi-population aware optimization method for training kernel-based MMD called MMD-MP, which alleviates the poor optimization of MMD. With a trained deep kernel, we design two MGT detection approaches for paragraph-based and sentence-based detection tasks, respectively. Extensive experiments on a variety of LLMs demonstrate the superiority of our methods in terms of test power and AUROC, especially in detecting unknown LLM texts.

## Acknowledgments

We would like to thank Feng Liu for insightful technical discussions. This work was partially supported by the National Natural Science Foundation of China (NSFC) 62072190, TCL science and technology innovation fund, the Young Scholar Project of Pazhou Lab (No. PZL2021KF0021), the NSFC General Program No. 62376235, Guangdong Basic and Applied Basic Research Foundation No. 2022A1515011652, and HKBU Faculty Niche Research Areas No. RC-FNRA-IG/22-23/SCI/04, Tencent Innovation Joint Project.

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

# APPENDIX

## CONTENTS

## A THEORETICAL ANALYSIS

Given a kernel $k_\omega$, we define MPP and its unbiased estimator as:

$$\mathrm{MPP}(\mathbb{P}, \mathbb{Q}; \mathcal{H}_k) := \mathbb{E}\left[k_\omega(X, X') - 2k_\omega(X, Y)\right]. \tag{12}$$

$$\widehat{\mathrm{MPP}}_u(S_\mathbb{P}, S_\mathbb{Q}; k_\omega) = \frac{1}{n(n-1)} \sum_{i \neq j} H^*_{ij}, \text{ where } H^*_{ij} := k_\omega(\mathbf{x}_i, \mathbf{x}_j) - k_\omega(\mathbf{x}_i, \mathbf{y}_j) - k_\omega(\mathbf{y}_i, \mathbf{x}_j). \tag{13}$$

For simplicity, we denote $\hat{\eta}_\omega = \widehat{\mathrm{MPP}}_u = \frac{1}{n(n-1)} \sum_{i \neq j} H^*_{ij}$ and $\eta_\omega = \mathbb{E}[H^*_{12}]$.

### A.1 PROOF OF PROPOSITION 1

**Proposition 1.** *(Asymptotics of $\widehat{\mathrm{MPP}}_u$). Under the alternative $\mathfrak{H}_1 : \mathbb{P} \neq \mathbb{Q}$, based on a standard central limit theorem, we have:*

$$\sqrt{n}(\widehat{\mathrm{MPP}}_u - \mathrm{MPP}) \xrightarrow{d} \mathcal{N}(0, \sigma^2_{\mathfrak{H}^*_1}), \tag{14}$$

*where $\sigma^2_{\mathfrak{H}^*_1} := 4\left(\mathbb{E}[H^*_{12} H^*_{13}] - \mathbb{E}[H^*_{12}]^2\right)$, $H^*_{12}$, $H^*_{13}$ denote different $H^*_{ij}$.*

*Proof.* Let $U$ denote a pair $(\mathbf{x}, \mathbf{y})$ and $h(U, U') = k_\omega(\mathbf{x}_i, \mathbf{x}_j) - k_\omega(\mathbf{x}_i, \mathbf{y}_j) - k_\omega(\mathbf{y}_i, \mathbf{x}_j)$. Based on the property of $U$-statistic and Lemma A in Section 5.2.1 of Serfling (2009), when $n \to \infty$, we have

$$n\mathrm{Var}[\hat{\eta}_\omega] \to 4\xi_\omega =: \sigma^2_\omega, \tag{15}$$

where we use $\sigma^2_\omega$ to denote $\sigma^2_{\mathfrak{H}^*_1}$ for simplicity, and $\xi_\omega$ is represented as:

$$\begin{aligned}
\xi_\omega &= \mathrm{Var}_U\left[\mathbb{E}_{U'}\left[h(U, U')\right]\right] \\
&= \mathbb{E}_U\left[\mathbb{E}_{U'}\left[h(U, U')\right]\mathbb{E}_{U''}\left[h(U, U'')\right]\right] - \mathbb{E}_U\left[\mathbb{E}_{U'}[h(U, U')]\right]^2 \\
&= \mathbb{E}[H^*_{12} H^*_{13}] - \mathbb{E}[H^*_{12}]^2.
\end{aligned}$$

Via Theorem A in Section 5.5.1 of Serfling (2009), we obtain the conclusion. □

### A.2 PROOF OF COROLLARY 1

**Corollary 1.** *(Test power of $\widehat{\mathrm{MPP}}_u$.) For reasonably large $n$, the probability of rejecting the null hypothesis $\mathfrak{H}_0 : \mathbb{P} = \mathbb{Q}$ when $\mathbb{P} \neq \mathbb{Q}$ is given by:*

$$\mathrm{Pr}^{\mathrm{MPP}}_{\mathfrak{H}^*_1, r} \to \Phi\Big(\frac{\sqrt{n}(\mathrm{MPP} + R(S_\mathbb{Q}))}{\sigma_{\mathfrak{H}^*_1}} - \frac{r}{\sqrt{n}\,\sigma_{\mathfrak{H}^*_1}}\Big), \tag{16}$$

*where $\mathrm{Pr}^{\mathrm{MPP}}_{\mathfrak{H}^*_1, r} := \Pr\left(n\left[\widehat{\mathrm{MPP}}_u + R(S_\mathbb{Q})\right] > r\right)$ and $R(S_\mathbb{Q}) = \frac{1}{n(n-1)} \sum_{i \neq j} k_\omega(\mathbf{y}_i, \mathbf{y}_j) > 0$, $\Phi$ is the standard normal cumulative distribution function.*

*Proof.* Via Proposition 1, we obtain

$$\begin{aligned}
\Pr\left(n\left[\widehat{\mathrm{MPP}}_u + R(S_\mathbb{Q})\right] > r\right) &= \Pr\left(\frac{\sqrt{n}(\widehat{\mathrm{MPP}}_u - \mathrm{MPP})}{\sigma_{\mathfrak{H}^*_1}} > \frac{r - n[\mathrm{MPP} + R(S_\mathbb{Q})]}{\sqrt{n}\,\sigma_{\mathfrak{H}^*_1}}\right) \\
&= 1 - \Phi\Big(\frac{r - n[\mathrm{MPP} + R(S_\mathbb{Q})]}{\sqrt{n}\,\sigma_{\mathfrak{H}^*_1}}\Big) \\
&= \Phi\Big(\frac{\sqrt{n}(\mathrm{MPP} + R(S_\mathbb{Q}))}{\sigma_{\mathfrak{H}^*_1}} - \frac{r}{\sqrt{n}\,\sigma_{\mathfrak{H}^*_1}}\Big).
\end{aligned}$$

□

### A.3 Proof of Theorem 1

To analyze the convergence of our method, we first describe the following relevant technical assumptions that have been analyzed by Liu et al. (2020).

- **(A)** The kernels $k_\omega$ are uniformly bounded by $\sup_{\omega\in\Omega}\sup_{x\in\mathcal{X}} k_\omega(x,x) \le \nu$.
- **(B)** The possible kernel parameters $\omega$ lie in a Banach space of dimension $D$. Furthermore, the set of possible kernel parameters $\Omega$ is bounded by $R_\Omega$, *i.e.*, $\Omega \subseteq \{\omega \mid \|\omega\| \le R_\Omega\}$.
- **(C)** The kernel parameters is $L_k$-Lipschitz for all data $\mathbf{x}, \mathbf{x}' \in \mathcal{X}$ and $\omega, \omega' \in \Omega$, *i.e.*,

$$|k_\omega(\mathbf{x},\mathbf{x}') - k_{\omega'}(\mathbf{x},\mathbf{x}')| \le L_k\|\omega - \omega'\|.$$

We first provide some results on the uniform convergence of $\hat{\eta}_\omega$ and $\hat{\sigma}^2_\omega$ based on $\epsilon$-net argument (Vershynin, 2018), which will be used to prove Theorem 1.

**Proposition 2.** *Under Assumptions (A) to (C), we have that with probability at least $1-\delta$,*

$$\sup_\omega |\hat{\eta}_\omega - \eta_\omega| \le \frac{6}{\sqrt{n}}\left[\nu\sqrt{2\log\frac{2}{\delta} + 2D\log\left(4R_\Omega\sqrt{n}\right)} + L_k\right].$$

*Proof.* We denote a random error function

$$\Phi(\omega) := \hat{\eta}_\omega - \eta_\omega.$$

Based on Assumption (B), we can use at most $T = (4R_\Omega/q)^D$ points $\{\omega_i\}_{i=1}^T$ such that for any $\omega \in \Omega$, $\min_i \|\omega - \omega_i\| \le q$ (Cucker & Smale (2002), Proposition 5).

Recall that $\hat{\eta}_\omega = \frac{1}{n(n-1)}\sum_{i\neq j} H^*_{ij}$, we construct a new population by replacing one data pair $(\mathbf{x}_1, \mathbf{y}_1)$ in $S_\mathbb{P}$ and $S_\mathbb{Q}$ with $(\mathbf{x}'_1, \mathbf{y}'_1)$, and thus obtain $\hat{\eta}'_\omega = \frac{1}{n(n-1)}\sum_{i\neq j} F^*_{ij}$, where $F$ is the same as $H$ except when $i$ or $j$ equals to 1.

We calculate the difference

$$\begin{aligned}|\hat{\eta}_\omega - \hat{\eta}'_\omega| &\le \frac{1}{n(n-1)}\sum_{i\neq j}|H^*_{ij} - F^*_{ij}| = \frac{1}{n(n-1)}\sum_{i>1}|H^*_{i1} - F^*_{i1}| + \frac{1}{n(n-1)}\sum_{j>1}|H^*_{1j} - F^*_{1j}| \\ &\le \frac{2}{n(n-1)}\sum_{i>1} 6\nu = \frac{12\nu}{n}.\end{aligned}$$

According to a union bound and McDiarmid's inequality (Mohri et al., 2018), we have

$$\begin{aligned}\Pr\left(\max_{i\in\{1,\dots,T\}}|\Phi(\omega_i)| \ge \epsilon\right) &\le \sum_{i=1}^T \Pr\left(|\Phi(\omega_i)| \ge \epsilon\right) \\ &= \sum_{i=1}^T \Pr\left(|\hat{\eta}_{\omega_i} - \eta_{\omega_i}| \ge \epsilon\right) = \sum_{i=1}^T \Pr\left(|\hat{\eta}_{\omega_i} - \mathbb{E}[\eta'_{\omega_i}]| \ge \epsilon\right) \\ &\le 2T\exp\left(-\frac{2\epsilon^2}{\sum_{j=1}^n (12\nu/n)^2}\right) = 2T\exp\left(-\frac{2\epsilon^2}{(12\nu)^2/n}\right).\end{aligned}$$

Let $2T\exp\left(-\frac{2\epsilon^2}{(12\nu)^2/n}\right) = \delta$, we obtain $\epsilon = \frac{12\nu}{\sqrt{2n}}\sqrt{\log\frac{2T}{\delta}}$. Thus, with probability at least $1-\delta$, we have

$$\max_{i\in\{1,\dots,T\}}|\Phi(\omega_i)| \le \frac{12\nu}{\sqrt{2n}}\sqrt{\log\frac{2T}{\delta}} \le \frac{6\nu}{\sqrt{n}}\sqrt{2\log\frac{2}{\delta} + 2D\log\frac{4R_\Omega}{q}}. \tag{17}$$

We next deviate the Lipschitz of $\Phi(\omega)$ based on Assumption (C).

$$|\hat{\eta}_\omega - \hat{\eta}_{\omega'}| \le \frac{1}{n(n-1)}\sum_{i\neq j}|H^{*(\omega)}_{ij} - H^{*(\omega')}_{ij}| \le \frac{1}{n(n-1)}\sum_{i\neq j} 3L_k\|\omega - \omega'\| = 3L_k\|\omega - \omega'\|,$$

$$|\eta_\omega - \eta_{\omega'}| = \left|\mathbb{E}\left[H_{12}^{*(\omega)}\right] - \mathbb{E}\left[H_{12}^{*(\omega')}\right]\right| \le \mathbb{E}\left|H_{12}^{*(\omega)} - H_{12}^{*(\omega')}\right| \le 3L_k\|\omega - \omega'\|.$$

Thus, we have

$$|\Phi(\omega) - \Phi(\omega')| = |\hat{\eta}_\omega - \eta_\omega - (\hat{\eta}_{\omega'} - \eta_{\omega'})| \le |(\hat{\eta}_\omega - \hat{\eta}_{\omega'}) + (\eta_{\omega\prime} - \eta_\omega)| \le 6L_k\|\omega - \omega'\|. \quad (18)$$

Combining the results of (17) and (18) and setting $q = \frac{1}{\sqrt{n}}$, we have that with probability at least $1-\delta$

$$\begin{aligned}
\sup_\omega |\Phi(\omega)| = |\Phi(\omega^*)| &= |\Phi(\omega^*) - \Phi(\omega_*) + \Phi(\omega_*)| \le |\Phi(\omega_*)| + |\Phi(\omega^*) - \Phi(\omega_*)| \\
&\le \max_{i\in\{1\ldots T\}} \|\Phi(\omega_i)\| + 6L_k q \\
&\le \frac{6\nu}{\sqrt{n}}\sqrt{2\log\frac{2}{\delta} + 2D\log\frac{4R_\Omega}{q}} + \frac{6L_k}{\sqrt{n}} \\
&= \frac{6}{\sqrt{n}}\left[\nu\sqrt{2\log\frac{2}{\delta} + 2D\log\left(4R_\Omega\sqrt{n}\right)} + L_k\right],
\end{aligned}$$

where $\omega^* = \arg_\omega \sup|\Phi(\omega)|$ and $\omega_* = \arg_\omega \min_{\omega_i \in\{\omega_1\ldots\omega_T\}} \|\omega^* - \omega_i\|$. □

Next, we present some lemmas to lay the foundation for establishing the uniform convergence of $\hat{\sigma}^2_\omega$. For simplicity, we denote $\hat{\sigma}_k = \hat{\sigma}_{\mathfrak{H}_1^*}$ and $\sigma_k = \sigma_{\mathfrak{H}_1^*}$.

**Lemma 1.** *Under Assumption (A), we have that with probability at least* $1-\delta$*.*

$$|\hat{\sigma}^2_\omega - \mathbb{E}\,\hat{\sigma}^2_\omega| \le 252\nu^2\sqrt{\frac{2}{n}\log\frac{2}{\delta}}.$$

*Proof.* We first estimate $|\hat{\sigma}^2_\omega - (\hat{\sigma}'_\omega)^2|$ when changing one data pair in $S_\mathbb{P}$ and $S_\mathbb{Q}$. To this end, we construct a new population by replacing one data pair $(\mathbf{x}_1, \mathbf{y}_1)$ in $S_\mathbb{P}$ and $S_\mathbb{Q}$ with $(\mathbf{x}'_1, \mathbf{y}'_1)$, and thus obtain $(\hat{\sigma}'_\omega)^2 = 4\left(\frac{1}{n^3}\sum_i\left(\sum_j F^*_{ij}\right)^2 - \left(\frac{1}{n^2}\sum_{ij} F^*_{ij}\right)^2\right)$, where $F$ is the same as $H$ except when $i$ or $j$ equals to 1. Recall that

$$\hat{\sigma}^2_\omega = 4\left(\frac{1}{n^3}\sum_i\left(\sum_j H^*_{ij}\right)^2 - \left(\frac{1}{n^2}\sum_{ij} H^*_{ij}\right)^2\right).$$

After changing one data pair, the difference of the first term changes with

$$\begin{aligned}
&\left|\frac{1}{n^3}\sum_i\left(\sum_j H^*_{ij}\right)^2 - \frac{1}{n^3}\sum_i\left(\sum_j F^*_{ij}\right)^2\right| \le \frac{1}{n^3}\sum_i\left|\left(\sum_j H^*_{ij}\right)^2 - \left(\sum_j F^*_{ij}\right)^2\right| \\
&= \frac{1}{n^3}\sum_i\left|\sum_{j\ell} H^*_{ij}H^*_{i\ell} - \sum_{j\ell} F^*_{ij}F^*_{i\ell}\right| \le \frac{1}{n^3}\sum_{ij\ell}\left|H^*_{ij}H^*_{i\ell} - F^*_{ij}F^*_{i\ell}\right| \\
&= \frac{1}{n^3}\sum_{i=1}\sum_{j\ell}\left|H^*_{ij}H^*_{i\ell} - F^*_{ij}F^*_{i\ell}\right| + \frac{1}{n^3}\sum_{i>1}\sum_{j=1,\ell=1}\left|H^*_{ij}H^*_{i\ell} - F^*_{ij}F^*_{i\ell}\right| \\
&\quad \frac{1}{n^3}\sum_{i>1}\sum_{j=1,\ell\ne1}\left|H^*_{ij}H^*_{i\ell} - F^*_{ij}F^*_{i\ell}\right| + \frac{1}{n^3}\sum_{i>1}\sum_{j\ne1,\ell=1}\left|H^*_{ij}H^*_{i\ell} - F^*_{ij}F^*_{i\ell}\right| \\
&\le \frac{1}{n^3}(n^2 18\nu^2 + (n-1)9\nu^2 + 2(n-1)^2 18\nu^2) \\
&= \left(\frac{6}{n} - \frac{7}{n^2} + \frac{3}{n^3}\right)9\nu^2,
\end{aligned}$$

where the penultimate line follows by the facts that: 1) $\left|H^*_{1j}H^*_{1\ell} - F^*_{1j}F^*_{1\ell}\right| \leq 18\nu^2$ due to $|H_{ij}| \leq 3\nu$, $|F_{ij}| \leq 3\nu$; 2) $|H^*_{i1}H^*_{i1} - F^*_{i1}F^*_{i1}| \leq \max\{H^*_{i1}H^*_{i1}, F^*_{i1}F^*_{i1}\} \leq 9\nu^2$; 3) when $i > 1$, $\ell \neq 1$, $|H^*_{i1}H^*_{i\ell} - F^*_{i1}F^*_{i\ell}| = |H^*_{i1}H^*_{i\ell} - F^*_{i1}H^*_{i\ell}| \leq |H^*_{i\ell}|\,|H^*_{i1} - F^*_{i1}| \leq (6\nu)(3\nu) = 18\nu$.

After changing one data pair, the difference of the second term changes with

$$\begin{aligned}
&\left|\left(\frac{1}{n^2}\sum_{ij} H^*_{ij}\right)^2 - \left(\frac{1}{n^2}\sum_{ij} F^*_{ij}\right)^2\right| \\
&= \frac{1}{n^4}\left|\sum_{ij} H^*_{ij} + \sum_{ij} F_{ij}\right|\left|\sum_{ij} H^*_{ij} - \sum_{ij} F^*_{ij}\right| \\
&\leq \frac{1}{n^2}(2\cdot 3\nu)\cdot \sum_{ij}\left|H^*_{ij} - F^*_{ij}\right| \\
&= \frac{1}{n^2}(2\cdot 3\nu)\cdot \left(\sum_{i=1,j\neq 1}\left|H^*_{ij} - F^*_{ij}\right| + \sum_{j=1,i\neq 1}\left|H^*_{ij} - F^*_{ij}\right|\right) \\
&\leq \frac{1}{n^2}(2\cdot 3\nu)\cdot (2n-1)6\nu \\
&= 36\nu^2\left(\frac{2}{n} - \frac{1}{n^2}\right).
\end{aligned}$$

Thus, we have

$$\begin{aligned}
|\hat{\sigma}^2_\omega - (\hat{\sigma}'_\omega)^2| &\leq 4\left[\left(\frac{6}{n} - \frac{7}{n^2} + \frac{3}{n^3}\right)9\nu^2 + 36\nu^2\left(\frac{2}{n} - \frac{1}{n^2}\right)\right] \\
&= 36\nu^2\left[\frac{14}{n} - \frac{11}{n^2} + \frac{3}{n^3}\right] \\
&\leq \frac{504\nu^2}{n}.
\end{aligned}$$

Using McDiarmid's inequality (Mohri et al., 2018), with probability at least $1-\delta$, we have

$$|\hat{\sigma}^2_\omega - \mathbb{E}\,\hat{\sigma}^2_\omega| \leq 252\nu^2\sqrt{\frac{2}{n}\log\frac{2}{\delta}}.$$

□

Since $\hat{\sigma}^2_\omega$ is not unbiased, we next estimate $|\mathbb{E}\,\hat{\sigma}^2_\omega - \sigma^2_\omega|$.

**Lemma 2.** *Under Assumption (A), we have*

$$|\mathbb{E}\,\hat{\sigma}^2_\omega - \sigma^2_\omega| \leq \frac{648\nu^2}{n}.$$

*Proof.* Recall that $\sigma^2_\omega$ and the expectation of $\hat{\sigma}^2_\omega$

$$\begin{aligned}
\sigma^2_\omega &= 4\left(\mathbb{E}[H^*_{12}H^*_{13}] - \mathbb{E}[H^*_{12}]^2\right) \\
\mathbb{E}\,\hat{\sigma}^2_\omega &= 4\left(\frac{1}{n^3}\sum_{ij\ell}\mathbb{E}\left[H^*_{i\ell}H^*_{j\ell}\right] - \frac{1}{n^4}\sum_{ijab}\mathbb{E}\left[H^*_{ij}H^*_{ab}\right]\right).
\end{aligned}$$

We decompose the summation term into terms with non-repeated indices and terms with repeated indices.

For the first term in $\mathbb{E}\,\hat{\sigma}^2_\omega$,

$$\begin{aligned}\frac{1}{n^3}\sum_{ij\ell}\mathbb{E}[H^*_{i\ell}H^*_{j\ell}] &= \frac{1}{n^3}\sum_{ij\ell:|\{i,j,\ell\}|=3}\mathbb{E}[H^*_{i\ell}H^*_{j\ell}] + \frac{1}{n^3}\sum_{ij\ell:|\{i,j,\ell\}|<3}\mathbb{E}[H^*_{i\ell}H^*_{j\ell}]\\ &= \frac{n(n-1)(n-2)}{n^3}\mathbb{E}[H^*_{12}H^*_{13}] + \left(1-\frac{n(n-1)(n-2)}{n^3}\right)\mathbb{E}[H^*_{i\ell}H^*_{j\ell}].\end{aligned}$$

Thus, we have

$$\begin{aligned}&\left|\frac{1}{n^3}\sum_{ij\ell}\mathbb{E}[H^*_{i\ell}H^*_{j\ell}] - \mathbb{E}[H^*_{12}H^*_{13}]\right|\\ &= \left|\left(\frac{n(n-1)(n-2)}{n^3}-1\right)\mathbb{E}[H^*_{12}H^*_{13}] + \left(1-\frac{n(n-1)(n-2)}{n^3}\right)\mathbb{E}[H^*_{i\ell}H^*_{j\ell}]\right|\\ &= \left(\frac{3}{n}-\frac{2}{n^2}\right)\left|\mathbb{E}[H^*_{12}H^*_{13}] - \mathbb{E}[H^*_{i\ell}H^*_{j\ell}]\right|\\ &\le \left(\frac{3}{n}-\frac{2}{n^2}\right)18\nu^2.\end{aligned}$$

Similarly, for the second term in $\mathbb{E}\,\hat{\sigma}^2_\omega$, note that

$$\begin{aligned}\frac{1}{n^4}\sum_{ijab}\mathbb{E}[H^*_{ij}H^*_{ab}] &= \frac{1}{n^4}\sum_{ijab:|\{i,j,a,b\}|=4}\mathbb{E}[H^*_{ij}H^*_{ab}] + \frac{1}{n^4}\sum_{ijab:|\{i,j,a,b\}|<4}\mathbb{E}[H^*_{ij}H^*_{ab}]\\ &= \frac{n(n-1)(n-2)(n-3)}{n^4}\mathbb{E}[H^*_{12}]^2 + \left(1-\frac{n(n-1)(n-2)(n-3)}{n^4}\right)\mathbb{E}[H^*_{ij}H^*_{ab}].\end{aligned}$$

Thus, we have

$$\begin{aligned}&\left|\frac{1}{n^4}\sum_{ijab}\mathbb{E}[H^*_{ij}H^*_{ab}] - \mathbb{E}[H^*_{12}H^*_{12}]\right|\\ &= \left|\left(\frac{n(n-1)(n-2)(n-3)}{n^4}-1\right)\mathbb{E}[H^*_{12}]^2 + \left(1-\frac{n(n-1)(n-2)(n-3)}{n^4}\right)\mathbb{E}[H^*_{ij}H^*_{ab}]\right|\\ &= \left(\frac{6}{n}-\frac{11}{n^2}+\frac{6}{n^3}\right)\left|\mathbb{E}[H^*_{12}]^2 - \mathbb{E}[H^*_{ij}H^*_{ab}]\right|\\ &\le \left(\frac{6}{n}-\frac{11}{n^2}+\frac{6}{n^3}\right)18\nu^2.\end{aligned}$$

Therefore, we conclude that

$$\begin{aligned}|\mathbb{E}\,\hat{\sigma}^2_\omega - \sigma^2_\omega| &\le 4\left[\left(\frac{3}{n}-\frac{2}{n^2}\right)18\nu^2 + \left(\frac{6}{n}-\frac{11}{n^2}+\frac{6}{n^3}\right)18\nu^2\right]\\ &\le \left(\frac{9}{n}-\frac{13}{n^2}+\frac{6}{n^3}\right)72\nu^2\\ &\le \frac{648\nu^2}{n}.\end{aligned}$$

□

Next, we deviate the Lipschitz of $\Psi(\omega) := \hat{\sigma}^2_\omega - \sigma^2_\omega$.

**Lemma 3.** *Under Assumptions (A) and (C), the Lipschitz of $\Psi(\omega) := \hat{\sigma}^2_\omega - \sigma^2_\omega$ is given by*

$$|\Psi(\omega) - \Psi(\omega')| \le 288 L_k \nu.$$

*Proof.* We first deal with the Lipschitz of $\Psi(\omega)$ by

$$|\Psi(\omega)-\Psi(\omega')| = |\hat{\sigma}_\omega^2 - \sigma_\omega^2 - \hat{\sigma}_{\omega'}^2 + \sigma_{\omega'}^2| \le |\hat{\sigma}_\omega^2 - \hat{\sigma}_{\omega'}^2| + |\sigma_{\omega'}^2 - \sigma_\omega^2|.$$

The first right term is bounded by

$$\begin{aligned}
&\left|\hat{\sigma}_\omega^2 - \hat{\sigma}_{\omega'}^2\right| \\
&= 4\left|\frac{1}{n^3}\sum_{ij\ell} H_{i\ell}^{*(\omega)} H_{j\ell}^{*(\omega)} - \frac{1}{n^3}\sum_{ij\ell} H_{i\ell}^{*(\omega')} H_{j\ell}^{*(\omega')} - \frac{1}{n^4}\sum_{ijab} H_{ij}^{*(\omega)} H_{ab}^{*(\omega)} + \frac{1}{n^4}\sum_{ijab} H_{ij}^{*(\omega')} H_{ab}^{*(\omega')}\right| \\
&\le \frac{4}{n^3}\sum_{ij\ell}\left|H_{i\ell}^{*(\omega)} H_{j\ell}^{*(\omega)} - H_{i\ell}^{*(\omega')} H_{j\ell}^{*(\omega')}\right| + \frac{4}{n^4}\sum_{ijab}\left|H_{ij}^{*(\omega)} H_{ab}^{*(\omega)} - H_{ij}^{*(\omega')} H_{ab}^{*(\omega')}\right| \\
&\le \frac{4}{n^3}\sum_{ij\ell}\left|H_{i\ell}^{*(\omega)} H_{j\ell}^{*(\omega)} - H_{i\ell}^{*(\omega)} H_{j\ell}^{*(\omega')}\right| + \left|H_{i\ell}^{*(\omega)} H_{j\ell}^{*(\omega')} + H_{i\ell}^{*(\omega')} H_{j\ell}^{*(\omega')}\right| \\
&\quad + \frac{4}{n^4}\sum_{ijab}\left|H_{ij}^{*(\omega)} H_{ab}^{*(\omega)} - H_{ij}^{*(\omega)} H_{ab}^{*(\omega')}\right| + \left|H_{ij}^{*(\omega)} H_{ab}^{*(\omega')} - H_{ij}^{*(\omega')} H_{ab}^{*(\omega')}\right| \\
&\le \frac{4}{n^3}\sum_{ij\ell}\left|H_{i\ell}^{*(\omega)}\right|\left|H_{j\ell}^{*(\omega)} - H_{j\ell}^{*(\omega')}\right| + \left|H_{j\ell}^{*(\omega')}\right|\left|H_{i\ell}^{*(\omega)} + H_{i\ell}^{*(\omega')}\right| \\
&\quad + \frac{4}{n^4}\sum_{ijab}\left|H_{ij}^{*(\omega)}\right|\left|H_{ab}^{*(\omega)} - H_{ab}^{*(\omega')}\right| + \left|H_{ab}^{*(\omega')}\right|\left|H_{ij}^{*(\omega)} - H_{ij}^{*(\omega')}\right| \\
&\le \frac{4}{n^3}\sum_{ij\ell} 2\left(3\nu\cdot 3L_k\|\omega-\omega'\|\right) + \frac{4}{n^4}\sum_{ij\ell} 2\left(3\nu\cdot 3L_k\|\omega-\omega'\|\right) \\
&= 144\nu L_k\|\omega-\omega'\|.
\end{aligned}$$

Similarly, the second term is bounded by

$$\begin{aligned}
|\sigma_\omega^2 - \sigma_{\omega'}^2| &\le 4\left|\mathbb{E}\left[H_{12}^{*(\omega)} H_{13}^{*(\omega)}\right] - \mathbb{E}\left[H_{12}^{*(\omega')} H_{13}^{*(\omega')}\right]\right| + 4\left|\mathbb{E}\left[H_{12}^{*(\omega)}\right]^2 - \mathbb{E}\left[H_{12}^{*(\omega')}\right]^2\right| \\
&\le 4\mathbb{E}\left|H_{12}^{*(\omega)} H_{13}^{*(\omega)} - H_{12}^{*(\omega')} H_{13}^{*(\omega')}\right| + 4\mathbb{E}\left|H_{12}^{*(\omega)} H_{34}^{*(\omega)} - H_{12}^{*(\omega')} H_{34}^{*(\omega')}\right| \\
&\le 144\nu L_k\|\omega-\omega'\|.
\end{aligned}$$

Thus, we obtain the result

$$|\Psi(\omega)-\Psi(\omega')| \le 288\nu L_k\|\omega-\omega'\|.$$

□

Relying on Lemmas 1 to 3, we provide the result of the uniform convergence of $\hat{\sigma}_\omega^2$.

**Proposition 3.** *Under Assumptions (A) to (C), with probability at least $1-\delta$, we have*

$$\sup_{\omega\in\Omega}\left|\hat{\sigma}_\omega^2 - \sigma_\omega^2\right| \le 252\nu^2\sqrt{\frac{2}{n}\log\frac{2}{\delta} + \frac{2}{n}D\log(4\sqrt{n}R_\Omega)} + \frac{648\nu^2}{n} + \frac{288\nu L_k}{\sqrt{n}}.$$

*Proof.* Similar to Proposition 2, we denote a random error function

$$\Psi(\omega) := \hat{\sigma}_\omega^2 - \sigma_\omega^2$$

Based on Assumption (B), we can use at most $T = (4R_\Omega/q)^D$ points $\{\omega_i\}_{i=1}^T$ such that for any $\omega\in\Omega$, $\min_i\|\omega-\omega_i\| \le q$ (Cucker & Smale (2002), Proposition 5). According to Lemmas 1 and 2 and a union bound, we have that with probability at least $1-\delta$,

$$\begin{aligned}
\max_{i\in\{1,\dots,T\}}|\Psi(\omega_i)| &\le \max_{i\in\{1,\dots,T\}}|\hat{\sigma}_{\omega_i}^2 - \mathbb{E}\,\hat{\sigma}_{\omega_i}^2 + \mathbb{E}\,\hat{\sigma}_{\omega_i}^2 - \sigma_{\omega_i}^2| \le \max_{i\in\{1,\dots,T\}}|\hat{\sigma}_{\omega_i}^2 - \mathbb{E}\,\hat{\sigma}_{\omega_i}^2| + \frac{648\nu^2}{n} \\
&\le 252\nu^2\sqrt{\frac{2}{n}\log\frac{2T}{\delta}} + \frac{648\nu^2}{n} \le 252\nu^2\sqrt{\frac{2}{n}\log\frac{2}{\delta} + \frac{2}{n}D\log\frac{4R_\Omega}{q}} + \frac{648\nu^2}{n}.
\end{aligned}$$

Combing Lemma 3 and setting $q{=}\frac{1}{\sqrt{n}}$, we have that with probability at least $1{-}\delta$

$$\begin{aligned}\sup_{\omega}|\Psi(\omega)| &= |\Psi(\omega^*)| = |\Psi(\omega^*)-\Psi(\omega_*)+\Psi(\omega_*)| \leq |\Psi(\omega_*)|+|\Psi(\omega^*)-\Psi(\omega_*)| \\ &\leq \max_{i\in\{1\dots T\}}\|\Psi(\omega_i)\|+288\nu L_k q \\ &\leq 252\nu^2\sqrt{\frac{2}{n}\log\frac{2}{\delta}+\frac{2}{n}D\log(4\sqrt{n}R_\Omega)}+\frac{648\nu^2}{n}+\frac{288\nu L_k}{\sqrt{n}}.\end{aligned}$$

where $\omega^*=\arg_\omega\sup|\Psi(\omega)|$ and $\omega_*=\arg_\omega\min_{\omega_i\in\{\omega_1\dots\omega_T\}}\|\omega^*-\omega_i\|$. □

Next, relying on the results of Propositions 2 and 3, we begin to prove Theorem 1.

**Theorem 1. (Uniform bound of *MMD-MP*.)** *Let $\omega$ parameterize uniformly bounded kernel functions $k_\omega$ in a Banach space of dimension $D$ with $\|\omega\|{\leq}R_\Omega$, $k_\omega$ be uniformly bounded by $\sup_{\omega\in\Omega}\sup_{\mathbf{x},\mathbf{x}'\in\mathcal{X}}k_\omega(\mathbf{x},\mathbf{x}'){\leq}\nu$ with $L_k$-Lipschitz, i.e., $|k_\omega(\mathbf{x},\mathbf{x}'){-}k_{\omega'}(\mathbf{x},\mathbf{x}')|\leq L_k\|\omega{-}\omega'\|$. Let $\bar{\Omega}_s$ be a set of $\omega$ for which $\sigma^2_{\mathfrak{H}_1^*}{\geq}s^2{>}0$. Taking $\lambda{=}n^{-1/3}$, with probability at least $1{-}\delta$, we have*

$$\sup_{\omega\in\bar{\Omega}_s}\|\hat{J}(S_{\mathbb{P}},S_{\mathbb{Q}};k_\omega)-J(\mathbb{P},\mathbb{Q};k_\omega)\| = \mathcal{O}\left(\frac{\nu}{s^2n^{1/3}}\left[\nu^2\sqrt{D\log(R_\Omega n){+}\log\frac{1}{\delta}}+\nu L_k+\frac{1}{s}\right]\right).$$

*Proof.* Let $\sigma^2_{\omega,\lambda}:=\sigma^2_\omega+\lambda$, we have

$$\begin{aligned}&\sup_{\omega\in\bar{\Omega}_s}|\frac{\hat{\eta}_\omega}{\hat{\sigma}_{\omega,\lambda}}-\frac{\eta_\omega}{\sigma_\omega}| \leq \sup_{\omega\in\bar{\Omega}_s}|\frac{\hat{\eta}_\omega}{\hat{\sigma}_{\omega,\lambda}}-\frac{\hat{\eta}_\omega}{\sigma_{\omega,\lambda}}|+\sup_{\omega\in\bar{\Omega}_s}|\frac{\hat{\eta}_\omega}{\sigma_{\omega,\lambda}}-\frac{\hat{\eta}_\omega}{\sigma_\omega}|+\sup_{\omega\in\bar{\Omega}_s}|\frac{\hat{\eta}_\omega}{\sigma_\omega}-\frac{\eta_\omega}{\sigma_\omega}| \\ &\leq \sup_{\omega\in\bar{\Omega}_s}|\hat{\eta}_\omega|\frac{|\hat{\sigma}^2_{\omega,\lambda}-\sigma^2_{\omega,\lambda}|}{\hat{\sigma}_{\omega,\lambda}\sigma_{\omega,\lambda}(\hat{\sigma}_{\omega,\lambda}+\sigma_{\omega,\lambda})}+\sup_{\omega\in\bar{\Omega}_s}|\hat{\eta}_\omega|\frac{|\sigma^2_{\omega,\lambda}-\sigma^2_\omega|}{\sigma_{\omega,\lambda}\sigma_\omega(\sigma_{\omega,\lambda}+\sigma_\omega)}+\sup_{\omega\in\bar{\Omega}_s}\frac{1}{\sigma_\omega}|\hat{\eta}_\omega-\eta_\omega| \\ &\leq \sup_{\omega\in\bar{\Omega}_s}\frac{3\nu}{\lambda s+\sqrt{\lambda}s^2}|\hat{\sigma}^2_\omega-\sigma^2_\omega|+\frac{3\nu\lambda}{(s^2+\lambda)s+\sqrt{s^2+\lambda}s^2}+\sup_{\omega\in\bar{\Omega}_s}\frac{1}{s}|\hat{\eta}_\omega-\eta_\omega| \\ &\leq \frac{3\nu}{\sqrt{\lambda}s^2}\sup_{\omega\in\bar{\Omega}_s}|\hat{\sigma}^2_\omega-\sigma^2_\omega|+\frac{3\nu\lambda}{2s^3}+\frac{1}{s}\sup_{\omega\in\bar{\Omega}_s}|\hat{\eta}_\omega-\eta_\omega|.\end{aligned}$$

According to Propositions 2 and 3, with probability at least $1{-}\delta$, the error bound is at most

$$\left(\frac{6\nu}{s\sqrt{n}}+\frac{756\nu^3}{s^2\sqrt{n\lambda}}\right)\sqrt{2\log\frac{2}{\delta}+2D\log(4R_\Omega\sqrt{n})}+\left(\frac{864\nu^2}{s^2\sqrt{n\lambda}}+\frac{6}{s\sqrt{n}}\right)L_k+\frac{3\nu\lambda}{2s^3}+\frac{1944\nu^3}{s^2n\sqrt{\lambda}}$$

Taking $\lambda=n^{-1/3}$, we get

$$\begin{aligned}&\left(\frac{6\nu}{s\sqrt{n}}+\frac{756\nu^3}{s^2n^{1/3}}\right)\sqrt{2\log\frac{2}{\delta}+2D\log(4R_\Omega\sqrt{n})}+\left(\frac{864\nu^2}{s^2n^{1/3}}+\frac{6}{s\sqrt{n}}\right)L_k+\frac{3\nu}{2s^3n^{1/3}}+\frac{1944\nu^3}{s^2n^{5/6}} \\ &=\frac{1}{s^2n^{1/3}}\left[\left(\frac{6\nu s}{n^{1/6}}{+}756\nu^3\right)\sqrt{2\log\frac{2}{\delta}{+}2D\log(4R_\Omega\sqrt{n})}{+}\left(864\nu^2{+}\frac{6s}{n^{1/6}}\right)L_k{+}\frac{3\nu}{2s}{+}\frac{1944\nu^3}{n^{1/2}}\right]\end{aligned} \tag{19}$$

Thus, we have that with probability at least $1{-}\delta$,

$$\sup_{\omega\in\bar{\Omega}_s}\|\hat{J}(S_{\mathbb{P}},S_{\mathbb{Q}};k_\omega)-J(\mathbb{P},\mathbb{Q};k_\omega)\| = \mathcal{O}\left(\frac{\nu}{s^2n^{1/3}}\left[\nu^2\sqrt{D\log(R_\Omega n){+}\log\frac{1}{\delta}}+\nu L_k+\frac{1}{s}\right]\right)$$

□

Note that the error bound of the optimization is close to the result of Liu et al. (2020), however, the detailed coefficient of each term in Eqn. (19) is totally different due to the different $H$ and $H^*$.

# B Related Work

## B.1 Large Language Models

Large Language Models (LLMs) hold significant importance in various fields due to their capabilities in natural language understanding (Ouyang et al., 2022; Bender & Koller, 2020; Allen, 1995; Wang et al., 2018), and demonstrate exceptional performance in several downstream tasks such as dialogue generation (Li et al., 2016; Huang et al., 2018; Ma et al., 2020), machine translation (Bahdanau et al., 2014; Wu et al., 2016; Caron et al., 2021) and text annotation (Pei et al., 2023; Pangakis et al., 2023; Tang et al., 2023b).

The transformer architecture (Bahdanau et al., 2014; Vaswani et al., 2017; Xiong et al., 2020) plays a pivotal role in language modeling by introducing a self-attention mechanism, enabling the model to selectively focus on different segments of the input sequence. Building upon the transformer, Devlin et al. (2018) introduce the Bidirectional Encoder Representations from Transformers (BERT), a standout model in the series of masked language models. Moreover, Liu et al. (2019) introduce RoBERTa, which employs a robust optimization strategy applied to the BERT pretraining method, achieving superior performance across various tasks compared to BERT.

Another well-known category of language models belongs to the GPT series (Radford et al., 2018; 2019; Black et al., 2021; Toan, 2023; Black et al., 2021). For example, Radford et al. (2018) introduce the generative pre-trained Transformer (GPT), which demonstrates remarkable performance across diverse language generation tasks due to extensive training on large-scale text data. Subsequently, GPT2 (Radford et al., 2019), GPT3 (Toan, 2023), and GPT-Neo (Black et al., 2021), expand the model structure, incorporating more parameters and training on a broader corpus, consequently surpassing the performance of their predecessors. OpenAI releases ChatGPT (OpenAI, 2022), which is trained on the GPT-3.5 architecture. This excellent model continually enhances its generative capabilities through Reinforcement Learning from Human Feedback (RLHF) (Christiano et al., 2017; Stiennon et al., 2020). With its revolutionary performance in generating coherent and contextually relevant text, GPT-4 (OpenAI, 2023) is expected to further advance the capabilities of large language models, pushing the boundaries of language understanding and generation.

## B.2 MGT Detection

Various machine-generated text (MGT) detection methods (Solaiman et al., 2019; Dou et al., 2022; Yang et al., 2021; Mitchell et al., 2023; Kirchenbauer et al., 2023; Tang et al., 2023a) have been developed and shown promising performance. In general, current detection methods can be roughly divided into two categories, *i.e.*, metric-based methods and model-based methods. Specifically, the former employ statistical metrics (*e.g.*, likelihood and entropy) extracted by LLMs to detect outliers, while the latter finetune a pre-trained LM (*e.g.*, RoBERTa (Liu et al., 2019)) to identify MGTs.

**Metric-based MGT detection methods.** These approaches leverage pre-trained LLMs or scoring models to measure the statistical discrepancy between human-written texts and machine-generated texts. Among them, the commonly used metrics involve log-likelihood (Solaiman et al., 2019), entropy, rank (Gehrmann et al., 2019) and log-rank (Mitchell et al., 2023). Different from the above, DetectGPT (Mitchell et al., 2023) proposes to compare log probability of texts under multiple perturbations, based on the assumption that the MGTs are more likely to lie in the local optimal of log probability, which achieves higher AUROC compared with other metrics. However, these metric-based detection methods are compromised with inferior performance when encountering a large language-domain gap between the generated text model and the scoring model.

**Model-based MGT detection methods.** These methods usually train a classification model using texts provided by both humans and LLMs. To be specific, OpenAI-D (Solaiman et al., 2019) finetunes a RoBERTa model with GPT2-generated texts and is used in detecting GPT2 outputs. ChatGPT-D (Guo et al., 2023) devises two manners (*i.e.*, using pure answered text or QA pairs) to train the model using HC3 dataset (Guo et al., 2023). Besides, Kumarage et al. (2023) train a classifier by combining standardized stylistic features and LLM-based text embeddings. Abburi et al. (2023) propose an ensemble neural network model using probabilities from pre-trained language models as features to train a text classifier. These methods train the classifier severely relying on MGTs, leading to unsatisfactory transferability for MGT detection.

More recently, an alternative detection paradigm is the watermark-based detection (He et al., 2022; Kirchenbauer et al., 2023), which is defined as unseen modifications to texts that hide identifying information. For example, Kirchenbauer et al. (2023) propose to inject the watermark by selecting a randomized "green token" set before a word is generated, and softly promoting "green tokens" while sampling. Such a watermark is hard to remove and can be detected by a statistical test with $p$-values. However, these methods rely on a tailored language model to add the watermark and thus only distinguish the texts generated by this model, limiting their application scenarios.

### B.3 Two-sample Test

Two-sample test (Gretton et al., 2012; Liu et al., 2020; Lopez-Paz & Oquab, 2017; Cheng & Cloninger, 2022) is a hypothesis test that aims to determine whether two populations are from a congruent distribution (Borgwardt et al., 2006). Since the traditional statistical methods such as t-tests and Kolmogorov-Smirnov tests (Larsen & Marx, 2013) are stuck with complex assumptions and low-dimensional spaces, a board set of kernel-based methods have surged to prominence, which construct a kernel mean embedding for each distribution and measure the difference between them.

Maximum mean discrepancy (MMD) serves as a highly efficacious metric to distinguish between two distributions (Gretton et al., 2012; Gao et al., 2021; Zhang et al., 2023a). Tolstikhin et al. (2016) further derive lower bounds for MMD estimation based on finite data for a radial universal kernel. To address kernel adaptation for the quadratic-time MMD, Liu et al. (2020) propose to choose the best kernel by splitting data. In addition, Kim et al. (2022) propose an adaptive two-sample test for testing equality between two Hölder densities supported on the real $d$-dimensional unit ball. Nonetheless, limited research has explored the optimization mechanism of kernel-based MMD. In this paper, we delve extensively into this through comprehensive empirical investigations and propose a novel optimization method to further improve the stability of training for kernel-based MMD.

An alternative strategy to compare distributions involves training a classifier between them, subsequently evaluating its accuracy. These approaches are commonly referred to as classifier two-sample tests. Among them, C2ST-S (Lopez-Paz & Oquab, 2017) uses the output of the softmax layer as representation, while the C2ST-L (Cheng & Cloninger, 2022) uses the output logits instead. These several methods are provable to be encompassed by kernel-based MMD (Liu et al., 2020).

# C More Details for Experiment Settings

## C.1 Datasets

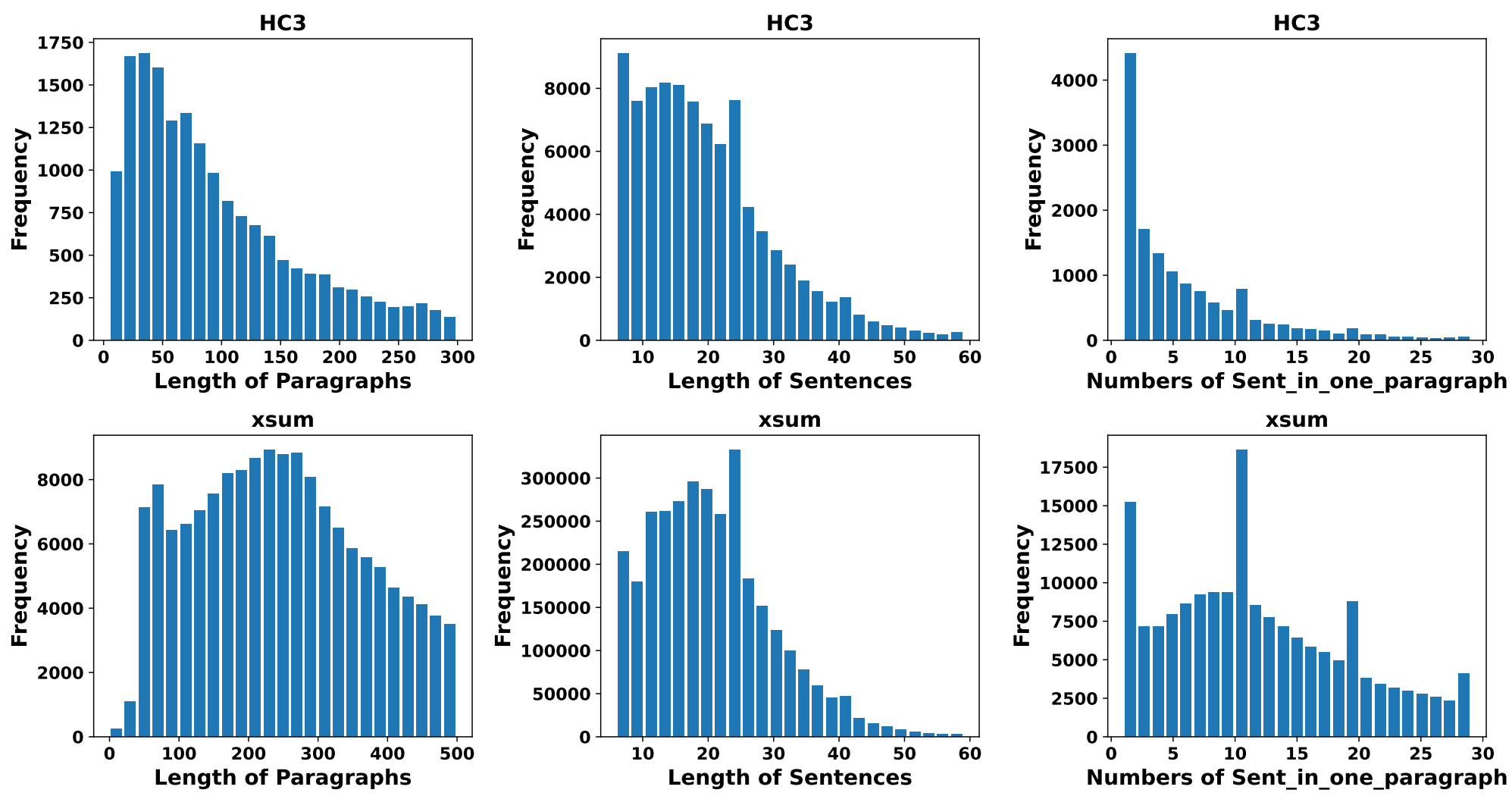

Figure 6: Illustration of statistical distributions of texts in HC3 and XSum.

Table 7: Parameter size of the pretrained LLMs adopted in the experiment.

| Model | GPT2 | Neo-S | GPT2-M | GPT3-S | Neo-L | GPT-j-6b | GPT4all-j |
|---|---|---|---|---|---|---|---|
| Parameters | 124M | 125M | 355M | 551M | 2.7B | 6B | 6B |

Human-written texts (HWTs) in our experiment come from the Human ChatGPT Comparison Corpu (HC3) dataset (Guo et al., 2023) and the Extreme Summarization (XSum) dataset (Güera & Delp, 2018). Specifically, HC3 contains $24,322$ question-answer pairs in the form of both long short-level paragraphs or sentences, covering a wide range of domains such as open-domain, computer science, finance, medicine, law, and psychology. Meanwhile, XSum contains $226,711$ news articles from BBC ranging from 2010 to 2017 and covering domains including news, politics, sports, weather, business, technology, science, health, family, education, entertainment and arts. For paragraph-based detection, we choose paragraphs with more than $5$ sentences for testing, while for sentence-based detection, we choose sentences with more than $5$ words since shorter sentences with fewer than $5$ words could be difficult for us to classify into some category. We also provide statistical distributions of texts from both HC3 and XSum in Figure 6, including the length of paragraphs, the length of sentences and the number of sentences in one paragraph.

For machine-generated texts (MGTs), we consider commonly used LLMs to generate, including ChatGPT (OpenAI, 2022), GPT2 series (Radford et al., 2019), GPT3-S (Toan, 2023), GPT-Neo series (Black et al., 2021), GPT4all-j (Anand et al., 2023), where the model parameters are shown in Table 7. Since HC3 already provides the texts generated by ChatGPT, we utilize the remaining models to generate texts according to the first 20 prompts of HWTs. The text generation strategy is similar to that of Mitchell et al. (2023).

## C.2 Implementation Details of Our Method

**Architecture Design of Deep Kernel.** The deep kernel $\phi_{\hat{f}}$ in our MMD-MP is a neural network $\phi$ equipped with a feature extractor $\hat{f}$. For the feature extractor $\hat{f}$, we employ OpenAI's RoBERTa-based GPT-2 detector model (Liu et al., 2019), which is the same as that of OpenAI-D (Solaiman et al., 2019), and consider its last hidden state as the feature of the input text. Each token in this feature extractor has a dimension of 768, and we set a maximum of 100 tokens per sentence. The network $\phi$ consists of a hidden-layer transformer followed by a projector and a multi-layer perceptron (MLP), where the projector reduces the data dimension from 768 to 512, while the MLP reduces the flattened data dimension from $51,200$ to 300. The data dimension during the whole procedure when

feeding a sentence into the kernel follows the sequence: $100\times768\rightarrow100\times512\rightarrow51,200\rightarrow300$. Note that we only optimize the network $\phi$ and fix the extractor $\hat{f}$ during training through all experiments.

**More detailed algorithms.** Our method is designed to effectively detect whether the given texts are from humans. To summarize, we first extract features from both HWT and MGT sentences using the fixed feature extractor $\hat{f}$. Then, we train the deep kernel $\phi_{\hat{f}}$ by maximizing the objective $J$ in Eqn. (10) (Algorithm 1) for both paragraph-level and sentence-level detection scenarios.

For paragraph-level detection, we compute the estimated MMD between given HWTs and test paragraph sentences ("$est$"). We then iteratively randomize and split these mixed sentences, calculating MMD values as "$perm_i$". We obtain test power by comparing "$perm_i$" with "$est$" (Algorithm 2). For sentence-level detection, using a set of HWTs as a reference set, we compute the MMD distance between the test single sentence and reference samples (Algorithm 3).

We conduct our experiments using Python 3.9 and Pytorch 2.0 on a server with 1× NVIDIA A800 GPU. We use Adam optimizer (Kingma & Ba, 2015; Yang et al., 2023) to optimize the kernel parameters. In Algorithm 1, we set $\lambda$ to $10^{-8}$ and batch size to 200, and learning rate to 0.00005 on HC3 and 0.00001 on XSum in all experiments. Following the setting of two-sample test in (Liu et al., 2020), we set the threshold $\alpha = 0.05$ to determine whether to reject or accept the null hypothesis when testing for two-sample test.

### C.3 Implementation Details of Baselines

**Two-sample test baselines.** For two-sample test, we compare our method with MMD-O, MMD-D (Liu et al., 2020), C2ST-S (Lopez-Paz & Oquab, 2017) and C2ST-L (Lopez-Paz & Oquab, 2017). The architectures of the first two baselines are the same as our MMD-MP, and the latter two are classifier two-sample test methods that use the same architecture as our MMD-MP except for an additional binary classifier. We run these baselines using the code [1] from Liu et al. (2020). Moreover, we set the learning rates of these four baselines to be 0.00005, 0.00005, 0.0001 and 0.0001, respectively on HC3, while 0.00001, 0.00001, 0.0005 and 0.0005 on XSum.

**Single-instance detection baselines.** For single-instance detection, we compare with metric-based detectors, *e.g.*, Log-Likelihood (Solaiman et al., 2019), Entropy, Rank (Gehrmann et al., 2019), Log-Rank and DetectGPT (Mitchell et al., 2023) and model-based detectors, *e.g.*, OpenAI-D (Solaiman et al., 2019) and ChatGPT-D (Guo et al., 2023). We implement these methods based on the code [2] from He et al. (2023). In addition, we use cross-entropy loss to optimize a deep neural network as a baseline, called CE-Classifier, whose model is the same as that of MMD-D and MMD-MP except for an additional binary classifier. Similar to the training of MMD-D and MMD-MP, we fix the extractor and only update the kernel network and the binary classifier, where the learning rate is set to 0.0001 on HC3 and 0.0005 on XSum, respectively.

**Kernel hyper-parameters selection.** For all experiments, we set $\epsilon$=$10^{-10}$ and $\sigma_q$=30. In the case of our MMD-MP, we set $\sigma_\phi$=55 when the sample size $n$=$3,100$ and $\sigma_\phi$=45 when $n$=$1,000$. Accordingly, for MMD, we set $\sigma_\phi$=20 when the sample size $n$=$3,100$ and $\sigma_\phi$=55 when $n$=$1,000$.

Through all experiments, we evaluate the methods under the training set with $3,100$ or $1,000$ paragraphs. During testing, we randomly take 100 paragraphs for two-sample test and $1,000$ sentences for single-instance detection. For each method, we repeat the experiments 10 times for synthetic data or 5 times for real-world data to ensure experimental reliability.

### C.4 Implementation Details on Synthetic Data

For the toy experiment in Section 4.1, we use the network following *high-dimensional Gaussian mixtures* settings of Liu et al. (2021). We train the deep kernel using synthetic data consisting of 200 instances with a learning rate of 0.00005 and test using $1,000$ sets, each containing 10 instances. Moreover, we consider different $\mu$ to represent data with different variances, *i.e.*, larger $\mu$ means larger variance. Furthermore, we use $L_2$-norm of the variance of data in $\mathbb{Q}$ to represent its variance.

[1] https://github.com/fengliu90/DK-for-TST
[2] https://github.com/xinleihe/MGTBench

# D More Intuitive Explanations for the optimization of Kernel-based MMD

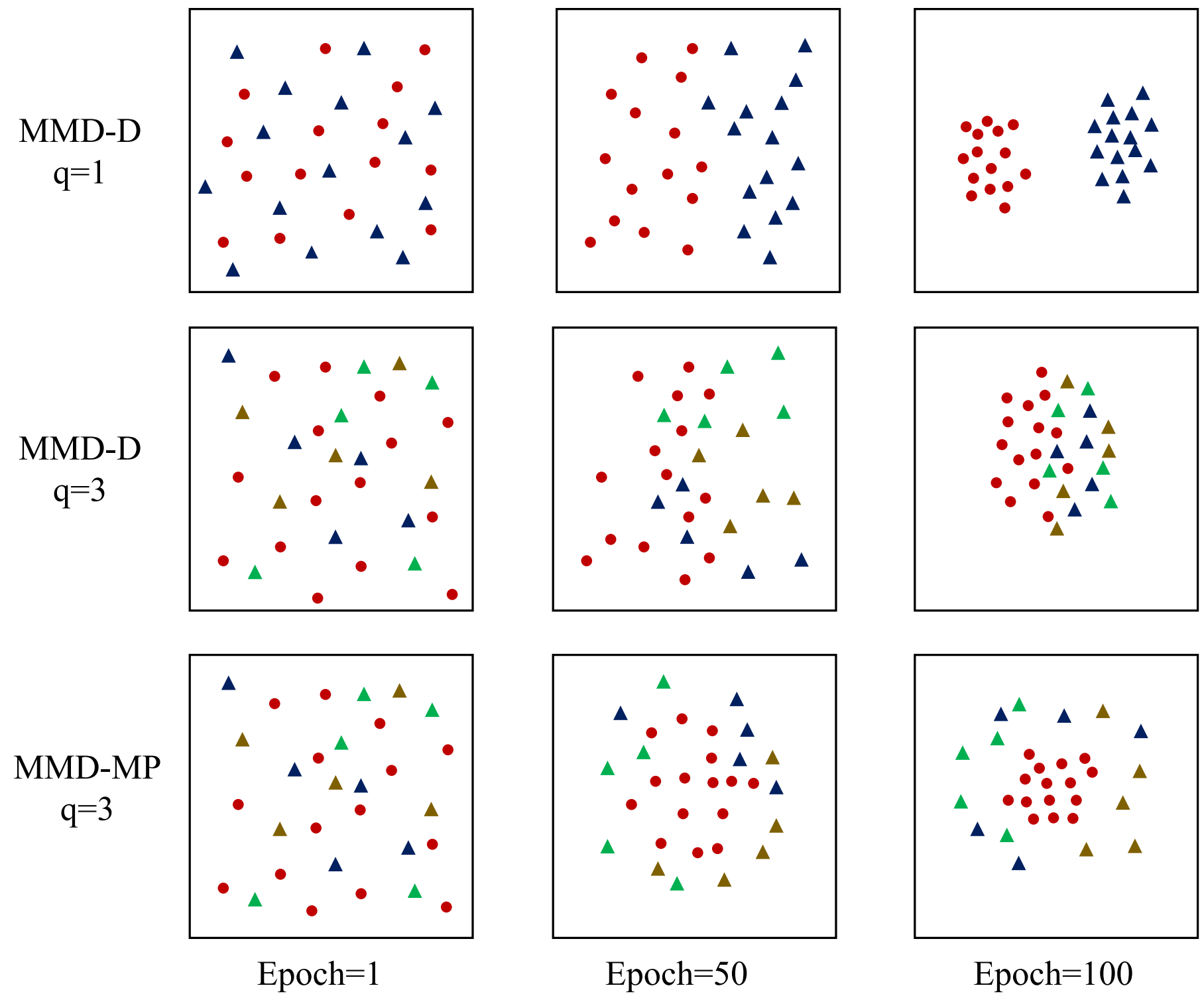

Figure 7: Illustration of the optimization of MMD-D and our MMD-MP under different $q$ MGT populations, where the red dots represent HWT samples, while the triangles of other colors represent MGT samples.

In Section 2.3, we have offered some insights into our motivation through empirical observations and detailed analyses to comprehend the optimization process of MMD-D and our MMD-MP. To gain a clear understanding, we provide additional intuitive figures, as demonstrated in Figure 7. For MMD-D optimization, it intuitively tends to separately aggregate human-written texts (HWTs) and all possible different-population machine-generated texts (MGTs), such as decreasing the intra-class distance, and simultaneously push them away from each other like increasing inter-class distance. When the MGT population $S^{tr}_{\mathbb{Q}}$ comprises different populations, *e.g.*, $q = 3$, this optimization presents challenges due to significant fluctuations (as discussed in Section 2.3). In contrast, our MMD-MP relaxes constraints on aggregating all MGTs populations (*i.e.*, removing the $k(Y, Y')$ from MMD), focusing more on fitting HWTs. This stabilizes MMD values and enhances transferability of detection (as discussed in Section 3.3).

## E IMPACT OF USING VARIANCE IN $\hat{J}$ DURING TRAINING

In our MMD-MP, we train the deep kernel $k_\omega$ by minimizing the objective $\hat{J}$ in Eqn. (7), which is the ratio of $\widehat{\mathrm{MPP}}_u$ and $\hat{\sigma}^2_{\mathfrak{H}_1^*}$. In this experiment, we investigate the impact of variance $\hat{\sigma}^2_{\mathfrak{H}_1^*}$. To achieve this, we remove the variance and only minimize $\widehat{\mathrm{MPP}}_u$ instead. Tables 8 and 9 show test power and AUROC of our method compared with that without using variance in $\hat{J}$ on HC3 given $3,100$ processed paragraphs generated by different LLMs. Obviously, MMD-MP demonstrates significant performance drop when variance optimization is omitted (*e.g.*, $6.62\% \downarrow$ of test power and $2.95\% \downarrow$ of AUROC on average). This decline emphasizes the significance of incorporating variance $\hat{\sigma}^2_{\mathfrak{H}_1^*}$ within the training process of kernel-based MMD to ensure stability in discrepancy estimation.

Table 8: Impact of using variance $\hat{\sigma}^2_{\mathfrak{H}_1^*}$ in $\hat{J}$ during training in terms of test power/100 on HC3 given $3,100$ processed paragraphs in training data.

| Method | ChatGPT | GPT3-S | Neo-S | ChatGPT Neo-S | ChatGPT GPT3-S |
|---|---|---|---|---|---|
| MMD-MP (w/o variance) | $90.78_{\pm1.11}$ | $80.74_{\pm2.88}$ | $69.76_{\pm2.70}$ | $73.87_{\pm3.25}$ | $81.09_{\pm2.00}$ |
| MMD-MP (w/ variance) | $\mathbf{93.21}_{\pm1.35}$ | $\mathbf{89.36}_{\pm2.91}$ | $\mathbf{79.68}_{\pm2.42}$ | $\mathbf{89.63}_{\pm1.94}$ | $\mathbf{91.96}_{\pm0.62}$ |

Table 9: Impact of using variance $\hat{\sigma}^2_{\mathfrak{H}_1^*}$ in $\hat{J}$ during training in terms of AUROC/100 on HC3 given $3,100$ processed paragraphs in training data.

| Method | ChatGPT | GPT3-S | Neo-S | ChatGPT Neo-S | ChatGPT GPT3-S |
|---|---|---|---|---|---|
| MMD-MP (w/o variance) | $89.39_{\pm1.66}$ | $88.93_{\pm1.32}$ | $88.53_{\pm1.07}$ | $90.06_{\pm0.75}$ | $92.47_{\pm0.45}$ |
| MMD-MP (w/ variance) | $\mathbf{96.20}_{\pm0.28}$ | $\mathbf{95.08}_{\pm0.32}$ | $\mathbf{92.04}_{\pm0.58}$ | $\mathbf{92.48}_{\pm0.37}$ | $\mathbf{94.61}_{\pm0.22}$ |

## F IMPACT OF USING MPP FOR TESTING

In our approach, we exclusively employ our proposed MPP during the training of the deep kernel but calculate MMD as a metric during testing. In this experiment, we investigate the impact of using MPP during testing. To this end, we replace $\widehat{\mathrm{MMD}}_u^2$ in Algorithms 2 and 3 with $\widehat{\mathrm{MPP}}_u$, denoting it as MMD-MP (MPP), which is distinct from our original method, MMD-MP (MMD). In Table 10, we synthesize a four-center Gaussian mixture data as used in Section 4.1 with $\mu \in \{0.02 \times i\}_{i=0}^{20}$, and distinguish them from the standard Gaussian distribution $\mathcal{N}(\mathbf{0}^d, \mathbf{I}^d)$. In Table 11, we directly test the transferability on unknown texts to further demonstrate the impact of using MPP during testing. The results in Tables 10 and 11 reveal that the performance of these two strategies is almost identical, which verifies the claim on the impact of MPP during testing in Section 3.3.

Table 10: Impact of using MPP during testing in terms of test power/100 on Synthetic data with different $\mu$.

| Data ($\mu$) | 0.40 | 0.38 | 0.36 | 0.34 | 0.32 | 0.30 | 0.28 |
|---|---|---|---|---|---|---|---|
| MMD-MP (MPP) | $99.84_{\pm0.07}$ | $99.63_{\pm0.13}$ | $99.22_{\pm0.21}$ | $98.48_{\pm0.43}$ | $97.11_{\pm0.71}$ | $94.74_{\pm1.28}$ | $90.95_{\pm2.19}$ |
| MMD-MP (MMD) | $99.84_{\pm0.06}$ | $99.64_{\pm0.14}$ | $99.24_{\pm0.26}$ | $98.47_{\pm0.37}$ | $97.20_{\pm0.73}$ | $94.77_{\pm1.21}$ | $91.03_{\pm2.06}$ |
| Data ($\mu$) | 0.26 | 0.24 | 0.22 | 0.20 | 0.18 | 0.16 | 0.14 |
| MMD-MP (MPP) | $85.58_{\pm3.13}$ | $78.01_{\pm4.14}$ | $69.11_{\pm5.02}$ | $59.75_{\pm5.34}$ | $51.18_{\pm5.21}$ | $44.60_{\pm4.56}$ | $39.59_{\pm4.22}$ |
| MMD-MP (MMD) | $85.53_{\pm3.23}$ | $78.13_{\pm4.23}$ | $69.09_{\pm4.94}$ | $59.61_{\pm5.48}$ | $51.17_{\pm5.23}$ | $44.77_{\pm4.61}$ | $39.55_{\pm4.28}$ |
| Data ($\mu$) | 0.12 | 0.10 | 0.08 | 0.06 | 0.04 | 0.02 | 0 |
| MMD-MP (MPP) | $36.03_{\pm3.47}$ | $33.29_{\pm2.85}$ | $31.48_{\pm2.23}$ | $30.17_{\pm1.97}$ | $29.50_{\pm1.98}$ | $29.04_{\pm1.95}$ | $28.87_{\pm1.92}$ |
| MMD-MP (MMD) | $35.89_{\pm3.51}$ | $33.22_{\pm2.73}$ | $31.43_{\pm2.28}$ | $30.46_{\pm2.10}$ | $29.62_{\pm2.13}$ | $29.17_{\pm2.04}$ | $28.98_{\pm1.96}$ |

Table 11: Impact of using MPP during testing in terms of test power/100 on HC3 given $3,100$ processed paragraphs in training data, where the deep kernel is trained by ChatGPT, Neo-S and GPT3-S but tested on other unknown texts.

| Method | Neo-L | GPT-j-6b | GPT4all-j |
|---|---|---|---|
| MMD-MP (MPP) | $61.01_{\pm2.22}$ | $55.34_{\pm3.44}$ | $74.93_{\pm0.36}$ |
| MMD-MP (MMD) | $61.00_{\pm2.17}$ | $55.64_{\pm3.48}$ | $74.99_{\pm0.39}$ |

# G REPLACING $k_\omega(X, X')$ WITH $k_\omega(Y, Y')$ IN MPP

In this section, we conduct an ablation study on HC3 by replacing the $k_\omega(X, X')$ with $k_\omega(Y, Y')$ in our proposed MPP in Eqn. (5) to comprehensively investigate its effectiveness.

We report test power and AUROC in Tables 12 and 13 given $1,000$ processed paragraphs in training data, where MMD-MP* replaces $k_\omega(X, X')$ with $k_\omega(Y, Y')$ in MPP when training. We observe that MMD-MP* has a large performance gain on test power compared with MMD-D when training with one-population data, but it degrades test power compared with MMD-MP and this gap widens as the number of population *i.e.*, $q$ increases. This phenomenon arises due to the challenge of optimizing the terms related to $S^{tr}_{\mathbb{Q}}$ when $q$ increases, as suggested by the first conclusion in Section 2.3.

Moreover, from Table 13, in the context of sentence-based detection, the performance of the MMD-MP* exhibits a substantial deterioration in terms of AUROC. One potential explanation for this phenomenon lies in the requirements of single-instance detection, which necessitates a referenced set $S^{re}_{\mathbb{P}}$ to calculate the distance between the test instance and the instances in $S^{re}_{\mathbb{P}}$. If we optimize $k_\omega(\mathbf{y}, \mathbf{y}')$ while neglecting optimization of $k_\omega(\mathbf{x}, \mathbf{x}')$, it could result in an effective aggregation of HWT instances while causing the MGT instances to diverge relatively. This could potentially lead to HWT instances "*enveloping*" MGT instances in a relatively divergent manner, thereby resulting in a situation where the distance between those referenced HWT instances and an HWT instance is likely greater than the distance between them and an MGT instance. Furthermore, even if we interchange the labels of MGT and HWT instances during testing, *i.e.*, replacing the results in the penultimate row of Table 13 with their complements to 100, the performance of MMD-MP* still remains inferior to that of our MMD-MP. In total, training with our proposed MMD-MP achieves consistent and significant performance improvements regardless of AUROC.

Table 12: Test power/100 on HC3 given $1,000$ processed paragraphs in training data, where MMD-MP* replaces $k_\omega(X, X')$ with $k_\omega(Y, Y')$ in MPP when training.

| Method | ChatGPT | GPT3-S | Neo-S | ChatGPT Neo-S | ChatGPT GPT3-S | ChatGPT GPT2-M GPT2-S | ChatGPT Neo-S GPT3-S | ChatGPT Neo-S Neo-L |
|---|---|---|---|---|---|---|---|---|
| MMD-D | $91.38_{\pm2.09}$ | $84.01_{\pm5.04}$ | $72.81_{\pm3.23}$ | $74.22_{\pm4.06}$ | $83.29_{\pm3.05}$ | $62.34_{\pm4.00}$ | $77.76_{\pm2.93}$ | $63.15_{\pm2.38}$ |
| MMD-MP* (Ours) | $\mathbf{92.32}_{\pm1.15}$ | $84.40_{\pm2.97}$ | $75.01_{\pm3.78}$ | $81.11_{\pm3.30}$ | $85.71_{\pm3.36}$ | $76.22_{\pm3.93}$ | $81.00_{\pm2.29}$ | $75.24_{\pm0.76}$ |
| MMD-MP (Ours) | $92.31_{\pm2.30}$ | $\mathbf{86.34}_{\pm5.37}$ | $\mathbf{76.35}_{\pm3.51}$ | $\mathbf{85.30}_{\pm1.99}$ | $\mathbf{89.05}_{\pm1.64}$ | $\mathbf{79.92}_{\pm3.88}$ | $\mathbf{85.54}_{\pm1.93}$ | $\mathbf{79.69}_{\pm0.78}$ |

Table 13: AUROC/100 on HC3 given $1,000$ processed paragraphs in training data, where MMD-MP* replaces $k_\omega(X, X')$ with $k_\omega(Y, Y')$ in MPP when training.

| Method | ChatGPT | GPT3-S | Neo-S | ChatGPT Neo-S | ChatGPT GPT3-S | ChatGPT GPT2-M GPT2-S | ChatGPT Neo-S GPT3-S | ChatGPT Neo-S Neo-L |
|---|---|---|---|---|---|---|---|---|
| MMD-D | $93.86_{\pm0.70}$ | $91.50_{\pm1.24}$ | $81.10_{\pm0.83}$ | $89.28_{\pm0.91}$ | $90.28_{\pm1.59}$ | $85.50_{\pm0.85}$ | $88.07_{\pm0.87}$ | $84.20_{\pm2.33}$ |
| MMD-MP* (Ours) | $6.63_{\pm0.76}$ | $10.48_{\pm1.70}$ | $12.50_{\pm1.50}$ | $11.46_{\pm1.36}$ | $11.07_{\pm0.92}$ | $18.76_{\pm1.25}$ | $11.57_{\pm0.43}$ | $18.85_{\pm1.05}$ |
| MMD-MP (Ours) | $\mathbf{95.95}_{\pm0.42}$ | $\mathbf{94.28}_{\pm0.57}$ | $\mathbf{89.61}_{\pm0.44}$ | $\mathbf{90.83}_{\pm0.79}$ | $\mathbf{93.46}_{\pm0.52}$ | $\mathbf{87.03}_{\pm0.59}$ | $\mathbf{91.25}_{\pm0.56}$ | $\mathbf{86.93}_{\pm0.52}$ |

## H More Results on Optimization Mechanism of MMD-D

In this section, we provide more consequences of $\mathbb{E}(k)$ in MMD and their variances under two optimization methods with different $q$ to further verify the conclusions obtained in Section 2.3.

From Figures 8 (a)-(d), we draw two observations: 1) Both $\mathbb{E}[k_\omega(\mathbf{x},\mathbf{x}')]$ and $\mathbb{E}[k_\omega(\mathbf{y},\mathbf{y}')]$ exhibit a generally increasing trend, while $\mathbb{E}[k_\omega(\mathbf{x},\mathbf{y})]$ shows relatively minor changes. 2) As the number of populations $q$ increases, $\mathbb{E}[k_\omega(\mathbf{y},\mathbf{y}')]$ becomes smaller than $\mathbb{E}[k_\omega(\mathbf{x},\mathbf{x}')]$, with the gap between them widening. These two observations coincide with those of Figure 2 and Section 2.3.

From Figures 9 (a)-(d), we obviously find that 1) The variance of MMD is predominantly determined by the variances of $\mathbb{E}[k_\omega(\mathbf{x},\mathbf{x}')]$ and $\mathbb{E}[k_\omega(\mathbf{y},\mathbf{y}')]$, while the impact of other terms is relatively minor. 2) When $S^{tr}_{\mathbb{Q}}$ comprises $q=4$ or $q=5$ populations, $\mathbb{E}[k_\omega(\mathbf{y},\mathbf{y}')]$ optimized by MMD-D has a significant variance, as well as $\mathbb{E}[k_\omega(\mathbf{x},\mathbf{x}')]$. These two phenomena corroborate the third and fourth observations made in Section 2.3, respectively.

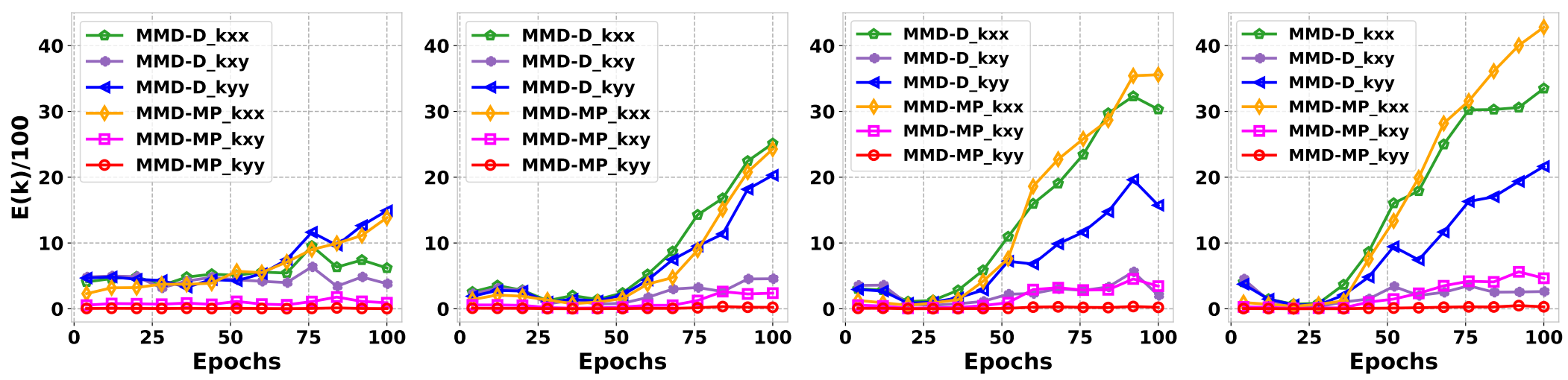

(a) $\mathbb{E}(k)$, $q$=1 (b) $\mathbb{E}(k)$, $q$=2 (c) $\mathbb{E}(k)$, $q$=4 (d) $\mathbb{E}(k)$, $q$=5

Figure 8: $\mathbb{E}(k)$ in MMD when training by MMD-D and MMD-MP (ours) with different $q$.

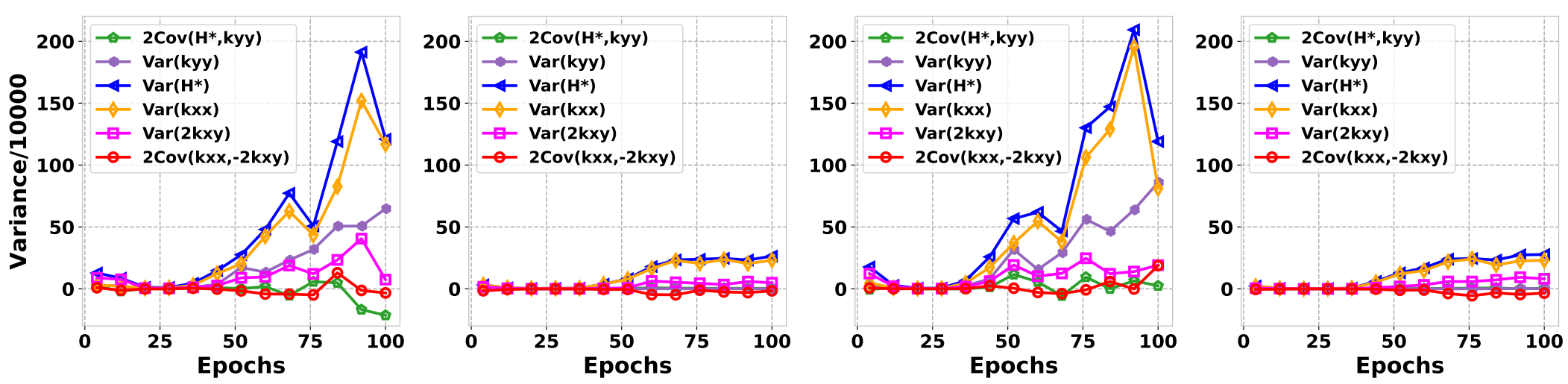

(a) Var(MMD-D), $q$=4 (b) Var(MMD-MP), $q$=4 (c) Var(MMD-D), $q$=5 (d) Var(MMD-MP), $q$=5

Figure 9: Variances related to MMD terms when training by MMD-D and MMD-MP (ours) with different $q$.

## I More Comparison Results over Unknown Texts on HC3

In this section, we conduct more experiments on unknown LLM-text to further evaluate our method. We train the models using texts generated by ChatGPT, GPT-Neo-S and GPT3-S on HC3, and then test on texts generated by GPT-Neo-L, GPT-j-6b and GPT4all-j. From Tables 14-15, our method outperforms the baselines by a large margin on test power and AUROC. Specially, our MMD-MP achieves an absolute improvement of 8.20%∼14.31%↑ on test power over MMD-D. Moreover, our method outperforms CE-Classifier by 3.44% ↑ and MMD-D by 3.55% ↑ of AUROC on average. These results further demonstrate the superior transferability of our method.

Table 14: Test Power/100 on unknown LLM-text.

| Method | Neo-L | GPT-j-6b | GPT4all-j |
|---|---|---|---|
| C2ST-S | $10.99_{\pm 4.18}$ | $8.79_{\pm 2.78}$ | $14.67_{\pm 2.48}$ |
| C2ST-L | $32.83_{\pm 6.77}$ | $27.81_{\pm 3.12}$ | $40.06_{\pm 3.59}$ |
| MMD-O | $12.94_{\pm 1.52}$ | $7.71_{\pm 2.66}$ | $17.08_{\pm 1.14}$ |
| MMD-D | $49.13_{\pm 4.32}$ | $41.33_{\pm 1.29}$ | $66.79_{\pm 1.39}$ |
| MMD-MP (Ours) | $\mathbf{61.00}_{\pm 2.17}$ | $\mathbf{55.64}_{\pm 3.48}$ | $\mathbf{74.99}_{\pm 0.39}$ |

Table 15: AUROC/100 on unknown LLM-text.

| Method | Neo-L | GPT-j-6b | GPT4all-j |
|---|---|---|---|
| CE-Classifier | $76.74_{\pm 1.22}$ | $73.43_{\pm 0.88}$ | $81.96_{\pm 0.89}$ |
| MMD-O | $54.85_{\pm 0.31}$ | $53.85_{\pm 0.86}$ | $52.92_{\pm 1.33}$ |
| MMD-D | $75.97_{\pm 0.73}$ | $74.21_{\pm 1.64}$ | $81.62_{\pm 0.61}$ |
| MMD-MP (Ours) | $\mathbf{79.79}_{\pm 0.17}$ | $\mathbf{77.73}_{\pm 0.86}$ | $\mathbf{84.93}_{\pm 0.51}$ |

# J Comparison Results on XSum

In this section, we provide more experimental results on XSum to further demonstrate the effectiveness of our proposed MMD-MP.

## J.1 Test Power on Paragraph-based Detection

In this part, we compare our MMD-MP with the state-of-the-art (SOTA) two-sample test method for detecting MGT on XSum and show test power on paragraph-based detection in Table 16. In this experiment, we use $1,000$ processed paragraphs to train the model. Experimental results in Table 16 demonstrate that our method significantly outperforms other methods on XSum. Specifically, our MMD-MP surpasses MMD-D by $5.91 \uparrow$ on average even with one training population. Moreover, MMD-MP outperforms MMD-D by an average of $9.43\% \uparrow$ on test power over the two training populations and exhibits an $8.54\% \uparrow$ increase over the three training populations. These results are consistent with the results on HC3, demonstrating the effectiveness of our proposed method.

Table 16: Test power$/100$ on XSum given $1,000$ processed paragraphs in training data.

| Method | GPT2-M | GPT3-S | Neo-S | GPT2-M
Neo-S | GPT2-M
GPT3-S | GPT2-M
Neo-S
GPT3-S | GPT2-M
Neo-S
Neo-L |
|---|---|---|---|---|---|---|---|
| C2ST-S | $9.04_{\pm 3.60}$ | $38.12_{\pm 2.64}$ | $38.31_{\pm 2.95}$ | $20.37_{\pm 2.25}$ | $22.43_{\pm 2.39}$ | $25.94_{\pm 3.29}$ | $17.67_{\pm 1.56}$ |
| C2ST-L | $32.64_{\pm 4.84}$ | $73.64_{\pm 2.03}$ | $76.07_{\pm 1.61}$ | $54.51_{\pm 1.76}$ | $54.85_{\pm 2.46}$ | $61.95_{\pm 2.47}$ | $50.62_{\pm 1.23}$ |
| MMD-O | $8.36_{\pm 1.61}$ | $16.26_{\pm 2.59}$ | $18.76_{\pm 1.33}$ | $12.80_{\pm 1.57}$ | $13.25_{\pm 2.23}$ | $14.25_{\pm 1.79}$ | $10.51_{\pm 1.35}$ |
| MMD-D | $39.99_{\pm 3.78}$ | $73.58_{\pm 4.41}$ | $76.87_{\pm 1.63}$ | $53.88_{\pm 2.82}$ | $57.33_{\pm 2.53}$ | $63.43_{\pm 2.10}$ | $55.06_{\pm 2.74}$ |
| MMD-MP (Ours) | $\mathbf{45.51}_{\pm 1.64}$ | $\mathbf{80.71}_{\pm 1.29}$ | $\mathbf{81.95}_{\pm 2.38}$ | $\mathbf{65.81}_{\pm 3.16}$ | $\mathbf{64.27}_{\pm 2.80}$ | $\mathbf{70.78}_{\pm 1.79}$ | $\mathbf{64.80}_{\pm 1.95}$ |

## J.2 AUROC on Sentence-based Detection

In this part, we compare our MMD-MP with the SOTA single-instance detection method for detecting MGT on XSum in terms of AUROC given $1,000$ processed paragraphs in training data. From Table 17, our MMD-MP consistently achieves the highest AUROC performance compared with other methods. These results further demonstrate the powerful distinguishability of our proposed method when detecting MGTs.

Table 17: AUROC$/100$ on XSum given $1,000$ processed paragraphs in training data.

| Method | GPT2-M | GPT3-S | Neo-S | GPT2-M
Neo-S | GPT2-M
GPT3-S | GPT2-M
Neo-S
GPT3-S | GPT2-M
Neo-S
Neo-L |
|---|---|---|---|---|---|---|---|
| Likelihood | $61.93_{\pm 1.37}$ | $45.97_{\pm 0.62}$ | $43.33_{\pm 1.36}$ | $52.74_{\pm 1.07}$ | $54.46_{\pm 0.95}$ | $50.41_{\pm 0.75}$ | $53.33_{\pm 0.72}$ |
| Rank | $72.11_{\pm 0.65}$ | $64.81_{\pm 1.10}$ | $63.93_{\pm 1.34}$ | $68.18_{\pm 0.82}$ | $68.72_{\pm 0.60}$ | $66.73_{\pm 0.63}$ | $66.72_{\pm 0.48}$ |
| Log-Rank | $65.37_{\pm 1.35}$ | $50.97_{\pm 0.56}$ | $49.46_{\pm 1.39}$ | $57.66_{\pm 1.26}$ | $58.78_{\pm 1.03}$ | $55.16_{\pm 0.93}$ | $57.98_{\pm 0.92}$ |
| Entropy | $54.92_{\pm 1.51}$ | $64.34_{\pm 1.00}$ | $69.85_{\pm 1.63}$ | $61.93_{\pm 1.11}$ | $59.75_{\pm 1.14}$ | $63.47_{\pm 0.25}$ | $61.11_{\pm 1.15}$ |
| DetectGPT-d | $60.08_{\pm 0.70}$ | $44.41_{\pm 1.22}$ | $40.28_{\pm 1.03}$ | $50.09_{\pm 0.81}$ | $52.41_{\pm 1.31}$ | $48.34_{\pm 0.61}$ | $50.96_{\pm 0.99}$ |
| DetectGPT-z | $61.39_{\pm 0.70}$ | $44.45_{\pm 1.20}$ | $40.61_{\pm 1.07}$ | $50.97_{\pm 0.84}$ | $53.05_{\pm 1.17}$ | $48.82_{\pm 0.57}$ | $51.90_{\pm 0.94}$ |
| OpenAI-D | $75.11_{\pm 0.57}$ | $84.56_{\pm 0.71}$ | $86.69_{\pm 0.12}$ | $80.74_{\pm 0.91}$ | $79.91_{\pm 1.02}$ | $82.36_{\pm 1.07}$ | $76.88_{\pm 0.46}$ |
| ChatGPT-D | $54.20_{\pm 1.52}$ | $50.92_{\pm 0.99}$ | $53.60_{\pm 1.40}$ | $54.40_{\pm 1.38}$ | $53.47_{\pm 0.98}$ | $53.69_{\pm 0.88}$ | $53.71_{\pm 0.99}$ |
| CE-Classifier | $78.58_{\pm 1.41}$ | $91.36_{\pm 0.87}$ | $92.89_{\pm 0.35}$ | $84.71_{\pm 0.80}$ | $85.61_{\pm 0.64}$ | $87.57_{\pm 0.30}$ | $84.06_{\pm 0.43}$ |
| MMD-O | $60.29_{\pm 0.66}$ | $65.62_{\pm 0.92}$ | $67.56_{\pm 0.98}$ | $62.01_{\pm 0.75}$ | $64.10_{\pm 1.52}$ | $64.46_{\pm 0.97}$ | $63.14_{\pm 1.10}$ |
| MMD-D | $72.37_{\pm 1.16}$ | $91.48_{\pm 0.39}$ | $90.01_{\pm 0.82}$ | $82.73_{\pm 0.73}$ | $87.59_{\pm 0.80}$ | $86.54_{\pm 0.92}$ | $82.78_{\pm 0.68}$ |
| MMD-MP (Ours) | $\mathbf{82.16}_{\pm 0.79}$ | $\mathbf{91.73}_{\pm 0.41}$ | $\mathbf{93.70}_{\pm 0.45}$ | $\mathbf{86.46}_{\pm 0.60}$ | $\mathbf{87.79}_{\pm 0.77}$ | $\mathbf{88.27}_{\pm 0.71}$ | $\mathbf{85.70}_{\pm 0.54}$ |

## J.3 Results on Challenging Unbalanced Training Data

In this part, we evaluate our method on unbalanced training data containing $2,000$ HWT and $400$ MGT training paragraphs over XSum. In Figure 10, we observe that our approach consistently outperforms baselines in terms of test power and AUROC. Critically, our MMD-MP surpasses the test power of $4.85\% \sim 9.84\% \uparrow$ than MMD-D, highlighting the stability of our method in detecting MGTs under unbalanced training data scenarios.

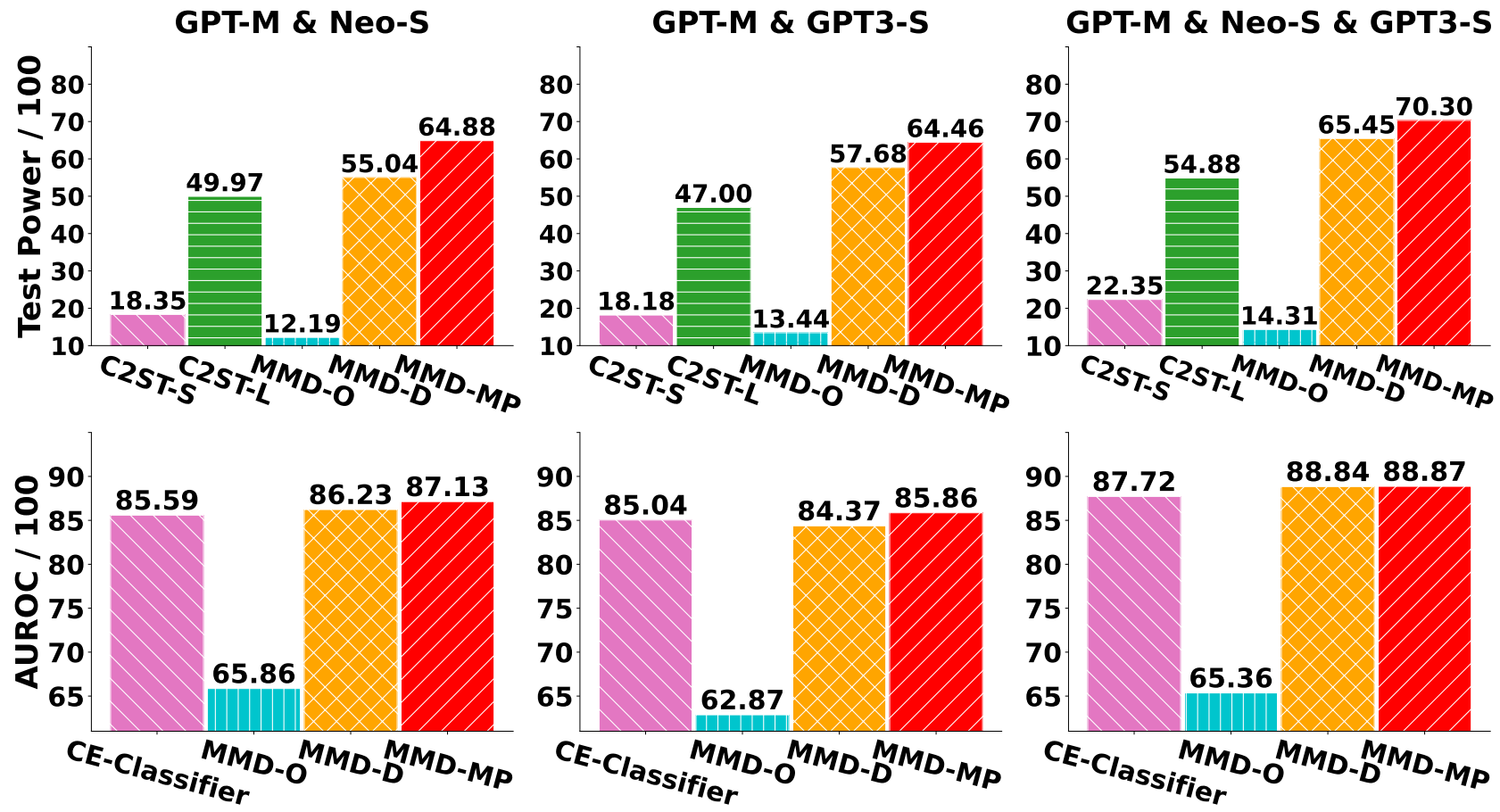

Figure 10: Test power and AUROC on XSum given $2,000$ HWT and $400$ MGT training paragraphs.

## K Results on Detecting Paragraph-level Sentences

It's crucial to note that our method can treat an entire paragraph as a single sentence for detection. However, it's important to note that excessively long paragraphs may exceed the processing capabilities of the pre-trained LLM. When detecting paragraph-level sentences, those that exceed maximum tokens will be truncated by the LLM. Even under this setting, we provide the results of our model to directly test with the random sampled $1,000$ paragraph-level sentences in Table 18. The results still demonstrate the effectiveness of our methods.

Table 18: AUROC/100 on HC3 given $3,100$ processed paragraphs for detecting paragraph-level sentences.

| Method | ChatGPT | GPT3-S | Neo-S | ChatGPT Neo-S | ChatGPT GPT3-S |
|---|---|---|---|---|---|
| Likelihood | $99.34_{\pm 0.11}$ | $74.62_{\pm 0.73}$ | $83.54_{\pm 0.21}$ | $90.76_{\pm 0.06}$ | $86.47_{\pm 1.33}$ |
| Rank | $91.88_{\pm 0.40}$ | $81.67_{\pm 0.75}$ | $87.56_{\pm 0.42}$ | $89.52_{\pm 0.10}$ | $86.83_{\pm 0.55}$ |
| Log-Rank | $99.37_{\pm 0.12}$ | $79.59_{\pm 0.64}$ | $88.00_{\pm 0.25}$ | $93.26_{\pm 0.18}$ | $89.01_{\pm 1.23}$ |
| Entropy | $7.04_{\pm 0.08}$ | $50.35_{\pm 0.14}$ | $47.94_{\pm 0.21}$ | $28.60_{\pm 0.30}$ | $28.73_{\pm 1.13}$ |
| DetectGPT-d | $94.31_{\pm 0.03}$ | $86.63_{\pm 0.92}$ | $87.31_{\pm 0.88}$ | $91.23_{\pm 0.66}$ | $90.43_{\pm 0.75}$ |
| DetectGPT-z | $96.28_{\pm 0.00}$ | $86.55_{\pm 0.94}$ | $87.60_{\pm 0.85}$ | $92.13_{\pm 0.46}$ | $91.36_{\pm 0.73}$ |
| OpenAI-D | $98.75_{\pm 0.08}$ | $99.62_{\pm 0.05}$ | $99.41_{\pm 0.09}$ | $98.94_{\pm 0.14}$ | $98.96_{\pm 0.04}$ |
| ChatGPT-D | $\mathbf{99.79}_{\pm 0.05}$ | $69.03_{\pm 0.03}$ | $67.40_{\pm 1.10}$ | $82.85_{\pm 0.88}$ | $85.88_{\pm 0.05}$ |
| CE-Classifier | $99.13_{\pm 0.15}$ | $98.29_{\pm 0.13}$ | $96.33_{\pm 0.40}$ | $95.71_{\pm 0.12}$ | $94.59_{\pm 0.33}$ |
| MMD-O | $76.29_{\pm 1.92}$ | $87.34_{\pm 0.96}$ | $86.64_{\pm 0.42}$ | $81.63_{\pm 0.74}$ | $81.32_{\pm 0.48}$ |
| MMD-D | $97.42_{\pm 0.19}$ | $99.65_{\pm 0.07}$ | $99.46_{\pm 0.16}$ | $98.65_{\pm 0.16}$ | $99.26_{\pm 0.05}$ |
| MMD-MP (Ours) | $99.59_{\pm 0.08}$ | $\mathbf{99.71}_{\pm 0.05}$ | $\mathbf{99.65}_{\pm 0.08}$ | $\mathbf{99.01}_{\pm 0.17}$ | $\mathbf{99.32}_{\pm 0.13}$ |

## L EXAMPLES OF CHATGPT TEXT DETECTION

To gain a clearer understanding of the effectiveness of our approach, we illustrate some examples of ChatGPT text detection, including storytelling, article writing, and text rewriting. As shown in Figures 11-13, we observe that most of the contents originating from ChatGPT, such as storytelling and article writing, are readily distinguishable by the detector. Sometimes, for some instances of text rewriting, such as human rewriting of ChatGPT text or ChatGPT rewriting human text (as shown in Figure 14), detection is less straightforward. This aligns with the experimental results in Mitchell et al. (2023).

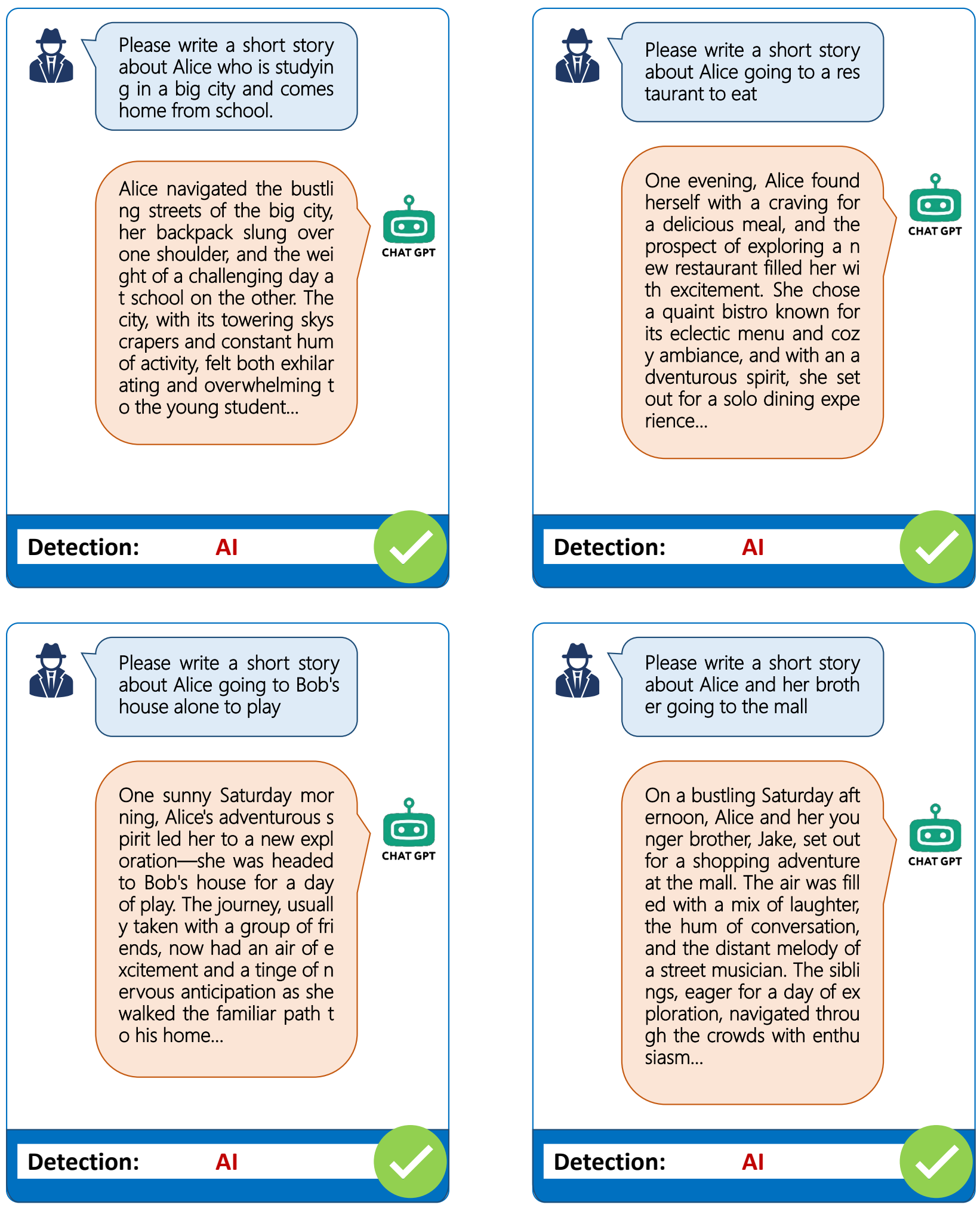

Figure 11: Examples of ChatGPT text detection on storytelling.

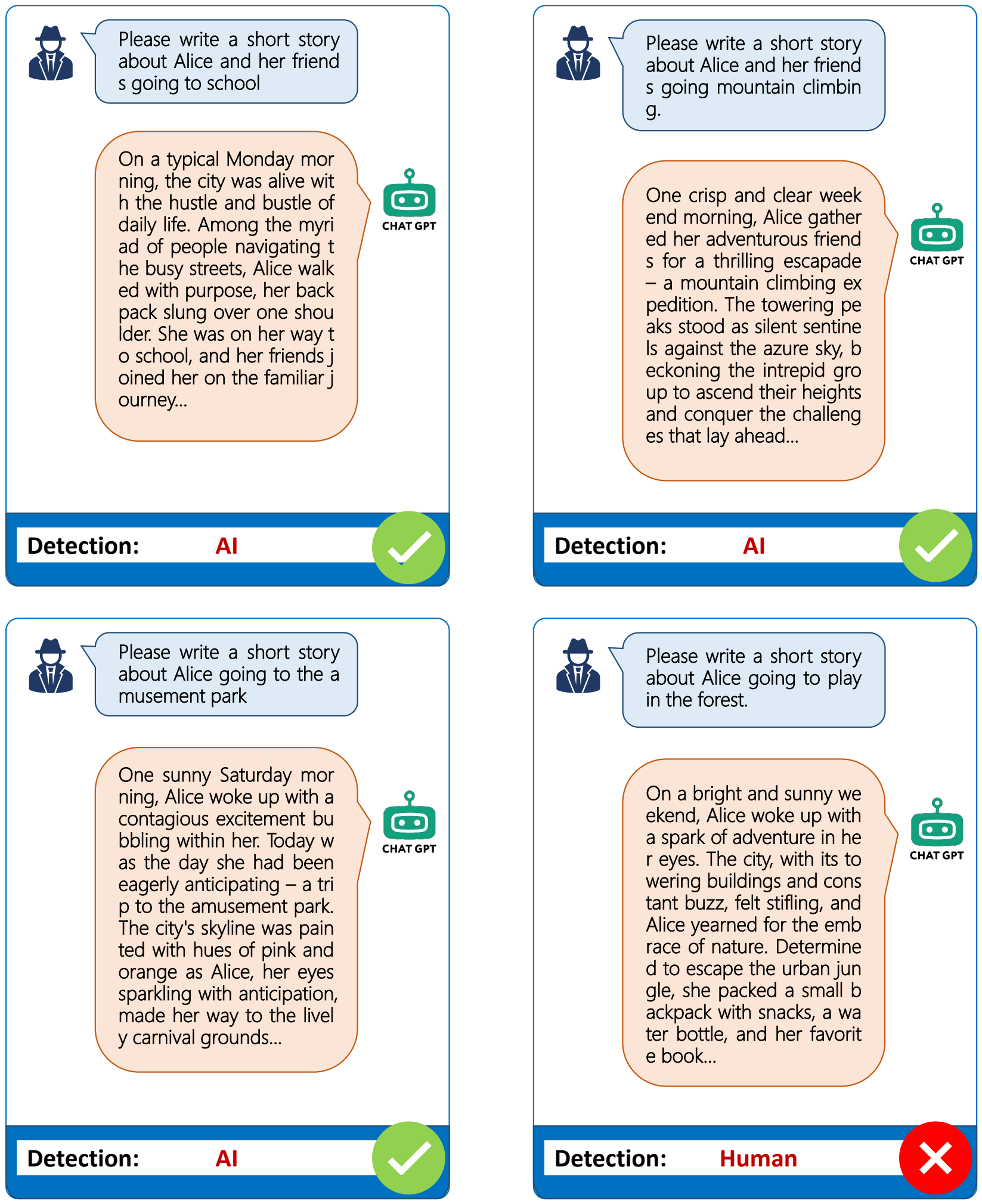

Figure 12: More examples of ChatGPT text detection on storytelling.

## M DISCUSSIONS AND FUTURE DIRECTIONS

Although we have empirically demonstrated that our MMD-MP has a lower variance of MMD value than the MMD-D method and provides a superior distributional discrepancy estimation when training on data from multiple populations, future research could explore the theoretical explanation of these findings to further justify its effectiveness. Additionally, our findings suggest that even when the kernel is trained on single-population data, our MMD-MP still outperforms MMD-D in terms of MGT detection performance. This observation motivates further investigation into the impact of the variance of data for training deep kernel-based MMD.

Furthermore, in the context of paragraph-based detection, we have not yet considered the inherent issue of *not independent and identically distributed* (Non-IID) text data in paragraph, which could break a basic assumption of the MMD tests (Grosse et al., 2017; Carlini & Wagner, 2017). The presence of data dependence within the observations can make it appear that two datasets from the same distribution are tested as different, rendering the test meaningless (Chwialkowski et al., 2014; Gao et al., 2021). One potential solution to this issue is using a wild bootstrap technique (Leucht & Neumann, 2013; Chwialkowski et al., 2014) to resample the MMD value during testing. This approach has already demonstrated success in adversarial detection (Gao et al., 2021), where adversarial examples are probably Non-IID since they are always generated with a pre-trained classifier.

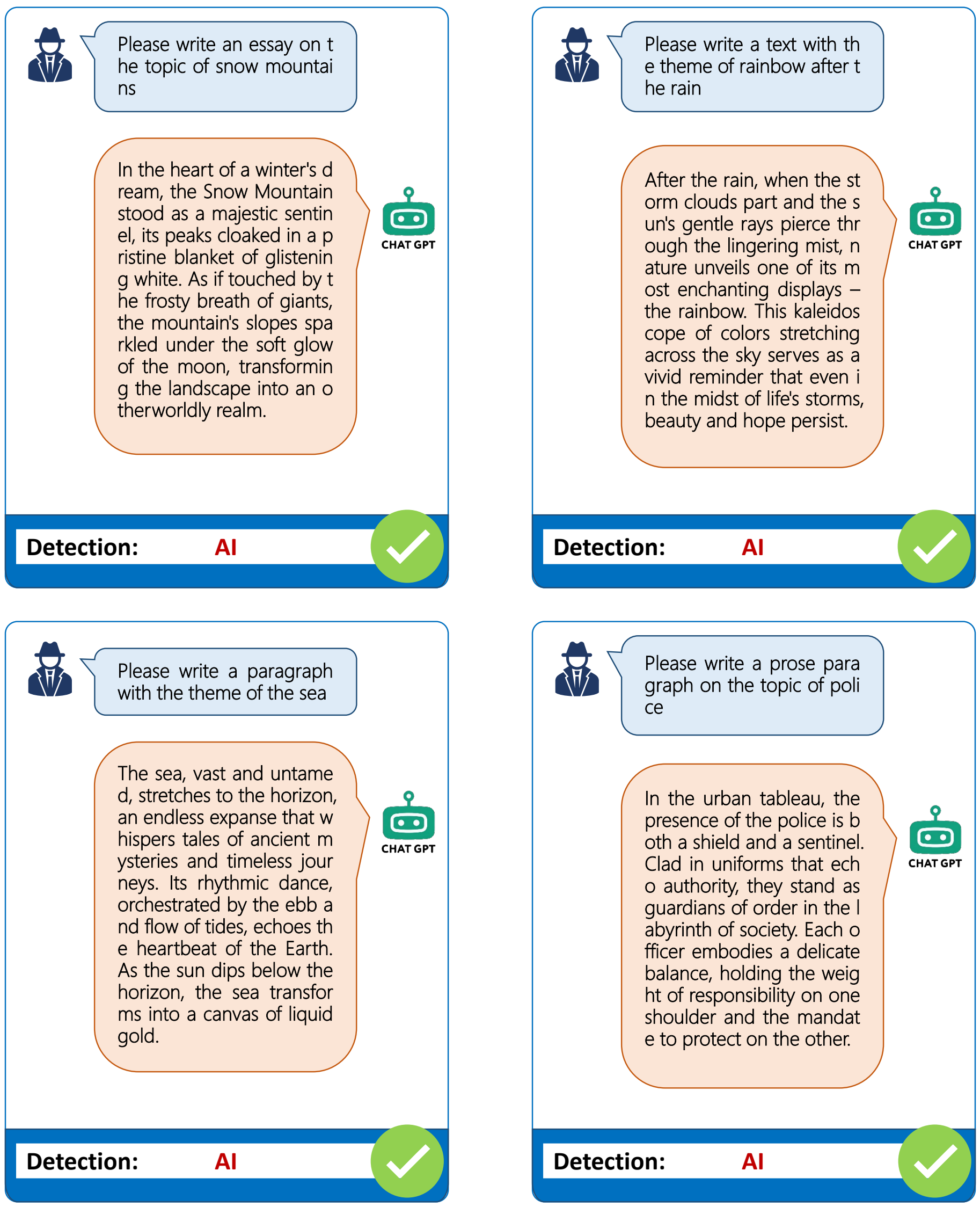

Figure 13: Examples of ChatGPT text detection on article writing.

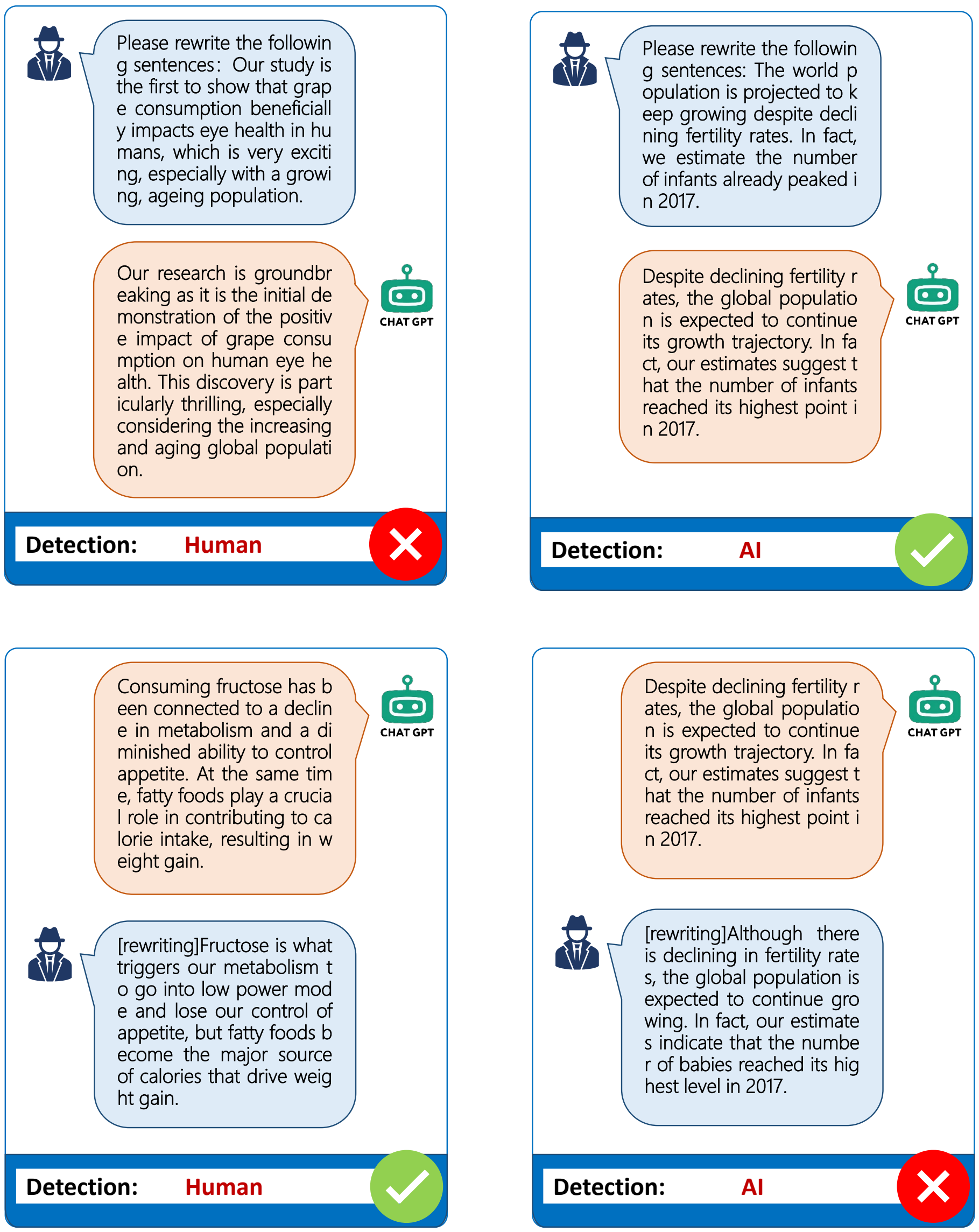

Figure 14: Examples of ChatGPT text detection on text rewriting.