**What Causes Optimization Dilemma of Kernel-based MMD?** Although kernel-based MMD is widely used to identify distributional discrepancies, limited literature has explored its optimization mechanism, possibly due to its complex mathematical format. For instance, when maximizing $\hat{J}$ in Eqn. (2), it is challenging to determine the individual changes in the MMD value and its variance, resulting in intricate analyses of each term in MMD. Yet, we still strive to observe some phenomena of these values through empirical studies to inspire our methods. To this end, we demonstrate some trends of the terms in MMD during training in Figure 2.

**Components in MMD and its variance.** For simplicity, we focus on three terms in MMD: $k_\omega(\mathbf{x}, \mathbf{x}')$, $k_\omega(\mathbf{x}, \mathbf{y})$ and $k_\omega(\mathbf{y}, \mathbf{y}')$, which are represented as kxx, kxy and kyy in Figure 2, respectively. We denote $H^* = k_\omega(\mathbf{x}, \mathbf{x}') - k_\omega(\mathbf{x}, \mathbf{y}') - k_\omega(\mathbf{y}', \mathbf{x})$. Then we decompose the variance of MMD into: $\mathrm{Var}(\mathbb{E}[H^* + k_\omega(\mathbf{y}, \mathbf{y}')]) = \mathrm{Var}(\mathbb{E}[H^*]) + \mathrm{Var}(\mathbb{E}[k_\omega(\mathbf{y}, \mathbf{y}')]) + 2\mathrm{Cov}(\mathbb{E}[H^*], \mathbb{E}[k_\omega(\mathbf{y}, \mathbf{y}')])$ and $\mathrm{Var}(\mathbb{E}[H^*]) = \mathrm{Var}(\mathbb{E}[k_\omega(\mathbf{x}, \mathbf{x}')]) - 2\mathrm{Cov}(\mathbb{E}[k_\omega(\mathbf{x}, \mathbf{x}')], \mathbb{E}[2k_\omega(\mathbf{x}, \mathbf{y})]) + \mathrm{Var}(\mathbb{E}[2k_\omega(\mathbf{x}, \mathbf{y})])$. Here, we use the notion of expectation to denote taking expectations across two populations sampled from MGTs and HWTs and the variance denotes taking variances within these sampled populations.

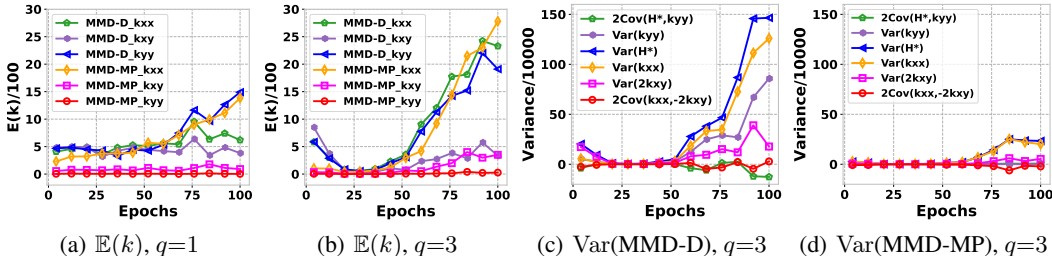

(a) $\mathbb{E}(k)$, $q$=1    (b) $\mathbb{E}(k)$, $q$=3    (c) Var(MMD-D), $q$=3    (d) Var(MMD-MP), $q$=3

Figure 2: $\mathbb{E}(k)$ in MMD and their variances under two optimization methods (MMD-MP is ours). Subfigures (a) and (b) depict the value of each $\mathbb{E}(k)$ in MMD during training by MMD-D and MMD-MP with $q$=1 and $q$=3, respectively. Subfigures (c) and (d) illustrate the variances of some terms associated with MMD, *i.e.*, $\sigma_{\mathfrak{H}_1}^2$ when training by MMD-D and MMD-MP, respectively.

### 2.3 OPTIMIZATION DILEMMA OF KERNEL-BASED MMD

We illustrate some critical observations related to the high variance problem in kernel-based MMD for MGT detection according to Figure 2, followed by explaining these phenomena.

**i)** From Figures 2 (a)-(b), both $\mathbb{E}[k_\omega(\mathbf{x}, \mathbf{x}')]$ and $\mathbb{E}[k_\omega(\mathbf{y}, \mathbf{y}')]$ exhibit a generally increasing trend, while $\mathbb{E}[k_\omega(\mathbf{x}, \mathbf{y})]$ shows relatively minor changes. **ii)** As the number of populations $q$ increases, $\mathbb{E}[k_\omega(\mathbf{y}, \mathbf{y}')]$ becomes smaller than $\mathbb{E}[k_\omega(\mathbf{x}, \mathbf{x}')]$, and the gap between them widens. **iii)** In Figure 2 (c), the variance of MMD is mainly determined by the variances of $\mathbb{E}[k_\omega(\mathbf{x}, \mathbf{x}')]$ and $\mathbb{E}[k_\omega(\mathbf{y}, \mathbf{y}')]$, while the impact of other terms is relatively minor. **iv)** When $S_{\mathbb{Q}}^{tr}$ comprises multiple distinct populations (*e.g.*, $q = 3$ in Figure 2 (c)), $\mathbb{E}[k_\omega(\mathbf{y}, \mathbf{y}')]$ optimized by MMD-D has a significant variance, as well as $\mathbb{E}[k_\omega(\mathbf{x}, \mathbf{x}')]$, which is consistent with the results of different $q$ in Appendix G.

**First**, **i)**, **ii)** indicate that as $q$ increases, when optimizing both $k_\omega(\mathbf{x}, \mathbf{x}')$ and $k_\omega(\mathbf{y}, \mathbf{y}')$ simultaneously in a Gaussian kernel (Bilmes et al., 1998), optimizing the aggregation of instances in $S_{\mathbb{Q}}^{tr}$-related terms becomes more challenging compared to $S_{\mathbb{P}}^{tr}$. **Second**, the possible explanation for the observations **iii)** **iv)** is that the optimization objective influences the optimization of different terms (*e.g.*, $S_{\mathbb{P}}^{tr}$ and $S_{\mathbb{Q}}^{tr}$) in the same way. Thus, the characteristics of their distributions after mapping by the same kernel function, such as the mean and variance of $k_\omega$, exhibit similar changing trends.

Furthermore, the optimized kernel function maximizing $\mathbb{E}[k_\omega(\mathbf{y}, \mathbf{y}')]$ will not only a) map the pairwise instances in $S_{\mathbb{Q}}^{tr}$ close to each other, making the mapped instances *more uniform*; but also b) enforce implicit *"pairing rules"* for aggregating instances in $S_{\mathbb{Q}}^{tr}$. These rules are shared to pair instances of $S_{\mathbb{P}}^{tr}$ throughout optimization. When $S_{\mathbb{Q}}^{tr}$ comprises different populations, the differences in $S_{\mathbb{Q}}^{tr}$-pairs might be large. Applying the paring rules may inadvertently map $S_{\mathbb{P}}^{tr}$ instances far from their class center, leading to increased fluctuations of $k_\omega(\mathbf{x}, \mathbf{x}')$ and thus larger $\mathrm{Var}(\mathbb{E}[k_\omega(\mathbf{x}, \mathbf{x}')])$. Similarly, optimizing $S_{\mathbb{P}}^{tr}$ negatively affects $S_{\mathbb{Q}}^{tr}$ but to a lesser extent because $S_{\mathbb{P}}$ is *IID*, meaning $S_{\mathbb{P}}$ instances share more similar statistical characteristics. Pairing rules for $S_{\mathbb{P}}^{tr}$ instances do not need to be as strong as those for aggregating non-*IID* $S_{\mathbb{Q}}^{tr}$ instances. Therefore, we will exclude $\mathbb{E}[k_\omega(\mathbf{y}, \mathbf{y}')]$ related term throughout optimization. We also explore the performance when replacing $k_\omega(X, X')$ with $k_\omega(Y, Y')$ as discussed in Appendix F.

## 3 PROPOSED METHODS

### 3.1 PROBLEM DEFINITION

In this paper, we consider detecting machine-generated texts (MGTs), where the testing texts can be from multiple different large language models (LLMs). The problem is formally defined as follows:

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

$. Last, the training algorithm is shown in Algorithm 1. With the trained kernel $k_{\omega}$, we design two approaches for MGT detection in Section 3.3.

We now study the uniform convergence of our proposed optimization function as follows.

**Theorem 1.** *(Uniform bound of MMD-MP) Let $\omega$ parameterize uniformly bounded kernel functions $k_{\omega}$ in a Banach space of dimension $D$ with $\|\omega\| \leq R_{\Omega}$, $k_{\omega}$ be uniformly bounded by $\sup_{\omega \in \Omega} \sup_{\mathbf{x}, \mathbf{x}' \in \mathcal{X}} k_{\omega}(\mathbf{x}, \mathbf{x}') \leq \nu$ with $L_k$-Lipschitz, i.e., $|k_{\omega}(\mathbf{x}, \mathbf{x}') - k_{\omega'}(\mathbf{x}, \mathbf{x}')| \leq L_k \|\omega - \omega'\|$. Let $\bar{\Omega}_s$ be a set of $\omega$ for which $\sigma_{\mathfrak{H}_1^*}^2 \geq s^2 > 0$. Taking $\lambda = n^{-1/3}$, with probability at least $1 - \delta$, we have*

$$\sup_{\omega \in \bar{\Omega}_s} \|\hat{J}(S_{\mathbb{P}}, S_{\mathbb{Q}}; k_{\omega}) - J(\mathbb{P}, \mathbb{Q}; k_{\omega})\| = \mathcal{O}\left( \frac{\nu}{s^2 n^{1/3}} \left[ \nu^2 \sqrt{D \log(R_{\Omega} n) + \log \frac{1}{\delta}} + \nu L_k + \frac{1}{s} \right] \right).$$

Detailed constants and proofs are given in Appendix A.3. Theorem 1 shows that our estimator $\hat{J}(S_{\mathbb{P}}, S_{\mathbb{Q}}; k_{\omega})$ converges uniformly over a ball in parameter space as $n$ increases. With enough training data, the estimator converges the optimal solution if the best kernel exists.

### 3.3 EXPLORING MMD-MP FOR MGT DETECTIONS

We consider MGT detection in two scenarios: paragraph-based detection and sentence-based detection. The former aims to detect whether the test paragraph follows the distribution of human-written paragraphs. We address this as a two-sample test. The latter focuses on distinguishing one single machine-generated sentence from HWTs. We consider this as a single-instance detection task.

**MGT Detection Under two-sample test (2ST).** For paragraph-based detection, we consider each sentence within the paragraph as an instance. The detailed procedure for 2ST using MMD-MP can be found in Algorithm 2. Note that we only optimize the kernel using MMD-MP during training and employ the MMD instead of the MPP to measure the distance between $S_{\mathbb{P}}$ and $S_{\mathbb{Q}}$ during testing. The rationale behind this is that we prefer the distance between $S_{\mathbb{P}}$ and $S_{\mathbb{Q}}$ to be zero when $\mathbb{P} = \mathbb{Q}$, rather than a negative value, *i.e.*, $-R(S_{\mathbb{Q}}) < 0$. Empirically, the performance of these two strategies is almost identical. We defer more discussion in Appendix E.

**MGT detection under single-instance detection (SID).** While paragraph-based detection is widely employed, there exist practical applications that require single-sentence detection, *e.g.*, online content filtering or false information recognition. Despite numerous works have shown MMD as a powerful means to measure the discrepancy between two distributions or populations, we still hope it can be employed to single-instance detection due to the powerful deep kernel. To achieve this, with a trained kernel, we calculate the distance between a set of referenced HWTs $S_{\mathbb{P}}^{re}$ and a test text $\tilde{\mathbf{y}}$ with Eqn. (10). The detailed procedure for SID using MMD-MP is shown in Algorithm 3.

$$\widehat{\mathrm{MMD}}_b^2 (S_{\mathbb{P}}^{re}, \{\tilde{\mathbf{y}}\}; k_{\omega}) = \frac{1}{n^2} \sum_{i,j=1}^{n} k_{\omega}(\mathbf{x}_i, \mathbf{x}_j) - \frac{2}{n} \sum_{i=1}^{n} k_{\omega}(\mathbf{x}_i, \tilde{\mathbf{y}})) + k_{\omega}(\tilde{\mathbf{y}}, \tilde{\mathbf{y}}). \qquad (10)$$

**Advantages of MMD-MP for MGT Detection over MMD-D.** We summarize the advantages of MMD-MP over MMD-D as two parts: 1) **More stable discrepancy estimation**: Although MMD-D has a similar $\mathbb{E}[k(\mathbf{x}, \mathbf{x})]$ with MMD-MP, its variance is much greater than MMD-MP (see Figures 2 (a), (c) and (d)), indicating that MMD-D exhibits poorer aggregation effects for $S_{\mathbb{P}}^{tr}$ compared with MMD-MP. Moreover, training MMD using MMD-MP results in a significantly lower variance (see Figures 2 (c) and (d)), which addresses the issue of high variance in MMD optimization, thus enhancing the stability of discrepancy estimation. 2) **Enhanced transferability**: Our MMD-MP focuses more on fitting HWTs $S_{\mathbb{P}}^{tr}$ compared with MMD-D when training the deep kernel, which reduces the reliance of

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

 | $62.83_{\pm0.90}$ | $43.64_{\pm5.92}$ | $30.68_{\pm2.37}$ | $34.62_{\pm2.73}$ | $46.66_{\pm2.95}$ |
| C2ST-L | $89.82_{\pm1.02}$ | $75.74_{\pm4.90}$ | $60.97_{\pm1.87}$ | $68.50_{\pm1.81}$ | $78.22_{\pm3.12}$ |
| MMD-O | $26.43_{\pm1.40}$ | $21.17_{\pm3.12}$ | $19.83_{\pm2.81}$ | $25.23_{\pm0.47}$ | $25.18_{\pm1.41}$ |
| MMD-D | $91.76_{\pm1.58}$ | $86.98_{\pm2.53}$ | $73.62_{\pm3.03}$ | $86.44_{\pm1.07}$ | $91.46_{\pm0.47}$ |
| MMD-MP (Ours) | $\mathbf{93.21}_{\pm1.35}$ | $\mathbf{89.36}_{\pm2.91}$ | $\mathbf{79.68}_{\pm2.42}$ | $\mathbf{89.63}_{\pm1.94}$ | $\mathbf{91.96}_{\pm0.62}$ |

Table 2: Test power/100 on HC3 given 1, 000 processed paragraphs in training data.

| Method | ChatGPT | GPT3-S | Neo-S | ChatGPT Neo-S | ChatGPT GPT3-S | ChatGPT GPT2-S GPT2-M | ChatGPT Neo-S GPT3-S | ChatGPT Neo-S Neo-L |
|---|---|---|---|---|---|---|---|---|
| C2ST-S | $60.32_{\pm2.56}$ | $38.06_{\pm4.49}$ | $27.65_{\pm2.34}$ | $34.48_{\pm3.70}$ | $40.89_{\pm3.79}$ | $24.97_{\pm2.05}$ | $32.04_{\pm2.41}$ | $24.53_{\pm3.08}$ |
| C2ST-L | $87.81_{\pm1.48}$ | $74.29_{\pm4.16}$ | $61.05_{\pm3.35}$ | $67.47_{\pm3.17}$ | $75.49_{\pm2.21}$ | $56.24_{\pm3.53}$ | $67.10_{\pm2.69}$ | $54.91_{\pm3.24}$ |
| MMD-O | $27.23_{\pm3.53}$ | $19.96_{\pm5.03}$ | $19.58_{\pm2.02}$ | $27.34_{\pm1.42}$ | $26.03_{\pm1.63}$ | $20.05_{\pm2.86}$ | $23.91_{\pm0.92}$ | $20.92_{\pm1.10}$ |
| MMD-D | $91.38_{\pm2.09}$ | $84.01_{\pm5.04}$ | $72.81_{\pm3.23}$ | $74.22_{\pm4.06}$ | $83.29_{\pm3.05}$ | $62.34_{\pm4.00}$ | $77.76_{\pm2.93}$ | $63.15_{\pm2.38}$ |
| MMD-MP (Ours) | $\mathbf{92.31}_{\pm2.30}$ | $\mathbf{86.34}_{\pm5.37}$ | $\mathbf{76.35}_{\pm3.51}$ | $\mathbf{85.30}_{\pm1.99}$ | $\mathbf{89.05}_{\pm1.64}$ | $\mathbf{79.92}_{\pm3.88}$ | $\mathbf{85.54}_{\pm1.93}$ | $\mathbf{79.69}_{\pm0.78}$ |

## 4.2 TEST POWER ON PARAGRAPH-BASED DETECTION

We compare our MMD-MP with the state-of-the-art (SOTA) two-sample test method for detecting MGTs on HC3 in terms of test power and defer the results on XSum in Appendix I.1. To broadly evaluate detection performance, we conduct experiments on various scenarios, including training on full training data, a limited number of training data, and unbalanced training data.

**Test power on full training data.** We conduct this experiment using 3, 100 processed paragraphs to train the model. Table 1 shows the detection performance under different training populations in terms of test power compared with other baselines, including one and two populations. The results show that MMD-MP exhibits superior test power compared with other methods, particularly in detecting Neo-S texts, where the test power is approximately 6% ↑ higher than MMD-D.

**Test power on a limited number of training data.** We utilize 1, 000 processed paragraphs to train the models with one, two, and three training populations, respectively. Table 2 demonstrates that our method achieves significantly higher test power performance compared with others. Remarkably, our method outperforms MMD-D by an average of 8.20% ↑ on test power over the two training populations and exhibits a 13.97% ↑ increase over the three training populations, suggesting that extreme instability of discrepancy estimation of MMD-D when dealing with multiple populations.

**Test power on challenging unbalanced training data.** In real-world scenarios, obtaining HWTs is easily feasible, while collecting MGTs poses more challenges. To thoroughly assess the performance of our approach, we conduct an evaluation with 2, 000 processed HWT and 400 MGT training paragraphs. As illustrated in the top of Figure 8 (see Appendix), our approach exhibits significantly superior performance compared with other methods, *e.g.*, surpassing the test power of 6.96%∼14.40% ↑ than MMD-D, highlighting its stability in detecting MGTs under unbalanced training data scenarios.

## 4.3 AUROC ON SENTENCE-BASED DETECTION

In this section, we compare our MMD-MP with the SOTA single-instance detection method for detecting MGTs on HC3 in terms of AUROC and defer the results on XSum in Appendix I.2.

**AUROC on full training data.** Table 3 shows that our MMD-MP achieves better AUROC than other baselines. Notably, our MMD-MP outperforms the SOTA model-based method, *i.e.*, ChatGPT-D with 1.20% ↑ of AUROC when detecting ChatGPT texts. Moreover, MMD-MP achieves an improvement of 0.22%∼1.71% ↑ on AUROC over MMD-D and 0.61%∼2.64% ↑ over the CE-Classifier method, illustrating the superiority of our method in detecting the single sentence.

**AUROC on a limited number of training data.** From Table 4, our MMD-MP achieves performance comparable to CE-classifier in detecting ChatGPT texts and surpasses other baselines in other scenarios. Particularly, our method outperforms MMD-D by 2.46%↑ and CE-Classifier by 1.39%↑ on average. Note that although the model of CE-Classifier is the same as MMD-D and MMD-MP except for an additional classifier, our MMD-MP demonstrates superior detection performance over CE-Classifier, indicating the powerful distinguishability of our method.

Table 3: AUROC/100 on HC3 given 3, 100 processed paragraphs.

| Method | ChatGPT | GPT3-S | Neo-S | ChatGPT Neo-S | ChatGPT GPT3-S |
|---|---|---|---|---|---|
| Likelihood | $89.82_{\pm0.03}$ | $60.56_{\pm1.32}$ | $61.18_{\pm1.25}$ | $75.81_{\pm0.51}$ | $75.05_{\pm0.25}$ |
| Rank | $73.20_{\pm1.49}$ | $71.96_{\pm1.01}$ | $72.09_{\pm0.51}$ | $72.74_{\pm0.74}$ | $72.34_{\pm1.38}$ |
| Log-Rank | $89.58_{\pm0.07}$ | $63.78_{\pm1.29}$ | $64.92_{\pm1.04}$ | $77.57_{\pm0.55}$ | $76.47_{\pm0.12}$ |
| Entropy | $31.53_{\pm0.90}$ | $54.34_{\pm1.33}$ | $56.19_{\pm0.33}$ | $44.08_{\pm0.24}$ | $42.08_{\pm2.01}$ |
| DetectGPT-d | $77.92_{\pm0.74}$ | $53.41_{\pm0.41}$ | $52.07_{\pm0.38}$ | $66.01_{\pm0.29}$ | $65.70_{\pm1.14}$ |
| DetectGPT-z | $81.07_{\pm0.77}$ | $53.45_{\pm0.53}$ | $52.28_{\pm0.31}$ | $67.54_{\pm0.19}$ | $67.32_{\pm1.02}$ |
| OpenAI-D | $78.57_{\pm1.55}$ | $84.05_{\pm0.71}$ | $84.86_{\pm0.87}$ | $81.20_{\pm0.95}$ | $80.68_{\pm1.64}$ |
| ChatGPT-D | $95.64_{\pm0.13}$ | $61.89_{\pm1.04}$ | $54.45_{\pm0.10}$ | $75.47_{\pm0.63}$ | $78.95_{\pm1.00}$ |
| CE-Classifier | $96.19_{\pm0.17}$ | $92.44_{\pm0.63}$ | $88.88_{\pm0.19}$ | $90.93_{\pm0.72}$ | $92.97_{\pm0.28}$ |
| MMD-O | $56.34_{\pm0.66}$ | $59.90_{\pm0.87}$ | $63.19_{\pm0.76}$ | $60.46_{\pm1.28}$ | $57.79_{\pm1.25}$ |
| MMD-D | $95.83_{\pm0.37}$ | $94.86_{\pm0.48}$ | $88.40_{\pm1.28}$ | $91.39_{\pm0.86}$ | $93.49_{\pm0.46}$ |
| MMD-MP (Ours) | $\mathbf{96.20}_{\pm0.28}$ | $\mathbf{95.08}_{\pm0.32}$ | $\mathbf{92.04}_{\pm0.58}$ | $\mathbf{92.48}_{\pm0.37}$ | $\mathbf{94.61}_{\pm0.22}$ |

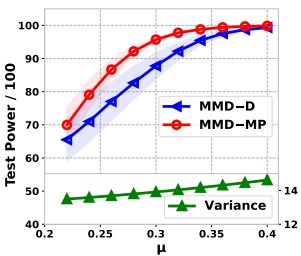

Figure 3: Impact of variance in training data on test power.

Table 4: AUROC/100 on HC3 given 1, 000 processed paragraphs in training data.

| Method | ChatGPT | GPT3-S | Neo-S | ChatGPT Neo-S | ChatGPT GPT3-S | ChatGPT GPT2-S GPT2-M | ChatGPT Neo-S GPT3-S | ChatGPT Neo-S Neo-L |
|---|---|---|---|---|---|---|---|---|
| CE-Classifier | $\mathbf{95.99}_{\pm0.40}$ | $91.40_{\pm0.37}$ | $87.27_{\pm0.52}$ | $88.13_{\pm0.44}$ | $91.59_{\pm0.36}$ | $84.89_{\pm0.61}$ | $88.91_{\pm0.51}$ | $84.15_{\pm1.30}$ |
| MMD-O | $54.64_{\pm1.69}$ | $61.52_{\pm1.18}$ | $61.93_{\pm2.22}$ | $58.28_{\pm1.65}$ | $57.92_{\pm1.32}$ | $57.86_{\pm1.39}$ | $59.78_{\pm0.61}$ | $58.07_{\pm1.20}$ |

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

 | $9.04_{\pm3.60}$ | $38.12_{\pm2.64}$ | $38.31_{\pm2.95}$ | $20.37_{\pm2.25}$ | $22.43_{\pm2.39}$ | $25.94_{\pm3.29}$ | $17.67_{\pm1.56}$ |
| C2ST-L | $32.64_{\pm4.84}$ | $73.64_{\pm2.03}$ | $76.07_{\pm1.61}$ | $54.51_{\pm1.76}$ | $54.85_{\pm2.46}$ | $61.95_{\pm2.47}$ | $50.62_{\pm1.23}$ |
| MMD-O | $8.36_{\pm1.61}$ | $16.26_{\pm2.59}$ | $18.76_{\pm1.33}$ | $12.80_{\pm1.57}$ | $13.25_{\pm2.23}$ | $14.25_{\pm1.79}$ | $10.51_{\pm1.35}$ |
| MMD-D | $39.99_{\pm3.78}$ | $73.58_{\pm4.41}$ | $76.87_{\pm1.63}$ | $53.88_{\pm2.82}$ | $57.33_{\pm2.53}$ | $63.43_{\pm2.10}$ | $55.06_{\pm2.74}$ |
| MMD-MP (Ours) | $\mathbf{45.51}_{\pm1.64}$ | $\mathbf{80.71}_{\pm1.29}$ | $\mathbf{81.95}_{\pm2.38}$ | $\mathbf{65.81}_{\pm3.16}$ | $\mathbf{64.27}_{\pm2.80}$ | $\mathbf{70.78}_{\pm1.79}$ | $\mathbf{64.80}_{\pm1.95}$ |

## I.2 AUROC ON SENTENCE-BASED DETECTION

In this part, we compare our MMD-MP with the SOTA single-instance detection method for detecting MGT on XSum in terms of AUROC given $1,000$ processed paragraphs in training data. From Table 17, our MMD-MP consistently achieves the highest AUROC performance compared with other methods. These results further demonstrate the powerful distinguishability of our proposed method when detecting MGTs.

Table 17: AUROC/100 on XSum given $1,000$ processed paragraphs in training data.

| Method | GPT2-M | GPT3-S | Neo-S | GPT2-M Neo-S | GPT2-M GPT3-S | GPT2-M Neo-S GPT3-S | GPT2-M Neo-S Neo-L |
|---|---|---|---|---|---|---|---|
| Likelihood | $61.93_{\pm1.37}$ | $45.97_{\pm0.62}$ | $43.33_{\pm1.36}$ | $52.74_{\pm1.07}$ | $54.46_{\pm0.95}$ | $50.41_{\pm0.75}$ | $53.33_{\pm0.72}$ |
| Rank | $72.11_{\pm0.65}$ | $64.81_{\pm1.10}$ | $63.93_{\pm1.34}$ | $68.18_{\pm0.82}$ | $68.72_{\pm0.60}$ | $66.73_{\pm0.63}$ | $66.72_{\pm0.48}$ |
| Log-Rank | $65.37_{\pm1.35}$ | $50.97_{\pm0.56}$ | $49.46_{\pm1.39}$ | $57.66_{\pm1.26}$ | $58.78_{\pm1.03}$ | $55.16_{\pm0.93}$ | $57.98_{\pm0.92}$ |
| Entropy | $54.92_{\pm1.51}$ | $64.34_{\pm1.00}$ | $69.85_{\pm1.63}$ | $61.93_{\pm1.11}$ | $59.75_{\pm1.14}$ | $63.47_{\pm0.25}$ | $61.11_{\pm1.15}$ |
| DetectGPT-d | $60.08_{\pm0.70}$ | $44.41_{\pm1.22}$ | $40.28_{\pm1.03}$ | $50.09_{\pm0.81}$ | $52.41_{\pm1.31}$ | $48.34_{\pm0.61}$ | $50.96_{\pm0.99}$ |
| DetectGPT-z | $61.39_{\pm0.70}$ | $44.45_{\pm1.20}$ | $40.61_{\pm1.07}$ | $50.97_{\pm0.84}$ | $53.05_{\pm1.17}$ | $48.82_{\pm0.57}$ | $51.90_{\pm0.94}$ |
| OpenAI-D | $75.11_{\pm0.57}$ | $84.56_{\pm0.71}$ | $86.69_{\pm0.12}$ | $80.74_{\pm0.91}$ | $79.91_{\pm1.02}$ | $82.36_{\pm1.07}$ | $76.88_{\pm0.46}$ |
| ChatGPT-D | $54.20_{\pm1.52}$ | $50.92_{\pm0.99}$ | $53.60_{\pm1.40}$ | $54.40_{\pm1.38}$ | $53.47_{\pm0.98}$ | $53.69_{\pm0.88}$ | $53.71_{\pm0.99}$ |
| CE-Classifier | $78.58_{\pm1.41}$ | $91.36_{\pm0.87}$ | $92.89_{\pm0.35}$ | $84.71_{\pm0.80}$ | $85.61_{\pm0.64}$ | $87.57_{\pm0.30}$ | $84.06_{\pm0.43}$ |
| MMD-O | $60.29_{\pm0.66}$ | $65.62_{\pm0.92}$ | $67.56_{\pm0.98}$ | $62.01_{\pm0.75}$ | $64.10_{\pm1.52}$ | $64.46_{\pm0.97}$ | $63.14_{\pm1.10}$ |
| MMD-D | $72.37_{\pm1.16}$ | $91.48_{\pm0.39}$ | $90.01_{\pm0.82}$ | $82.73_{\pm0.73}$ | $87.59_{\pm0.80}$ | $86.54_{\pm0.92}$ | $82.78_{\pm0.68}$ |
| MMD-MP (Ours) | $\mathbf{82.16}_{\pm0.79}$ | $\mathbf{91.73}_{\pm0.41}$ | $\mathbf{93.70}_{\pm0.45}$ | $\mathbf{86.46}_{\pm0.60}$ | $\mathbf{87.79}_{\pm0.77}$ | $\mathbf{88.27}_{\pm0.71}$ | $\mathbf{85.70}_{\pm0.54}$ |

## I.3 RESULTS ON CHALLENGING UNBALANCED TRAINING DATA

In this part, we evaluate our method on unbalanced training data containing $2,000$ HWT and $400$ MGT training paragraphs over XSum. In Figure 9, we observe that our approach consistently outperforms baselines in terms of test power and AUROC. Critically, our MMD-MP surpasses the test power of $4.85\%\sim9.84\% \uparrow$ than MMD-D, highlighting the stability of our method in detecting MGTs under unbalanced training data scenarios.

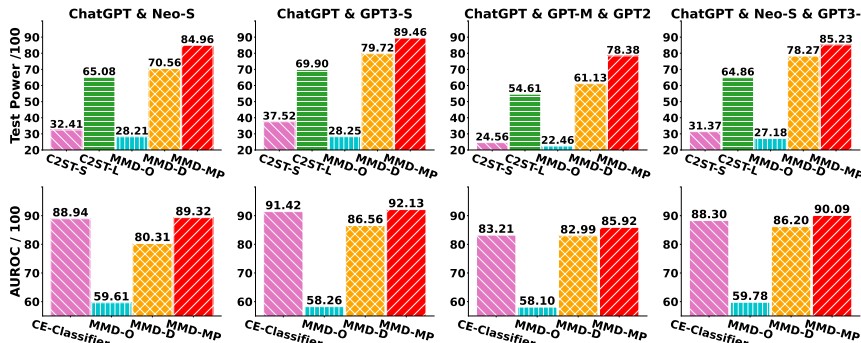

Figure 8: Test power and AUROC on HC3 given $2,000$ HWT and $400$ MGT training paragraphs.

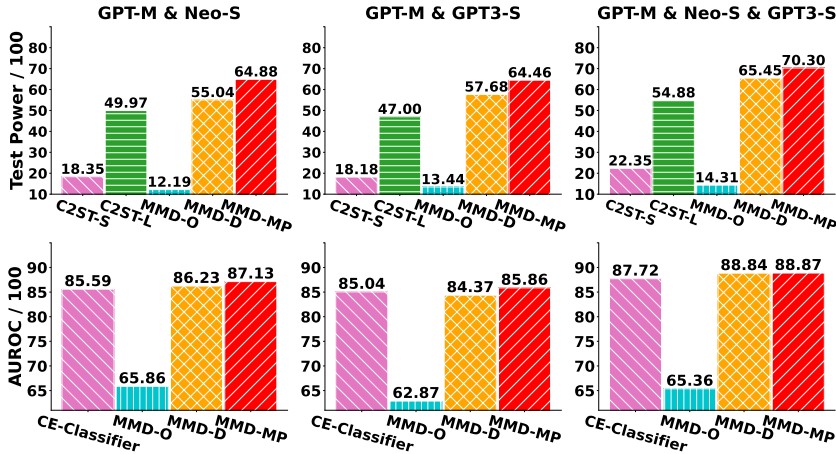

Figure 9: Test power and AUROC on XSum given $2,000$ HWT and $400$ MGT training paragraphs.

## J DISCUSSIONS AND FUTURE DIRECTIONS

Although we have empirically demonstrated that our MMD-MP has a lower variance of MMD value than the MMD-D method and provides a superior distributional discrepancy estimation when training on data from multiple populations, future research could explore the theoretical explanation of these findings to further justify its effectiveness. Additionally, our findings suggest that even when the kernel is trained on single-population data, our MMD-MP still outperforms MMD-D in terms of MGT detection performance. This observation motivates further investigation into the impact of the variance of data for training deep kernel-based MMD.

Furthermore, in the context of paragraph-based detection, we have not yet considered the inherent issue of *not independent and identically distributed* (Non-IID) text data in paragraph, which could break a basic assumption of the MMD tests (Grosse et al., 2017; Carlini & Wagner, 2017). The presence of data dependence within the observations can make it appear that two datasets from the same distribution are tested as different, rendering the test meaningless (Chwialkowski et al., 2014; Gao et al., 2021). One potential solution to this issue is using a wild bootstrap technique (Leucht & Neumann, 2013; Chwialkowski et al., 2014) to resample the MMD value during testing. This approach has already demonstrated success in adversarial detection (Gao et al., 2021), where adversarial examples are probably Non-IID since they are always generated with a pre-trained classifier.