# OpenReview forum: "Detecting Machine-Generated Texts by Multi-Population Aware Optimization for Maximum Mean Discrepancy"
_ICLR.cc/2024/Conference — ICLR 2024 poster_

### Official Review · Reviewer_4WLs · 2023-10-30

**Soundness:** 3 good
**Presentation:** 4 excellent
**Contribution:** 3 good
**Rating:** 6
**Confidence:** 4

**Summary:**

The paper delves into the ever-growing challenge of differentiating machine-generated texts (MGTs) from human-written texts due to the increasingly sophisticated capabilities of large language models like ChatGPT. The authors highlight the potential risks MGTs pose, especially in scenarios where accuracy and authenticity are crucial. Their primary approach is based on the exploitation of maximum mean discrepancy (MMD) to discern the subtle distributional discrepancies between MGTs and human texts.
However, a pivotal point of contention is the assertion that directly training a detector using MMD on diverse MGTs leads to a spike in MMD variance. This is due to the potential multiplicity of text populations from various LLMs, which the authors argue could compromise the efficacy of MMD in differentiating the two sample types. In response, they introduce the concept of MMD-MP, a multi-population aware optimization method. This novel solution is said to counteract the variance increases, leading to a more consistent and reliable detection mechanism.
Two distinct methods, based on paragraph and sentence structures, were developed relying on MMD-MP. The authors have furnished empirical evidence, demonstrating the superiority of their approach when tested across a range of LLMs.
While the research appears thorough and the problem well-identified, certain areas raise eyebrows. For instance, the premise that MMD variance surges significantly with diverse MGTs warrants further investigation. Additionally, the paper's claim of consistently outperforming all existing baselines across a multitude of LLMs might be viewed with a touch of skepticism until validated by external sources. Nevertheless, this paper's contributions to the realm of MGT detection are commendable and merit attention.

**Strengths:**

1. Innovative Approach to a Contemporary Challenge: The paper addresses the pressing and contemporary issue of distinguishing machine-generated texts (MGTs) from human-written texts, especially given the sophisticated capabilities of modern LLMs. The introduction of the multi-population aware optimization method, MMD-MP, is a novel approach that seeks to enhance the accuracy and stability of MGT detection, providing a fresh perspective on the problem.

2. The authors haven't limited their research to just theoretical propositions; they have put their methods to the test. They conducted extensive experiments across various LLMs, providing empirical evidence of the superiority of their approach. Such thorough testing not only validates their methods but also offers a benchmark for future research in this domain.

3. The paper offers clear theoretical insights, particularly the exploration of the optimization mechanism of MMD and its associated challenges. Propositions and corollaries, such as the asymptotics of MPP and test power, further underscore the research's depth and provide a strong theoretical foundation for their method. The use of statistical properties and theoretical analysis makes the paper robust and comprehensive.

**Weaknesses:**

1. The paper relies heavily on mathematical formulations, such as the objective function
$$ J(P, Q; k_u) = MPP(P, Q; k_u)/σ(P, Q; k_u) $$
and the estimator equation
$ MPP_u(S_p, S_q; k_u) = \frac{1}{n(n − 1)} \sum H^{ij} $,
where  $ H^{ij}=k_u(x_i, x_j) − k_u(x_i, y_j) − k_u(y_i, x_j) $.
While these equations are crucial for understanding the paper's methodology, they can be intimidating and might make the content less accessible to readers who aren't well-versed in mathematical notation or kernel-based methodologies.

2. The paper introduces MMD-MP to handle the variance issue in MMD when dealing with multiple text populations. However, the mathematical representation
$ σ^2_{s} = 4 (E[H^2_{i3}] − E[H^2_{i2}] ) $,
where
$ H^2_{i3}, H^2_{i2} $
denote different $ H^{ij} $, suggests that the variance calculation is still inherently tied to the distance metrics. This could lead to concerns about the real-world applicability and stability of the proposed solution, especially in scenarios with vast and diverse text samples.

3. The algorithms presented, such as the training deep kernel with MMD-MP (Algorithm 1) and testing with MMD-MP for 2ST (Algorithm 2), are specifically tailored to address the MGT detection problem. Given the intricate mathematical formulations, it may be challenging to adapt or generalize these algorithms to other domains or problems without significant modifications. For researchers or practitioners looking for broader applications, this specificity might be a limitation.

**Questions:**

Sure, based on the weaknesses and the content of the image, here are two questions:

1. Given the intricate mathematical representations like
$ J(P, Q; k_u) = MPP(P, Q; k_u)/σ(P, Q; k_u) $
and
$ MPP_u(S_p, S_q; k_u) = \frac{1}{n(n − 1)} \sum H^{ij} $,
how do you envision making the paper's methodologies more accessible to a wider audience, especially those who might not be well-acquainted with kernel-based techniques?

2.  The paper proposes MMD-MP to address the variance issue in MMD, represented by
$ σ^2_{s} = 4 (E[H^2_{i3}] − E[H^2_{i2}] ) $.
How confident are the authors in the robustness and stability of this method when applied to real-world scenarios with vast and diverse text samples? Furthermore, how can the proposed algorithms be adapted to other domains or challenges beyond MGT detection?

---

> ### Author Response · Authors · 2023-11-20
> **Responses to Reviewer 4WLs [1/2]**
>
> We thank the reviewer for the encouraging comments and detailed suggestions. Responses are below:
>
> >Q1. "MMD-MP,..., is a novel approach, providing **a fresh perspective** on the problem", "Such **thorough testing** not only validates their methods but also **offers a benchmark for future research** in this domain", "...**clear theoretical insights**,...**further underscore the research’s depth** and provide **a strong theoretical foundation**", "The use of statistical properties and theoretical analysis makes the paper **robust and comprehensive**".
>
> **A1.** We sincerely appreciate your high praise and encouraging comments on our work. The acknowledgment of MMD-MP as offering a fresh perspective and the recognition of our comprehensive testing and clear theoretical insights are deeply appreciated. The comments about the robustness and potential impact of our paper as a benchmark for future research inspire us to further advance our contributions in this domain.
>
> >Q2. The paper relies heavily on mathematical formulations, such as the objective function $J(P, Q; k_u) = MPP(P, Q; k_u)/σ(P, Q; k_u)$ and the estimator equation $MPP_u(S_p, S_q; k_u) = \frac{1}{n(n − 1)} \sum H^{ij}$, where $H^{ij}=k_u(x_i, x_j) − k_u(x_i, y_j) − k_u(y_i, x_j)$. While these equations are crucial for understanding the paper's methodology, they can be intimidating and might make the content less accessible to readers who aren't well-versed in mathematical notation or kernel-based methodologies. How do you envision making the paper’s methodologies more accessible to a wider audience, especially those who might not be well-acquainted with kernel-based techniques?
>
> **A2.** Thank you for your insightful suggestions on improving the readability of our paper for broader audiences. Our paper indeed requires some relatively complex mathematics to develop the core approach. Nevertheless, this may pose some difficulties  for general  readers in understanding paper. To amend this, we have carefully revised paper (e.g., adding more intuitions in Section 2 and Appendix) to make the paper easier to understand.
> Specifically, we have made the following revisions:
>
> * **More intuitive MMD Explanations:** We have added intuitive explanations for MMD's distance, e.g., viewing $k(X,X′)$ or $k(Y,Y′)$ as an intra-class distance and $k(X,Y)$ as an inter-class distance in Section 2.1, aiding for understanding its definition.
> * **More intuitive explanations for the optimization of kernel-based MMD:** For MMD-D optimization, we highlight that it intuitively tends to separately aggregate human-written texts (HWTs) and all possible different-population machine-generated texts (MGTs) like decreasing intra-class distance, and simultaneously push them away from each other like increasing inter-class distance. This presents challenges due to significant fluctuations or higher MMD variance. In contrast, our MMD-MP relaxes constraints on aggregating all MGTs populations (i.e., removing the $k(Y,Y′)$ from MMD), focusing more on fitting HWTs. This stabilizes MMD values and enhances the transferability of detection. We have included more illustrative figures to depict this procedure in Section D.
> * **Extensive intuitive observations with detailed analyses：** We have provided extensive intuitive explanations in Section 2.2, 2.3 and Appendix D, G, H, offering insights into the motivation through empirical observations in figures. Moreover, we have carefully provided more detailed analyses to support these observations, aiding readers in comprehending the core of our method.
> * **More detailed algorithms presentation:** We have provided a more detailed algorithmic presentation in Section C.2, offering clear steps for Algorithms 1, 2, and 3. This ensures readers have a lucid pathway to understand our method's workflow.
> * **Clearer objective function construction:** At the beginning of Section 3.2 of our method, we have introduced the optimization objective function $J$ and followed it with theoretical derivations. This enables readers to directly observe the optimization objective function without delving into the followed derivation process, guiding even those finding statistical theory challenging. We have provided a clearer description of this.
>
> Our goal is to strike a balance between maintaining the rigor of our theoretical analyses and providing enough assistance for readers to understand the key aspects of our method.

---

> ### Author Response · Authors · 2023-11-20
> **Responses to Reviewer 4WLs [2/2]**
>
> >Q3. The paper introduces MMD-MP to handle the variance issue in MMD when dealing with multiple text populations. However, the mathematical representation $σ^2_{s} = 4 (E[H^2_{i3}] − E[H^2_{i2}] )$, where $H^2_{i3}, H^2_{i2}$ denote different $H^{ij}$, suggests that the variance calculation is still inherently tied to the distance metrics. This could lead to concerns about the real-world applicability and stability of the proposed solution, especially in scenarios with vast and diverse text samples. How confident are the authors in the robustness and stability of this method when applied to real-world scenarios with vast and diverse text samples?
>
> **A3.** We appreciate your insightful comments on the robustness and stability of our method. To address your concern, we would like to emphasize that the use of distance metrics in our method brings the following advantages.
>
> * **Better transferability to diverse texts:**  Our method is able to adapt for detecting diverse types of texts. Empirical studies across different text domains demonstrate the superior transferability of our method. Specifically, we train the deep kernel using texts generated by ChatGPT, GPT2 and GPT2-m, and test on texts generated by other LLMs like GPT-Neo-L, GPT-j-6b and GPT4all-j. The results in Tables 5 and 6 in our paper show our method's effectiveness across different domains compared with other baselines, indicating better adaptation.
> * **Reduced dependency on specific features and enhanced robustness:** Relying on distance metrics, our method may reduce the dependency on specific features or representations, contributing to its ability to generalize to different text domains. Moreover,  metric-based detectors, as validated in [r1, r2, r3], have demonstrated robustness against adversarial attacks, extending the applicability of our method to some safety-critical systems.
>
>     [r1] Detecting Adversarial Data by Probing Multiple Perturbations Using Expected Perturbation Score, ICML 2023.
>
>     [r2] Maximum mean discrepancy test is aware of adversarial attack, ICML 2021.
>
>     [r3] The "Beatrix'' Resurrections: Robust Backdoor Detection via Gram Matrices, NDSS 2023.
>
> * **Effective detection on diverse scenarios:** We conducted experiments on two diverse datasets, HC3 and XSum. The HC3 dataset comprises 24,322 sentences or paragraphs of varying lengths, while the XSum dataset consists of 226,711 news articles. These two datasets **encompass diverse practical language domains such as finance, healthcare, Wikipedia, and a mix of genuine and fake news articles**. Across these datasets, our method consistently demonstrated superior detection capabilities when applied to full training data, a limited number of training data, or imbalanced-training data, indicating its stability and effectiveness across a wide range of scenarios (see the results in Tables 1-6).
>
> >Q4. The algorithms presented, such as the training deep kernel with MMD-MP (Algorithm 1) and testing with MMD-MP for 2ST (Algorithm 2), are specifically tailored to address the MGT detection problem. Given the intricate mathematical formulations, it may be challenging to adapt or generalize these algorithms to other domains or problems without significant modifications. For researchers or practitioners looking for broader applications, this specificity might be a limitation. How can the proposed algorithms be adapted to other domains or challenges beyond MGT detection?
>
> **A4.** Thank you for your constructive comments. Currently, our method, MMD-MP, is elaborately designed to address the specific challenge of MGT detection, which **aligns with the main research goal of our paper**. Our goal is to provide an effective solution for text detection and make our methods more accessible for readers seeking proficiency in this domain.
>
> However, it's crucial to note that **our method is not limited to text detection alone**. Our proposed detection methods can be easily adaptive to other domains (e.g., image domains) just by replacing the text data with preferred data, such as distribution shift detection.
>
> For example, following the settings in [r4], we perform distribution shift detection by detecting CIFAR-10 from CIFAR-10.1. We use the same network as [r4] and present the test power in Table I. The presented experimental results highlight the adaptability of our approach beyond the scope of text detection, showing its versatility across different domains and challenges.
>
> Table I: Comparison of our method with baselines in terms of test power for **detecting CIFAR-10 from CIFAR-10.1**.
> | Method     | MMD-O    | C2ST-S   | C2ST-L   | MMD-D    | MMD-MP (ours) |
> | ---------- | -------- | -------- | -------- | -------- | ------------- |
> | Test Power | $35.30$ | $25.40$ | $30.80$ | $66.30$ | $\mathbf{82.70}$  |

---

> ### Author Response · Authors · 2023-11-21
> **Looking forward to the response from Reviewer 4WLs**
>
> Dear Reviewer 4WLs,
>
> We would like to thank you for your valuable comments on our paper. We sincerely hope that our response has addressed your initial concerns. If there are still unclear parts to you, please kindly let us know. We will do our best to address them.
>
> Best regards,
>
> Authors of #1084

---

> ### Comment · Reviewer_4WLs · 2023-11-21
> **Responses to #1084**
>
> Thank you to the authors for their good rebuttal. Upon revisiting the paper and the responses with care, I can see that they have indeed addressed most of my primary concerns. While the MMD method might not be novel in some areas, its effectiveness in distinguishing between LLM-generated and human text is evident from the authors' results. The addition of MMD-MP, where 'MP' accounts for multiple text populations from various LLMs, enhances its practicality.
>
> However, there are 3 tips as for improvement in this paper:
>
> 1. While the work focuses on LLMs, the authors do not provide a section on LLM-related work; I only find a section in the Appendix (sections B.1 and B.2), focusing primarily on statistical methods. A dedicated section introducing LLM-related work would be beneficial. It would be better if the authors could include references to LLM-generated text works [A-G] in Appendix B.3, enhancing the paper's comprehensiveness.
>
> 2. Another minor question is: why not use open-source LLMs, given that the API of the OpenAI “Turbo-type” model is costly, especially considering the number of samples in your dataset? And the response rate in terms of tokens per minute is also limited.
>
> 3. Based on my experience, there are some tips in the official documents for tuning hyperparameters that can control the **diversity** of the generated texts. This would be beneficial for future discussions. (The experiments in this work are generally comprehensive, so the authors don't need to conduct additional experiments.)
>
>
> [A] Tang, R., Chuang, Y. N., & Hu, X. (2023). The science of detecting llm-generated texts. arXiv preprint arXiv:2303.07205.
>
> [B] Dou, Y., Forbes, M., Koncel-Kedziorski, R., Smith, N. A., & Choi, Y. (2021). Is GPT-3 text indistinguishable from human text? SCARECROW: A framework for scrutinizing machine text. arXiv preprint arXiv:2107.01294.
>
> [C] Wu, J., Yang, S., Zhan, R., Yuan, Y., Wong, D. F., & Chao, L. S. (2023). A Survey on LLM-gernerated Text Detection: Necessity, Methods, and Future Directions. arXiv preprint arXiv:2310.14724.
>
> [D] Abburi, H., Suesserman, M., Pudota, N., Veeramani, B., Bowen, E., & Bhattacharya, S. (2023). Generative ai text classification using ensemble llm approaches. arXiv preprint arXiv:2309.07755.
>
> [E] Pei, X., Li, Y., & Xu, C. (2023). GPT Self-Supervision for a Better Data Annotator. arXiv preprint arXiv:2306.04349.
>
> [F] Pangakis, N., Wolken, S., & Fasching, N. (2023). Automated Annotation with Generative AI Requires Validation. arXiv preprint arXiv:2306.00176.
>
> [G] Tang, R., Han, X., Jiang, X., & Hu, X. (2023). Does synthetic data generation of llms help clinical text mining?. arXiv preprint arXiv:2303.04360.
>
> I will improve my score for this work later and am looking forward to the authors' response.

---

> ### Author Response · Authors · 2023-11-21
> **Thanks to Reviewer 4WLs**
>
> We are happy to receive your positive feedback on our rebuttal and your careful consideration of our responses. In response to your valuable insights, we have made the following improvements to the paper:
>
> * We have added a new section on LLM-related work in Appendix B, and discussed your mentioned works [A-G] to enhance the comprehensiveness of our paper.
>
> * Regarding the use of open-source LLMs, we would like to clarify that all the models used in our paper are open-source, such as Openai-D and ChatGPT-D.
>
> * We appreciate your suggestion regarding hyperparameter tuning to control the diversity of generated texts. We acknowledge the importance of techniques such as tuning temperature setting and sampling strategy, and we will consider these in our future work.

---

> ### Author Response · Authors · 2023-11-22
> **Thanks to Reviewer 4WLs**
>
> Dear Reviewer 4WLs,
>
> Thank you for increasing your score! Your invaluable comments have strengthened our paper a lot!
>
> Best regards,
>
> Authors of #1084

---

### Official Review · Reviewer_DBzA · 2023-11-01

**Soundness:** 3 good
**Presentation:** 3 good
**Contribution:** 4 excellent
**Rating:** 6
**Confidence:** 3

**Summary:**

The paper addresses the challenge of distinguishing between machine-generated and human-generated text. The authors propose using Maximum Mean Discrepancy (MMD) as a tool to identify subtle distributional discrepancies. To account for variations in text generated by different LLMs, they introduce a novel approach called multi-population aware optimization for kernel-based MMD (MMD-mp), with each population representing the distribution of one LLM. The MMD-mp method is applied in two settings: single sentence detection and paragraph detection, where individual sentences within a paragraph are treated as separate instances. The paper evaluates these settings separately and explores the impact of training data number on performance.

**Strengths:**

1. The paper's motivation for using MMD to discriminate between highly similar distributions, particularly in short sentences, is compelling. Short sentences often pose a greater challenge due to their simpler structure and wording, and MMD offers an effective solution.
2. The proposed method is supported by both theoretical analysis and experimental results.
3. The experimental design is comprehensive. The experiments involving the detection of text generated by unknown or unseen LLMs are interesting. Additionally, evaluating both short and long text is a well-rounded approach. The investigation into how the number of training data instances impacts performance strengthens the effectiveness of the proposed MMD method.

**Weaknesses:**

One recent SOTA baseline for detecting machine-generated texts is missing [1]. The paper has also shown impressive performance in short texts and needs to be discussed and compared in the paper.

[1]Stylometric Detection of AI-Generated Text in Twitter Timelines, Kumarage et. al.

**Questions:**

1. Does the inclusion of a more diverse range of LLMs' generated text in the training corpus improve or degrade performance?
2. How does the introduction of less capable LLMs' text into the training dataset, with distributions significantly different from other language models, affect the method's performance?
3. Is it feasible to treat an entire paragraph as a single sentence for detection, and what is the performance in this setting?

---

> ### Author Response · Authors · 2023-11-20
> **Responses to Reviewer DBzA [1/2]**
>
> We thank the reviewer for the encouraging comments and suggestions. Responses are below:
>
> >Q1. One recent SOTA baseline for detecting machine-generated texts is missing [1]. The paper has also shown impressive performance in short texts and needs to be discussed and compared in the paper.
> [1]Stylometric Detection of AI-Generated Text in Twitter Timelines, Kumarage et. al.
>
> **A1.** Thank you for the valuable suggestion. We have discussed the baseline [1] in our Related Work. Additionally, we have compared our method with this baseline at sentence-level detection on HC3. Table I shows that our method outperforms the baseline, demonstrating a higher AUROC.
>
> Table I: Comparison with the baseline [1] in terms of AUROC on HC3 dataset given 3,100 processed paragraphs.
> | Method        | ChatGPT     | GPT3-S      | Neo-S       | ChatGPT  & Neo-S | ChatGPT & GPT3-S |
> |:---------------:|:-------------:|:-------------:|:-------------:|:-----------------:|:------------------:|
> | FT_Stylo [1] | $94.57_{\pm 0.01}$ | $89.72_{\pm 0.00}$ | $82.17_{\pm 0.01}$ | $88.59_{\pm 0.00}$ | $83.92_{\pm 0.01}$ |
> | MMD-D         | $95.83_{\pm 0.37}$ | $94.86_{\pm 0.48}$ | $88.40_{\pm 1.28}$ | $91.39_{\pm 0.86}$ | $93.49_{\pm 0.46}$ |
> | MMD-MP (Ours) | $\mathbf{96.20}_{\pm 0.28}$ | $\mathbf{95.08}_{\pm 0.32}$ | $\mathbf{92.04}_{\pm 0.58}$ | $\mathbf{92.48}_{\pm 0.37}$ | $\mathbf{94.61}_{\pm 0.22}$ |
>
> >Q2. Does the inclusion of a more diverse range of LLMs' generated text in the training corpus improve or degrade performance?
>
> **A2.** Thank you for your insightful comments on the impact of including a more diverse range of LLMs' generated text during training on model performance. Based on our current experiments, the majority of cases show **a decrease in performance**. For instance, as shown in Tables II, III,IV, V,  for both MMD-D and our MMD-MP, when ChatGPT is separately mixed with texts from GPT3-S and Neo-S for training, the performance of the mixed model on their respective original text domains tends to decrease. While including more diverse texts may enhance overall generalization capabilities, its performance within the original text domains could be degraded. Exploring this trade-off is a consideration for our future work.
>
> Table II: AUROC of MMD-D before and after mixing ChatGPT and GPT3-S texts during training.
>
> |  Text domain  |      ChatGPT       |       GPT3-S       |
> |:-------------:|:------------------:|:------------------:|
> | before mixing | $95.83_{\pm 0.37}$ | $94.86_{\pm 0.48}$ |
> | after  mixing | $94.80_{\pm 0.60}$ | $93.92_{\pm 0.62}$ |
>
> Table III: AUROC of MMD-D before and after mixing ChatGPT and Neo-S texts during training.
>
> |  Text domain  |      ChatGPT       |       Neo-S        |
> |:-------------:|:------------------:|:------------------:|
> | before mixing | $95.83_{\pm 0.37}$ | $94.86_{\pm 0.48}$ |
> | after  mixing | $93.77_{\pm 0.98}$ | $89.41_{\pm 0.92}$ |
>
> Table IV: AUROC of our MMD-MP before and after mixing ChatGPT and GPT3-S texts during training.
>
> |  Text domain  |      ChatGPT       |       GPT3-S       |
> |:-------------:|:------------------:|:------------------:|
> | before mixing | $96.20_{\pm 0.28}$ | $95.08_{\pm 0.32}$ |
> | after  mixing | $95.02_{\pm 0.44}$ | $94.22_{\pm 0.49}$ |
>
> Table V: AUROC  of our MMD-MP before and after mixing ChatGPT and Neo-S texts during training.
>
> |  Text domain  |      ChatGPT       |       Neo-S        |
> |:-------------:|:------------------:|:------------------:|
> | before mixing | $96.20_{\pm 0.28}$ | $92.04_{\pm 0.58}$ |
> | after  mixing | $94.29_{\pm 0.66}$ | $90.11_{\pm 0.66}$ |
>
> >Q3. How does the introduction of less capable LLMs' text into the training dataset, with distributions significantly different from other language models, affect the method's performance?
>
> **A3.** We appreciate the insightful question. **The introduction of less capable LLMs' text into the training can yield varying effects on overall detection performance**.
> * As illustrated in Table 3, on the HC3 dataset, when incorporating ChatGPT text into GPT3-S the AUROC performance **decreases** to 94.61, contrasting with their respective individual training AUROC values (GPT3-S: 95.08, ChatGPT: 96.20).
> * As shown in Table 17, on the XSum dataset, introducing Neo-S (125 m) text to GPT2-M and GPT3-S results in an AUROC **improvement** from 87.79 to 88.27.
>
> This phenomenon also exists in other baselines. We hypothesize that several factors contribute to this complexity, including **the sample size, distribution characteristics, inherent language styles, and domain differences** among the diverse LLMs. We leave this for our future work.

---

> ### Author Response · Authors · 2023-11-20
> **Responses to Reviewer DBzA [2/2]**
>
> >Q4. Is it feasible to treat an entire paragraph as a single sentence for detection, and what is the performance in this setting?
>
> **A4.** Yes, our method can treat an entire paragraph as a single sentence for detection. However, it's important to note that excessively long paragraphs may exceed the processing capabilities of the pre-trained LLM. When detecting paragraph-level sentences, those that exceed maximum tokens will be truncated by the LLM. Even under this setting, we provide the results of our model to directly test with the random sampled $1,000$ paragraph-level sentences in Table IV. The results still demonstrate the effectiveness of our methods.
>
> Table VI: Comparison of our method with baselines in terms of AUROC for **detecting paragraph-level sentences**.
> | Method        | ChatGPT     | GPT3-S      | Neo-S       | ChatGPT  & Neo-S | ChatGPT & GPT3-S |
> |:---------------:|:-------------:|:-------------:|:-------------:|:-----------------:|:------------------:|
> | Likelihood    | $99.34_{\pm 0.11}$ | $74.62_{\pm 0.73}$ | $83.54_{\pm 0.21}$ | $90.76_{\pm 0.06}$ | $86.47_{\pm 1.33}$ |
> | Rank          | $91.88_{\pm 0.40}$ | $81.67_{\pm 0.75}$ | $87.56_{\pm 0.42}$ | $89.52_{\pm 0.10}$ | $86.83_{\pm 0.55}$ |
> | Log-Rank      | $99.37_{\pm 0.12}$ | $79.59_{\pm 0.64}$ | $88.00_{\pm 0.25}$ | $93.26_{\pm 0.18}$ | $89.01_{\pm 1.23}$ |
> | Entropy       | $7.04_{\pm 0.08}$ | $50.35_{\pm 0.14}$ | $47.94_{\pm 0.21}$ | $28.60_{\pm 0.30}$ | $28.73_{\pm 1.13}$ |
> | DetectGPT-d   | $94.31_{\pm 0.03}$ | $86.63_{\pm 0.92}$ | $87.31_{\pm 0.88}$ | $91.23_{\pm 0.66}$ | $90.43_{\pm 0.75}$ |
> | DetectGPT-z   | $96.28_{\pm 0.00}$ | $86.55_{\pm 0.94}$ | $87.60_{\pm 0.85}$ | $92.13_{\pm 0.46}$ | $91.36_{\pm 0.73}$ |
> | OpenAI-D      | $98.75_{\pm 0.08}$ | $99.62_{\pm 0.05}$ | $99.41_{\pm 0.09}$ | $98.94_{\pm 0.14}$ | $98.96_{\pm 0.04}$ |
> | ChatGPT-D     | $\mathbf{99.79}_{\pm 0.05}$ | $69.03_{\pm 0.03}$ | $67.40_{\pm 1.10}$ | $82.85_{\pm 0.88}$ | $85.88_{\pm 0.05}$ |
> | CE-Classifier | $99.13_{\pm 0.15}$ | $98.29_{\pm 0.13}$ | $96.33_{\pm 0.40}$ | $95.71_{\pm 0.12}$ | $94.59_{\pm 0.33}$ |
> | MMD-O         | $76.29_{\pm 1.92}$ | $87.34_{\pm 0.96}$ | $86.64_{\pm 0.42}$ | $81.63_{\pm 0.74}$ | $81.32_{\pm 0.48}$ |
> | MMD-D           | $97.42_{\pm 0.19}$ | $99.65_{\pm 0.07}$ | $99.46_{\pm 0.16}$ | $98.65_{\pm 0.16}$ | $99.26_{\pm 0.05}$ |
> | MMD-MP (Ours) | $99.59_{\pm 0.08}$ | $\mathbf{99.71}_{\pm 0.05}$ | $\mathbf{99.65}_{\pm 0.08}$ | $\mathbf{99.01}_{\pm 0.17}$ | $\mathbf{99.32}_{\pm 0.13}$ |

---

> ### Author Response · Authors · 2023-11-22
> **Looking forward to the response from Reviewer DBzA**
>
> Dear Reviewer DBzA,
>
> We would like to thank you for your valuable comments on our paper. We sincerely hope that our response has addressed your initial concerns. If there are still unclear parts to you, please kindly let us know. We will do our best to address them.
>
> Best regards,
>
> Authors of #1084

---

### Official Review · Reviewer_1AtS · 2023-11-09

**Soundness:** 4 excellent
**Presentation:** 3 good
**Contribution:** 3 good
**Rating:** 8
**Confidence:** 3

**Summary:**

This paper first points out the existing optimization dilemma on using vanilla MMD to detect machine-generated text, and then proposes MMD-MP, a multi-population aware optimization method for maximum mean discrepancy. MMD-MP avoids variance increases and improve the stability to emasure the distribution discrepancy.  Relying on MMD-MP, this paper proposes two methods for machine-generated text detection. Experiments were done on different LLMs (GPT-2s, GPT-3s, chatGPT, GPT-Neo) and different LLM settings.

**Strengths:**

- This paper introduces MMD-MP, a multi-population aware optimization method, which achieves better machine-generated text detection rate than baseline.
- This paper has a very clear mathematical formulation of the MMD-MP method, as well as provide reasoning on why they choose to make this optimization on top of vanilla MMD.

**Weaknesses:**

- Test power is not very straightforward to interpret. I suggest adding an easier-to-understand explanation in additional to the current definition in section 2. In my understanding, the higher the test power number is, the more “confident” the model is to the idea that the two distribution is different, right?
- Easier-to-read figures:
    - This might be a nit-pick but figure 1 and 2 are very packed. I understand that the authors want to plot MMD-D and MMD-MP out for better comparison, but maybe consider removing the triangles & squares & circles?
    - What does the last two column mean in table 1? Is it when $q=3$, so the three “clusters” are human-written text, ChatGPT text, and GPT3-S text?
- The method is overall, slightly complicated. The experiments when $q=3$ are conducted on “different enough” language models. I wonder if better LMs are included in the comparison (GPT-4, chatGPT), how will the method perform.

**Questions:**

- extra parenthesis in section 1 paragraph 2 citation.
- what is the definition of X’ and Y’ in definition 1? Are they also RVs from P and Q?
- Metrics — why choose test power, not something like macro-accuracy?
- Although the appendix mentioned that you use the same decoding method as the detectGPT paper, they actually have a section that test out different settings of decoding method. Can you tell me what parameter and sampling method you are using?
- I might have a misunderstanding but why only generate MGT with the first 20 prompts? What are these for? Since the evaluation metric paragraph mentioned 100 paragraphs / 1,000 sentences were used for testing, and section 4.2 mentioned mentioned 3,100 paragraphs were used for training.

---

> ### Author Response · Authors · 2023-11-20
> **Responses to Reviewer 1AtS [1/2]**
>
> We thank the reviewer for the encouraging comments and suggestions. Responses are below:
>
> >Q1. Test power is not very straightforward to interpret. I suggest adding an easier-to-understand explanation in additional to the current definition in section 2. In my understanding, the higher the test power number is, the more “confident” the model is to the idea that the two distribution is different, right?
>
> **A1.** Thank you for your valuable comments. You are correct; the concept of test power represents the model's confidence in identifying differences between two distributions. A higher test power indicates a greater level of certainty regarding the distributional discrepancy. We have clarified this in detail in Section 2.1 of our paper.
>
> >Q2. This might be a nit-pick but figure 1 and 2 are very packed. I understand that the authors want to plot MMD-D and MMD-MP out for better comparison, but maybe consider removing the triangles & squares & circles?
>
> **A2.** We appreciate your suggestion. The decision to use different markers and line types in Figures 1 and 2 is made with consideration for print quality.
>
> >Q3. What does the last two column mean in table 1? Is it when $q=3$, so the three “clusters” are human-written text, ChatGPT text, and GPT3-S text?
>
> **A3.** The last two columns in Table 1 correspond to the detection of ChatGPT texts and GPT3-S texts from human-written texts. This scenario aligns with the case where $q=2$ represents two machine-generated text populations. We have clarified this in our experiments.
>
> >Q4. The method is overall, slightly complicated.
>
> **A4.** We appreciate your constructive comments. We have taken measures to enhance the presentation of intuitive explanations for our method.
>
> * **More intuitive MMD Explanations:** We have added intuitive explanations for MMD's distance, e.g., viewing $k(X,X′)$ or $k(Y,Y′)$ as an intra-class distance and $k(X,Y)$ as an inter-class distance in Section 2.1, aiding for understanding its definition.
> * **More intuitive explanations for the optimization of kernel-based MMD:** For MMD-D optimization, we highlight that it intuitively tends to separately aggregate human-written texts (HWTs) and all possible different-population machine-generated texts (MGTs) like decreasing intra-class distance, and simultaneously push them away from each other like increasing inter-class distance. This presents challenges due to significant fluctuations or higher MMD variance. In contrast, our MMD-MP relaxes constraints on aggregating all MGTs populations (i.e., removing the $k(Y,Y′)$ from MMD), focusing more on fitting HWTs. This stabilizes MMD values and enhances the transferability of detection. We have included more illustrative figures to depict this procedure in Section D.
> * **Extensive intuitive observations with detailed analyses：** We have provided extensive intuitive explanations in Section 2.2, 2.3 and Appendix D, G, H, offering insights into the motivation through empirical observations in figures. Moreover, we have carefully provided more detailed analyses to support these observations, aiding readers in comprehending the core of our method.
> * **More detailed algorithms presentation:** We have provided a more detailed algorithmic presentation in Section C.2, offering clear steps for Algorithms 1, 2, and 3. This ensures readers have a lucid pathway to understand our method's workflow.
> * **Clearer objective function construction:** At the beginning of Section 3.2 of our method, we have introduced the optimization objective function $J$ and followed it with theoretical derivations. This enables readers to directly observe the optimization objective function without delving into the followed derivation process, guiding even those finding statistical theory challenging. We have provided a clearer description of this.
>
> Our goal is to strike a balance between maintaining the rigor of our theoretical analyses and providing enough assistance for readers to understand the key aspects of our method.
>
> >Q5. The experiments when $q=3$ are conducted on “different enough” language models. I wonder if better LMs are included in the comparison (GPT-4, chatGPT), how will the method perform.
>
> **A5.** Thank you for your suggestion. Regarding the inclusion of better language models like GPT-4 and ChatGPT, we acknowledge the potential improvement that superior models may bring. However, considering factors like the slower generation process of GPT-4 series, we have demonstrated the transferability performance of our method when training on texts from ChatGPT, GPT-2, and GPT2-M and testing on GPT4all-j texts in Tables 5, 6 in the paper. In our future work, we plan to train the detector on a more diverse dataset containing both ChatGPT and GPT-4 texts, creating a more inclusive evaluation platform.

---

> ### Author Response · Authors · 2023-11-20
> **Responses to Reviewer 1AtS [2/2]**
>
> >Q6. extra parenthesis in section 1 paragraph 2 citation.
>
> **A6:** We have fixed it.
>
> >Q7. What is the definition of X’ and Y’ in definition 1? Are they also RVs from P and Q?
>
> **A7:** Yes. We have detailed this.
>
> >Q8. Metrics — why choose test power, not something like macro-accuracy?
>
> **A8:** Test power is widely employed in detecting differences between two distributions, providing **a reliable measure of a model's ability to identify discrepancies** [r1, r2, r3]. It is particularly suitable for **binary classification problems**, where the goal is to identify whether two sampled populations are from the same distribution.
>
> On the other hand, metrics like macro-accuracy are more suited for **multi-class classification tasks**, where the emphasis is on correctly classifying each class independently. In the task of distinguishing between machine-generated and human-written texts (a binary classification), test power aligns more closely with the requirements of the problem.
>
> [r1] Learning deep kernels for non-parametric two-sample tests, ICML 2020.
>
> [r2] Meta two-sample testing: Learning kernels for testing with limited data, NeurIPS 2021.
>
> [r3] MMDA ggregated Two-Sample Test, JMLR 2023.
>
> >Q9. Although the appendix mentioned that you use the same decoding method as the detectGPT paper, they actually have a section that test out different settings of decoding method. Can you tell me what parameter and sampling method you are using?
>
> **A9:** Thank you for your comments. In our experiments, we utilized the default settings of the decoding method as presented in the detectGPT paper. Specifically, we employed temperature sampling as the sampling strategy with a temperature value consistent with the default settings in detectGPT. Additionally, we set the number of perturbations to $100$ in our experimental evaluations.
>
> >Q10. I might have a misunderstanding but why only generate MGT with the first 20 prompts? What are these for? Since the evaluation metric paragraph mentioned 100 paragraphs / 1,000 sentences were used for testing, and section 4.2 mentioned 3,100 paragraphs were used for training.
>
> **A10:** The strategy of using the first $20$ prompts to generate MGTs is consistent with that of detectGPT paper. This approach aims to mitigate the influence of different corpora on detection performance. Specifically, for constructing MGTs, we utilize the initial $20$ tokens from ChatGPT text in HC3 dataset as prompts, allowing other LLMs to continue the text generation process.

---

> ### Author Response · Authors · 2023-11-22
> **Looking forward to the response from Reviewer 1AtS**
>
> Dear Reviewer 1AtS,
>
> We have addressed your initial concerns regarding our paper. We are happy to discuss them with you in the openreview system if you feel that there still are some concerns/questions. We also welcome new suggestions/comments from you!
>
> Best regards,
>
> Authors of #1084

---

### Official Review · Reviewer_bnWc · 2023-11-10

**Soundness:** 2 fair
**Presentation:** 3 good
**Contribution:** 2 fair
**Rating:** 6
**Confidence:** 3

**Summary:**

This paper investigates the detection of LLM-generated text based on MMD, a measure of distributional similarity. MMD-based detectors work well for text generated by only a single LLM, but less well when text can come from multiple LLMs. The paper proposes a solution to deal with this.

MMD-based detectors are trained with a loss that involves comparisons between paired text samples that are either both human generated, both machine generated, or mixed. The proposed solution is to drop the comparison between the samples that are both machine generated, as this can cause problems when such samples come from different LLMs. Lots of math is provided to justify this strategy.

The proposed method is compared to various baselines in a cross-validation setup on a corpus of 3000 human- and machine-generated paragraphs. Two settings are used: comparing the power of a hypothesis test for whether two paragraphs are generated by the same method; and classification performance in detecting whether a single sentence is machine-generated. The proposed method outperforms all baselines on these tasks, and it is robust to generated text coming from multiple LLMs.

**Strengths:**

The paper attacks a crucial problem and it is clearly written.

The paper is technically very sophisticated, and provides extensive theoretical justification for the proposed methods.

Results on the tested setting are very clearly positive.

**Weaknesses:**

The paper considers variations in text due to using different LLMs, but it basically ignores variation due to different genres, styles, and human authors. I think this is an artifact of sticking to a single small corpus (in the main paper), where it is easy to generate new LLM outputs by using the provided prompts, but all other attributes have to remain fixed. These other attributes are probably more important in real-world applications.

I am not sure whether experiments on such a small training corpus are meaningful. Due to the cross-validation setup, it is very possible that all models that are trained on the corpus are over-fitting to this setting. Baselines like ChatGPT-D that may be trained on much larger external corpora are therefore at a disadvantage. Another potential disadvantage is that the single-sentence detection test is not very realistic; it would have been better to perform this test at the paragraph level. A final problem with the experiment section is that many details are missing - see questions below.

The proposed method is quite complicated. Apart from complexity of implementation, a potentially large disadvantage is that the loss is quadratic in training corpus size. The authors don’t address this issue at all.

Although it has a lot of supporting math, the main contribution is actually a small modification (dropping a term in the loss), that is quite intuitive. There is nothing wrong with this, but the paper would have been stronger had it focused more on providing this intuition up front, and less on the formal details.

**Questions:**

Nit: human-written text -> HWT, human-generated text -> HGT [choose one]

Figure 1 is quite hard to interpret at this point in the paper.

If I interpret (3) correctly, this loss is quadratic in sample size, which would seem to limit the quantity of data it is practical to train on. Do you use an approximation to deal with this?

Where do the different machine-generated populations in 2.2 come from? In what sense is there only a single human population (since there are also lots of different kinds of human-generated texts).

Related to the above, how is (3) generalized to deal with multiple populations? Do you keep the machine samples separate somehow, or merge them?

p6 “... numerous works have shown MMD as a powerful means…” - please provide citations.

s4 What language is the corpus?

s4 What architecture and scale of neural models do you use? Are these the same for your model and all baselines? Which baselines train on the provided corpus, and which don’t? What are the hyper-parameters for the kernel functions? Is there anything about the setting (model scale, data size) that might be optimal for your model but not the baselines?

s4 Please provide more details about how test power was compared for these methods. If it was derived from theoretical estimates for both your methods and C2ST-L, can you be sure both are equally good estimates of the true power?

In Tables 1-4, ChatGPT has consistently the best results for all methods, indicating that it is easier to distinguish from human text. But one would expect it to be harder, since it is a better model than the others. Do you have an explanation for this?

---

> ### Author Response · Authors · 2023-11-20
> **Responses to Reviewer bnWc [1/4]**
>
> We thank the reviewer for the encouraging comments and suggestions. Responses are below:
>
> >Q1. The paper considers variations in text due to using different LLMs, but it basically ignores variation due to different genres, styles, and human authors. I think this is an artifact of sticking to a single small corpus (in the main paper), where it is easy to generate new LLM outputs by using the provided prompts, but all other attributes have to remain fixed. These other attributes are probably more important in real-world applications.
>
> **A1.** Thank you for your insightful comments. We appreciate your consideration regarding the scope of text variations in our experiments. Due to space constraints, we did not provide a detailed description of the training corpus in the main body of our paper. However, it's essential to note that **our experiments comprehensively cover diverse genres, styles, and human authors considerations**.
>
> The datasets employed, including HC3 and XSum, **encompass diverse language domains such as finance, healthcare, Wikipedia, and a mix of genuine and fake news articles**. Our training data, excluding ChatGPT for comparison, involves machine-generated text from the initial $20$ tokens of each prompt. Notably, we rigorously conduct these experiments with at least five different random seeds for each setting, ensuring robustness in our findings.
>
> Our approach, coupled with this diverse training dataset and careful experimental design, allows us to capture variations in styles, genres, and other linguistic attributes. We can also increase more types of LLM texts to further improve the adaption and scalability of our method, which leaves for our future work. We have clarified this in Section C.1.
>
> >Q2. I am not sure whether experiments on such a small training corpus are meaningful. Due to the cross-validation setup, it is very possible that all models that are trained on the corpus are over-fitting to this setting. Baselines like ChatGPT-D that may be trained on much larger external corpora are therefore at a disadvantage.
>
>
> **A2.** Thank you for your insightful comments on the size of the training corpus. We understand the importance of addressing potential overfitting in smaller datasets and would like to provide a comprehensive response from the following perspectives.
> * **Fairness in comparison:** To ensure a fair comparison, given that ChatGPT-D is fine-tuned on HC3 dataset, we train a deep kernel on the same HC3 dataset to align with this baseline for comparison, aiming to establish parity in the evaluation process. Additionally, the XSum, we employed, is a commonly used dataset for evaluating text detection methods [r1, r2].
>
>     [r1] DetectGPT: Zero-Shot Machine-Generated Text Detection using Probability Curvature, ICML 2023.
>
>     [r2] RADAR: Robust AI-Text Detection via Adversarial Learning, NeurIPS 2023.
> * **Dataset diversity:** HC3 and XSum datasets, comprising diverse language domains such as finance, healthcare, Wikipedia, and a mix of genuine and fake news articles, offer a rich and varied landscape for evaluating the effectiveness of text detection methods. This diversity ensures that our evaluation captures a broad spectrum of linguistic styles and content.
> * **Effectiveness across various scenarios:** Our method consistently demonstrates superior detection performance across different scenarios within these datasets. Whether applied to the full training data, a limited number of training instances, or imbalanced-training data, our approach maintains its effectiveness (see Tables 1-6). This consistency suggests the adaptability and versatility of our method across a wide range of scenarios.
>
> In the future, we plan to train the detector on a more diverse corpus covering a broader range of scenarios. We believe that our approach, based on the above considerations, provides meaningful insights into text detection across various real-world contexts.

---

> ### Author Response · Authors · 2023-11-20
> **Responses to Reviewer bnWc [2/4]**
>
> >Q3. Another potential disadvantage is that the single-sentence detection test is not very realistic; it would have been better to perform this test at the paragraph level.
>
> **A3.** We appreciate your suggestion to consider paragraph-level detection. In our method, we introduce two distinct methods for text detection: one focuses on sentence-level detection and another on paragraph-level detection.
>
> * For the sentence-level approach, we treat each sentence individually, recognizing the importance of short text detection, especially in applications like online content filtering, advertising fraud detection and false information recognition. This method allows us to assess the credibility of individual sentences within larger textual contexts, providing valuable insights into various real-world scenarios. However, we also recognize the **limitations of many LLMs when it comes to handling particularly long paragraphs**. Most LLMs face challenges with lengthy text inputs, as they reach token limits and may truncate content.
>
> * In contrast, our paragraph-level detection method does not encounter the above issues, **offering an advantage in scenarios involving extended textual passages**. Our paragraph-level detection approach involves breaking down paragraphs into individual sentences and then treats the distribution of sentences within a paragraph as a whole, employing a two-sample test method for detection.
>
> It’s crucial to note that **our sentence-level approach is also adaptive to detect paragraph-level sentences**. We can treat the entire paragraph as a single entity for single-sample detection. We have presented these results in Table I.  From the table, our approach also yields superior or comparable performance with baselines, providing flexibility in detection strategies based on specific application needs. We have included this in Section K in Appendix.
>
> Table I: Comparison of our method with baselines in terms of AUROC for **detecting paragraph-level sentences**.
> | Method        | ChatGPT     | GPT3-S      | Neo-S       | ChatGPT  & Neo-S | ChatGPT & GPT3-S |
> |:---------------:|:-------------:|:-------------:|:-------------:|:-----------------:|:------------------:|
> | Likelihood    | $99.34_{\pm 0.11}$ | $74.62_{\pm 0.73}$ | $83.54_{\pm 0.21}$ | $90.76_{\pm 0.06}$ | $86.47_{\pm 1.33}$ |
> | Rank          | $91.88_{\pm 0.40}$ | $81.67_{\pm 0.75}$ | $87.56_{\pm 0.42}$ | $89.52_{\pm 0.10}$ | $86.83_{\pm 0.55}$ |
> | Log-Rank      | $99.37_{\pm 0.12}$ | $79.59_{\pm 0.64}$ | $88.00_{\pm 0.25}$ | $93.26_{\pm 0.18}$ | $89.01_{\pm 1.23}$ |
> | Entropy       | $7.04_{\pm 0.08}$ | $50.35_{\pm 0.14}$ | $47.94_{\pm 0.21}$ | $28.60_{\pm 0.30}$ | $28.73_{\pm 1.13}$ |
> | DetectGPT-d   | $94.31_{\pm 0.03}$ | $86.63_{\pm 0.92}$ | $87.31_{\pm 0.88}$ | $91.23_{\pm 0.66}$ | $90.43_{\pm 0.75}$ |
> | DetectGPT-z   | $96.28_{\pm 0.00}$ | $86.55_{\pm 0.94}$ | $87.60_{\pm 0.85}$ | $92.13_{\pm 0.46}$ | $91.36_{\pm 0.73}$ |
> | OpenAI-D      | $98.75_{\pm 0.08}$ | $99.62_{\pm 0.05}$ | $99.41_{\pm 0.09}$ | $98.94_{\pm 0.14}$ | $98.96_{\pm 0.04}$ |
> | ChatGPT-D     | $\mathbf{99.79}_{\pm 0.05}$ | $69.03_{\pm 0.03}$ | $67.40_{\pm 1.10}$ | $82.85_{\pm 0.88}$ | $85.88_{\pm 0.05}$ |
> | CE-Classifier | $99.13_{\pm 0.15}$ | $98.29_{\pm 0.13}$ | $96.33_{\pm 0.40}$ | $95.71_{\pm 0.12}$ | $94.59_{\pm 0.33}$ |
> | MMD-O         | $76.29_{\pm 1.92}$ | $87.34_{\pm 0.96}$ | $86.64_{\pm 0.42}$ | $81.63_{\pm 0.74}$ | $81.32_{\pm 0.48}$ |
> | MMD-D           | $97.42_{\pm 0.19}$ | $99.65_{\pm 0.07}$ | $99.46_{\pm 0.16}$ | $98.65_{\pm 0.16}$ | $99.26_{\pm 0.05}$ |
> | MMD-MP (Ours) | $99.59_{\pm 0.08}$ | $\mathbf{99.71}_{\pm 0.05}$ | $\mathbf{99.65}_{\pm 0.08}$ | $\mathbf{99.01}_{\pm 0.17}$ | $\mathbf{99.32}_{\pm 0.13}$ |

---

> ### Author Response · Authors · 2023-11-20
> **Responses to Reviewer bnWc [3/4]**
>
> >Q4. The proposed method is quite complicated. Apart from complexity of implementation, a potentially large disadvantage is that the loss is quadratic in training corpus size. The authors don’t address this issue at all.
>
> **A4.** We appreciate your insights into our proposed method and the concerns raised regarding the quadratic nature of the loss concerning the training corpus size.
>
> **Simplified method overview:** Contrary to the perceived complexity, our detection method is implemented with a straightforward manner. To summarize, we first extract features from both human-written text (HWT) and machine-generated text (MGT) sentences using a fixed feature extractor. Then, we train a deep kernel by maximizing the objective $J$ in Eqn. (4) (Algorithm 1) for both paragraph-level and sentence-level detection scenarios.
>
> * For paragraph-level detection, we compute the estimated MMD between given HWTs and test paragraph sentences ("$est$"). We then iteratively randomize and split these mixed sentences, calculating MMD values as "$perm_i$". We obtain test power by comparing "$perm_i$" with "$est$" (Algorithm 2).
> * For sentence-level detection, using a set of HWTs as a reference set, we compute the MMD distance between the test single sentence and reference samples (Algorithm 3).
>
> We have provided a more detailed and clarified description of this process in Section C.2.
>
> **Concern on quadratic loss:** While the optimization function exhibits a quadratic relationship with sample size $n$, in practice, we implement this through the use of **mini-batch sampling during training**. Then each iteration of training in Algorithm 1 costs $\mathcal{O}(m^2)$, where $m$ is the minibatch size. This practice not only addresses computational concerns but also maintains the model's modest performance. More advanced techniques to enhance computational efficiency refer to [r3].
>
> >Q5. Although it has a lot of supporting math, the main contribution is actually a small modification (dropping a term in the loss), that is quite intuitive. There is nothing wrong with this, but the paper would have been stronger had it focused more on providing this intuition up front, and less on the formal details.
>
> **A5.** We appreciate your insightful suggestion and acknowledgment of the paper's main contribution, which involves a simple yet impactful strategy by removing a term in the optimization function of the original kernel-based MMD. We have provided the behind intuition in Section 2.3 and Section D.
>
> In Section 2.3, we have delved into the empirical verification of high variance during training in the original MMD. Through extensive experiments and discussions, we have illustrated that, training the kernel using MMD with multiple-population machine-generated texts (MGTs) compromises its optimization, as the intra-class distance $k(Y, Y')$ aggregates all different-population MGTs, resulting in highly fluctuating MMD. Based on these findings, we remove the term $k(Y, Y')$ in MMD.
>
> Moreover, to provide a clearer intuition, we have **incorporated additional figures in a new part, i.e., Section D, to illustrate the benefits of omitting  $k(Y, Y')$**  from a more intuitive perspective. These visuals aim to enhance the reader's understanding of the motivation of our proposed method, emphasizing its significance and relevance in the overall methodology. We hope these measures contribute to a more accessible and intuitive presentation of our key contributions.
>
> >Q6. Figure 1 is quite hard to interpret at this point in the paper.
>
> **A6.** We appreciate your comments on Figure 1.  It illustrates the  MMD values, MMD variances, and the test power of MMD-D and our MMD-MP during the optimization process. As the number of $S_Q$ populations (i.e., $q$) increases, MMD-D shows an increase in MMD, accompanied by a sharp rise in variance, resulting in unstable test power during testing. In contrast, our MMD-MP exhibits minimal variance in MMD values, leading to higher and more stable test power during testing. We have clarified this in the paper to enhance the figure's interpretability.
>
> >Q7. If I interpret (3) correctly, this loss is quadratic in sample size, which would seem to limit the quantity of data it is practical to train on. Do you use an approximation to deal with this?
>
> **A7.** Yes, we use minibatch data with batchsize $m=200$ to train the deep kernel through all our experiments.

---

> ### Author Response · Authors · 2023-11-20
> **Responses to Reviewer bnWc [4/4]**
>
> >Q8. Where do the different machine-generated populations in 2.2 come from? In what sense is there only a single human population (since there are also lots of different kinds of human-generated texts).
>
> **A8.** We appreciate your insightful comments. The diverse machine-generated populations stem from **various language models (LLMs)**, each showing unique qualities, diversity, and stylistic features attributed to **differences in architecture, parameters, training methods, and training data**.
>
> In contrast, human-written texts are treated as a single population due to **shared characteristics**. Human-generated texts exhibit linguistic commonalities, including grammatical structures, vocabulary, and expression styles. Additionally, they collectively reflect shared cultural elements and values. Collaboration among diverse authors further contributes to the usage of similar language, fostering cohesion in human-generated text.
>
> >Q9. Related to the above, how is (3) generalized to deal with multiple populations? Do you keep the machine samples separate somehow, or merge them?
>
> **A9.** We merge multiple populations as a distribution in Eqn (3).
>
> >Q10. p6 “... numerous works have shown MMD as a powerful means…” - please provide citations.
>
> **A10.** We have cited more references.
>
> >Q11. s4 What language is the corpus?
>
> **A11.** The corpus in our current experiments is English.
>
> >Q12. What architecture and scale of neural models do you use? Are these the same for your model and all baselines? Which baselines train on the provided corpus, and which don’t? What are the hyper-parameters for the kernel functions? Is there anything about the setting (model scale, data size) that might be optimal for your model but not the baselines?
>
> **A12.** **1) Architecture:** Our MMD-MP employs a neural network $\phi$ as the deep kernel $\phi_{\hat{f}}$, equipped with a feature extractor $\hat{f}$. For the feature extractor $\hat{f}$, we utilize OpenAI's RoBERTa-based GPT-2 detector model, identical to OpenAI-D. The last hidden state of this model is considered the feature of the input text. MMD-D shares the same architecture as MMD-MP, while other baselines—CE-Classifier, C2ST-S, and C2ST-L—use a similar architecture, with the exception of an additional binary classifier. **2) Detectors:** ChatGPT-D, CE-Classifier, C2ST-S, C2ST-L, MMD-D, and our MMD-MP are trained using the provided corpus. **3) Kernel hyper-parameters ($\omega={\epsilon, \phi, \sigma_{\phi}, \sigma_q}$) selection:** For all experiments, we set $\epsilon=10^{-10}$ and $\sigma_q=30$. In the case of our MMD-MP, we set $\sigma_{\phi}=55$ when the sample size $n=3,100$ and $\sigma_{\phi}=45$ when $n=1,000$. Accordingly, for MMD, we set $\sigma_{\phi}=20$ when the sample size $n=3,100$ and $\sigma_{\phi}=55$ when $n=1,000$. We have included these details Appendix C.3.
>
> >Q13. s4 Please provide more details about how test power was compared for these methods. If it was derived from theoretical estimates for both your methods and C2ST-L, can you be sure both are equally good estimates of the true power?
>
> **A13.** Two-sample test baselines are equally good estimates of the true power. We evaluate these baselines (e.g., CST-L, MMD-D) using the same metric, i.e., test power, following [r4, r5, r6]. These methods just employ different training strategies, but the evaluation is conducted uniformly, ensuring comparability in terms of test power.
>
> [r4] Learning deep kernels for non-parametric two-sample tests, ICML 2020.
>
> [r5] Meta two-sample testing: Learning kernels for testing with limited data, NeurIPS 2021.
>
> [r6] Maximum mean discrepancy test is aware of adversarial attacks, ICML 2021.
>
> >Q14. In Tables 1-4, ChatGPT has consistently the best results for all methods, indicating that it is easier to distinguish from human text. But one would expect it to be harder, since it is a better model than the others. Do you have an explanation for this?
>
> **A14.** Thank you for your thoughtful comments. Despite the expectation that a better model would be harder to distinguish from human text, our experiments in the paper indicate that the representational capacity or size of the language model (LLM) does not necessarily correlate with the difficulty of detecting its generated text. For instance, (GPT3-S 551m with AUROC 86.34) versus (Neo-S 125m with AUROC 76.35). We hypothesize that the intricate language patterns introduced by ChatGPT might play a crucial role in this phenomenon.

---

> ### Author Response · Authors · 2023-11-22
> **Looking forward to the response from Reviewer bnWc**
>
> Dear Reviewer bnWc,
>
> We have tried our best to address all the concerns and provided explanations to all questions. We sincerely hope that our answer has addressed your initial concerns. Kindly let us know if you have any other concerns, and we will do our best to address them.
>
> Best regards,
>
> Authors of #1084

---

> > ### Comment · Reviewer_bnWc · 2023-11-22
> > **thanks for the response**
> >
> > Thanks for your extremely thorough response, and all the added explanations and information. I have no doubts about the inherent soundness of your method, but I am still not convinced that your experiments are meaningful.
> >
> > The main problem is that you achieve such good results when training only on ~3k paragraphs, and the results are better for ChatGPT than for weaker models. To me, this points to corpus-specific artifacts in EC3, perhaps because it is a question-answering setting, and ChatGPT has certain characteristic ways of answering the fixed pattern of questions in that corpus. I don’t think this would be representative of performance in a truly blind setting (in appendix J you show results on XSum, but those unfortunately don’t include ChatGPT).
> >
> > On the question of sentence-level vs paragraph-level, I understand the distinction between your sentence-level and paragraph-level tasks. My point was that it’s relatively easy for LLMs to generate correct sentences, and hence more difficult to detect human vs LLM sentences. In fact, really good performance at the sentence level is almost suspicious because one wouldn’t expect to be able to reliably detect sentences from a powerful LLM, except by using tricks like relying on data-dependent cues. Paragraphs are another matter because they give enough scope for detecting subtle inconsistencies.
> >
> > Achieving such good performance on EC3 with ChatGPT and a collection of older LLMs is an impressive feat, but I think more analysis (including some examples!) are needed to show that this performance will generalize to other settings.

---

> ### Author Response · Authors · 2023-11-23
> **Thanks to Reviewer bnWc [1/2]**
>
> Thank you for your detailed feedback and continued engagement with our work. We appreciate your positive remarks on the soundness of our method and the thoroughness of our response. We understand your concerns about the meaningfulness of our experiments and would like to address each of your points:
>
>
> >Q1. The main problem is that you achieve such good results when training only on ~3k paragraphs, and the results are better for ChatGPT than for weaker models. To me, this points to corpus-specific artifacts in HC3, perhaps because it is a question-answering setting, and ChatGPT has certain characteristic ways of answering the fixed pattern of questions in that corpus. I don’t think this would be representative of performance in a truly blind setting (in Appendix J you show results on XSum, but those unfortunately don’t include ChatGPT).
>
> **A1.** We appreciate your careful examination of our results and your insightful comments on the **training corpus** and **ChatGPT results on XSum**. To address your concerns, we provide the following clarification:
>
> * **Training corpus**: We evaluate our method on HC3 for two primary reasons: 1) To ensure a fair comparison with the baseline detection method, ChatGPT-D, which is based on fine-tuning on the HC3 dataset; 2) The HC3 dataset, sourced from diverse language domains and encompassing ChatGPT texts, provides a realistic representation for real-world evaluation. Additionally, we filter the shorter sentences or paragraphs for training, the remaining are ~3,000 paragraphs. Despite the smaller training set (3,100  or 1,000 paragraphs), our method consistently outperformed ChatGPT-D in detection performance, as demonstrated in Tables 3 and 4.
>
>     Moreover, to address your concern about corpus-specific artifacts, we also conduct the experiments on the XSum dataset as XSum involves diverse news articles and differs from the question-answering format of HC3. The results in Section J are consistent with those on HC3, demonstrating the superiority of our method compared with other baselines.
>
> * **Lack of ChatGPT results on XSum**: Although we conducted experiments on the XSum dataset with other baseline detection methods, we did not specifically test for detecting ChatGPT-generated texts on XSum. This is due to the **unavailability of ChatGPT-generated texts on XSum**. Our model and dataset are free and open-source, and while the HC3 dataset contains ChatGPT texts, XSum lacks such instances.
>
> We acknowledge the significance of evaluating our approach on more diverse datasets, and **we plan to conduct such evaluations on additional datasets in our revised version**.

---

> ### Author Response · Authors · 2023-11-23
> **Thanks to Reviewer bnWc [2/2]**
>
> >Q2. On the question of sentence-level vs paragraph-level, I understand the distinction between your sentence-level and paragraph-level tasks. My point was that it’s relatively easy for LLMs to generate correct sentences, and hence more difficult to detect human vs LLM sentences. In fact, really good performance at the sentence level is almost suspicious because one wouldn’t expect to be able to reliably detect sentences from a powerful LLM, except by using tricks like relying on data-dependent cues. Paragraphs are another matter because they give enough scope for detecting subtle inconsistencies.
>
> **A2.** Thank you for your constructive comments. We appreciate your understanding of the distinction between sentence-level and paragraph-level tasks in text detection. In our paper, we include both these two tasks, **recognizing the varying granularity and challenges introduced by LLMs in text detection**. This strategy can **provide a comprehensive evaluation of all text detection methods**. Additionally, we recognize the practical need for sentence-level detection [r1, r2] in certain scenarios, such as short video comment detection [r3, r4] or spam filtering [r2, r5]. By offering methods for different granularity levels, our approach ensures **adaptability to diverse detection scenarios**.
>
> [r1] SeqXGPT: Sentence-Level AI-Generated Text Detection, EMNLP 2023.
>
> [r2] An attention-based unsupervised adversarial model for movie review spam detection, TMM 2021.
>
> [r3] Fakesv: A multimodal benchmark with rich social context for fake news detection on short video platforms, AAAI 2023.
>
> [r4] Hateful Comment Detection and Hate Target Type Prediction for Video Comments, CIKM  2023.
>
> [r5] Collective Opinion Spam Detection: Bridging Review Networks and Metadata, KDD 2015.
>
> Moreover, while we recognize the challenges posed by the proficiency of large language models (LLMs) in generating accurate sentences, we would like to emphasize the effectiveness of our method in sentence-level detection.  This is attributed to the **powerful ability of the kernel-based Maximum Mean Discrepancy (MMD) to measure subtle distribution differences**. Moreover, our proposed optimization approach **enhances the stability of the training of the deep kernel**, resulting in a more comprehensive improvement in detection performance.
>
> If you have any specific suggestions or recommendations on how we could further address or enhance this aspect in our paper, we would be grateful to consider them.
>
> >Q3. Achieving such good performance on HC3 with ChatGPT and a collection of older LLMs is an impressive feat, but I think more analysis (including some examples!) are needed to show that this performance will generalize to other settings.
>
> **A3.** Thank you for recognizing the impressive performance achieved on HC3 with ChatGPT. We appreciate your suggestion for more analysis and examples to illustrate the generalizability of our results to other settings. To achieve this, **we have added a new section L in Appendix to include some examples and provided additional analysis** in the revised paper, aiming to give a clearer understanding of the robustness and applicability of our method. In our future work, we will explore the generalizability of our approach across more diverse LLM text settings.

---

### Author Response · Authors · 2023-11-20
**General Response**

Dear ACs and Reviewers,

We sincerely appreciate your time and effort in reviewing our paper and providing constructive feedback. Besides the response to each reviewer, here we would like to further 1) thank reviewers for their recognition of our work and 2) highlight the major modifications in our revision:

1. **We are glad that the reviewers appreciate and recognize our novelty and contributions.**
    * "MMD-MP is a **novel** approach, providing **a fresh perspective** on the problem"; "The paper's motivation for using MMD is **compelling**"; "It is **clearly written**." [4WLs, bnWc, DBzA]
    * "The paper offers **clear theoretical insights**, **further underscore the research’s depth** and provide **a strong theoretical foundation**"; "The paper provides **extensive theoretical justification**"; "The proposed method is **supported by theoretical analysis**"; "The paper provides **clear mathematical formulation**." [4WLs, bnWc, DBzA, 1AtS]
    * "The proposed method outperforms all baselines on these tasks, and it is **robust to generated text coming from multiple LLMs**"; "The experimental design is **comprehensive**"; "The proposed method achieves **better machine-generated text detection rate** than baselines". [bnWc, DBzA, 1AtS]
    * "The use of statistical properties and theoretical analysis makes the paper **robust and comprehensive**". [4WLs]
    * "Such **thorough testing** not only validates their methods but also **offers a benchmark for future research** in this domain".[4WLs]

2. **We summarize the main modifications in our revised paper (highlighted in blue).**
    * We add more intuitive explanations for the optimization of kernel-based MMD in Appendix D. [Reviewers 4WLs, bnWc]
    * We revise the statements of the caption in Figure 1. [Reviewer bnWc]
    * We add more detailed  descriptions about Algorithm 1, 2 and 3 (see Appendix C.2 in the revised paper). We also include more detailed descriptions about the training corpus (see Appendix C.1 in the revised paper). Last, we give more details about the kernel hyper-parameter selection in  Appendix C.3 of the revised paper. [Reviewer bnWc]
    * We add more results on paragraph-level sentence detection in Appendix K of the revised paper. From the new results, our methods still yield superior performance than baselines on paragraph-level sentence detection. [Reviewers bnWc, DBzA]
    * We add a new section on the related work of LLMs in Appendix B.1. [Reviewer 4WLs]
    * We provide some examples of ChatGPT text detection in Appendix L. [Reviewer bnWc]

Best regards,

The Authors

---

### Meta-Review · Area_Chair_D4Tg · 2023-12-15

**Metareview:**

The paper addresses the challenge of distinguishing between machine-generated and human-generated text. The authors propose using Maximum Mean Discrepancy (MMD) as a tool to identify subtle distributional discrepancies. To account for variations in text generated by different LLMs, they introduce a novel approach called multi-population aware optimization for kernel-based MMD (MMD-mp), with each population representing the distribution of one LLM. The MMD-mp method is applied in two settings: single sentence detection and paragraph detection, where individual sentences within a paragraph are treated as separate instances.
The paper attacks a crucial problem and it is clearly written. It has a very clear mathematical formulation of the MMD-MP method, as well as providing reasoning on why they choose to make this optimization on top of vanilla MMD. Results on the tested setting are positive.
Almost all concerns from the reviewers were addressed and the paper has been updated.

**Justification For Why Not Higher Score:**

A poster presentation is ideal for the topic and the contribution of the paper.

**Justification For Why Not Lower Score:**

There is no reason to not accept the paper.

---

### Decision · Program_Chairs · 2024-01-16

Accept (poster)